

# Mapping and characteristics of avalanches on mountain glaciers with Sentinel-1

Marin Kneib[1,2], Amaury Dehecq[1], Fanny Brun[1], Fatima Karbou[3], Laurane Charrier[1], Silvan Leinss[4], Patrick Wagnon[1], Fabien Maussion[2,5]

[1]Institut des Géosciences de l'Environnement, Université Grenoble-Alpes, CNRS, IRD, Grenoble, 38400, France
[2]Department of Atmospheric and Cryospheric Sciences, University of Innsbruck, Innsbruck, 6020, Austria
[3]Centre d'Etudes de la Neige, Université Grenoble-Alpes, CNRS, CNRM, Météo France, Grenoble, 38400, France
[4]GAMMA Remote Sensing, Bern, 3073, Switzerland
[5]School of Geographical Sciences, University of Bristol, Bristol, BS8 1QU, UK

*Correspondence to*: Marin Kneib (marin.kneib@gmail.com)

**Abstract.** Avalanches are important contributors to the mass balance of glaciers located in mountain ranges with steep topographies. They result in localised over-accumulation that is seldom accounted for in glacier models, due to the difficulty
to quantify this contribution, let alone the occurrence of avalanches in these remote regions. Here, we developed an approach to semi-automatically map avalanche deposits over long time periods and at scales of multiple glaciers, utilising imagery from Sentinel-1 Synthetic Aperture Radar (SAR). This approach performs particularly well for scenes acquired in winter and in the morning, but can also be used to identify avalanche events throughout the year. We applied this method to map 16,302 avalanche deposits over a period of five years at a 6 to 12 days interval over the Mt Blanc massif (European Alps), the Everest
(Central Himalaya) and Hispar (Karakoram) regions. These three survey areas are all characterised by steep mountain slopes, but also present contrasting climatic characteristics. Our results enable the identification of avalanche hotspots at the surface of these glaciers and allow us to quantify the avalanche activity and its spatio-temporal variability across the three regions. The avalanche deposits are preferentially located at lower elevations relative to the hypsometry of the glacierized catchments, and are also constrained to a smaller elevation range at the Asian sites, where they have a limited influence on their extensive
debris-covered tongues. Avalanche events coincide with solid precipitation events, which explains the high avalanche activity in winter in the Mt Blanc massif and during the monsoon in the Everest region. However, there is also a time lag of 1-2 months, visible especially in the Everest region, between the precipitation and avalanche events, indicative of some snow retention on the mountain headwalls. Ultimately, this study provides critical insights into these mass redistribution processes as well as tools to account for their influence on glacier mass balance.



## 1 Introduction

Mountain glaciers usually gain mass via solid precipitation falling in their accumulation area that is then advected downstream with ice flow, so the mass balance is traditionally expected to increase with elevation (Benn and Lehmkuhl, 2000). For catchments with strong topographic gradients, there can be large mass inputs from mountain headwalls at localised portions

of the glacier, both in the accumulation and ablation zones, which leads to non-linear patterns of glacier surface mass balance (Miles et al., 2021; Kirkbride and Deline, 2013; Brun et al., 2019). Avalanches, defined here as the process of gravitational mass redistribution (in the form of snow, ice or rocks) to lower elevation from surrounding slopes, are important contributors to the mass balance of glaciers (Benn and Lehmkuhl, 2000; Laha et al., 2017). These inputs, which vary in size and originate from the redistribution of snow or ice from mountain headwalls or hanging glaciers, contribute to the persistence of glaciers

at low altitudes (Hughes, 2008; DeBeer and Sharp, 2009; Carturan et al., 2013) and could therefore, to some extent, buffer the depletion of mountain water resources (Burger et al., 2018). Such buffering effect is however strongly dependent on the mass supply from avalanches, and small variations in this supply may have important consequences for the overall glacier mass balance (Purdie et al., 2015). Furthermore, the presence or the absence of avalanches on a glacier may influence the interpretation of the glacier boundaries, which are known to vary considerably depending on the method or the definition

applied (Kaushik et al., 2022; Nuimura et al., 2015).

Due to the lack of field or remote sensing observations, the temporal and spatial occurrence of these events is however generally unknown, as is their relative contribution to glacier mass balance (Laha et al., 2017). We expect avalanches in glacierized catchments to differ at least partly from off-glacier snow avalanches. Indeed, one can expect a different seasonality in these

avalanches, as snow can accumulate even during the melt season at the elevations of the accumulation areas. Furthermore, these gravitational mass contributions are not limited to snow avalanches but also likely include wind-blown snow from steep headwalls (Sommer et al., 2015), ice avalanches from seracs or hanging glaciers (Pralong and Funk, 2006) or rock avalanches that are suspected to contribute to the development of on-glacier debris cover (Berthier and Brun, 2019; Scherler and Egholm, 2020; McCarthy et al., 2022). Such processes can to some extent be represented implicitly in glacio-hydrological models using

flow-routing algorithms of excess snow (Gruber, 2007; Bernhardt and Schulz, 2010; Mimeau et al., 2019), but these parameterizations are often difficult to calibrate and rely on a limited number of avalanche outlines from a small number of optical images (Bernhardt and Schulz, 2010; Ragettli et al., 2015).

Indeed, very little data exists in remote glacierized mountain catchments on the occurrence of such avalanche events, contrary

to populated valleys where they are monitored, generally based on field observations, for hazard management (Maggioni and Gruber, 2003; Schweizer et al., 2020; Bourova et al., 2016). This is even more true in remote ranges of High Mountain Asia (HMA), despite a number of recent efforts to quantify the avalanche activity in parts of the range devoid of long-term avalanche monitoring (Caiserman et al., 2022; Singh et al., 2022; Acharya et al., 2023). Several strategies have been proposed to derive





hazard maps in such a data-scarce region. Some recent catastrophic events such as the extreme avalanches and landslides

triggered by the 2015 Gorkha earthquake in Nepal have been carefully mapped and analysed (Kargel et al., 2016; Fujita et al., 2017), but they do not allow consistent hazard assessment. Recent promising efforts have used end-of-season optical satellite images to derive inventories of major avalanche deposits (Caiserman et al., 2022; Singh et al., 2022), which has the advantage of providing a spatially unbiased dataset, but remains limited to the largest deposits and does not give any information on the temporal variability of these events. More generally, it is possible to identify avalanche deposits in very high-resolution (<5m)

images taken within a few days from one another (Lato et al., 2012; Bühler et al., 2009) based on surface texture changes, but these approaches are hindered by the dependence on the availability of cloud-free acquisitions which need to be tasked, thus limiting them to small regions and targeted time periods (Hafner et al., 2021; Eckerstorfer et al., 2016). These data limitations highlight the need for quantitative inventories of avalanche events, with as little spatial and temporal bias as possible. This is becoming a possibility thanks to the use of optical and Synthetic Aperture Radar (SAR) satellite products, Sentinel-1

especially, which currently allow the near real-time inventory of avalanches across mountain ranges at high temporal resolution (Eckerstorfer et al., 2019).

In recent years, numerous approaches have been developed to detect avalanche deposits from freely-available Sentinel-1 SAR satellite data (Vickers et al., 2016; Eckerstorfer et al., 2019; Abermann et al., 2019; Karas et al., 2022). These methods rely on

the detection of increases in the backscatter between two successive images caused by the increase in surface roughness at the location of the avalanche deposits (Leinss et al., 2020; Wesselink et al., 2017). Such approaches have been applied at various spatial and temporal scales, and are now implemented across entire regions at an operational level (Eckerstorfer et al., 2019; Karas et al., 2022). The validity of these approaches has been demonstrated by the comparison with high-resolution optical and field observations (Leinss et al., 2020; Hafner et al., 2021). Sentinel-1 satellites have a high repeat frequency and are

unaffected by clouds, making them a promising way to derive avalanche characteristics in data-scarce regions (Yang et al., 2020). There remain limitations to these approaches, especially as they fail to detect smaller events (<4000 m$^2$) or have a high rate of false detections in the case of transitions from wet to dry snow that also result in increasing the SAR backscatter (Eckerstorfer et al., 2019). Even though initial observations seem to confirm the ability of such approaches to identify large avalanches in glacierized environments (Leclercq et al., 2021), this on-glacier avalanche detection potential remains to be

assessed quantitatively. Furthermore, the sensitivity of the method to image repetition, e.g. 6 days in Europe vs 12 days in HMA, has not been assessed yet.

Here, we develop a new approach to semi-automatically derive avalanche deposits from Sentinel-1 images and apply it to a full five-year period across three glacierized regions with different topo-climatic characteristics, in the European Alps, the

Central Himalaya and the Karakoram. Our goal is to evaluate the suitability of this method to map on-glacier avalanches on a broad scale and to derive the main spatio-temporal characteristics of the identified deposits in these three regions. To this end we (1) calibrate and evaluate our automated mapping approach at each site and assess its transferability to other sites, (2)



extract the size-frequency characteristics of avalanches at various spatial scales over a period of five years and (3) evaluate the implications for the glacier mass balance.

**2 Data**

We focus on three survey areas located in the European Alps (Mt. Blanc massif; Fig. 1b), the Central Himalaya (Everest region; Fig. 1a) and the Karakoram (Hispar region; Fig. 1c). All three zones are characterised by a large number of glaciers and by a relatively steep topography with more than 50% of the slopes steeper than 30° in the glacierized catchments (Fig. 1d-e), which we defined as the area covered by the glaciers and their upstream area. This is indicative of a strong avalanche potential

(Hughes, 2008; Laha et al., 2017). These three zones are located in contrasting climatic regimes. The Everest region receives most of its precipitation during the monsoon season, which is also the warmest period of the year (Wagnon et al., 2013, 2021), leading to summer-type accumulation glaciers. The more westerly-driven climate in the Karakoram results in more temporally-distributed precipitation over the Hispar region, with more important snowfall in the winter (Li et al., 2020; Shaw et al., 2022). The Mt Blanc massif, in the European Alps, also receives most of its solid precipitation in the winter (Vionnet et al., 2019).


For each survey domain we derived the entire time series of Sentinel-1 images for the period 11/2017 - 10/2022 for the two sites in HMA along two ascending and descending orbits, and the period 11/2016 - 10/2021 for the Mt Blanc region. Indeed, Sentinel-1B experienced malfunction in December 2021 and the acquisition frequency dropped from 6 to 12 days over the European Alps (Table 1). This had little impact for the HMA sites, which had been monitored almost solely by Sentinel-1A,

and only from the second half of 2017 at regular time intervals. Despite systematic acquisition strategy, there were a few gaps (<10%) in the time series of the Mt Blanc and Everest regions, which were more important in the descending acquisitions over Hispar (65% gaps, with no images from October 2020 onwards, Table 1). For all three survey domains the ascending acquisitions were made late in the afternoon and the descending acquisitions early in the morning (Table 1).





**Figure 1: The different survey domains to which were applied the avalanche mapping (a-c). RGI 6.0 outlines are shown in black, the mapping extents for the ascending (resp. descending) scenes are shown in blue (red). The red triangle in (a) indicates the location of the Pyramid precipitation gauge. Background images are the AW3D30 30m multidirectional hillshades. (d) Hypsometry of all glacier catchments of the three survey domains and (e) their proportion of slopes steeper than 30° per elevation bands.**

**Table 1: Characteristics of the Sentinel-1 acquisitions in the ascending and descending orbits for each of the three survey domains.**

| S1 scenes | Study period | Relative orbit | Revisit time | Acquisition time | Number of image pairs | Temporal gaps | Training period | Number of image pairs used for validation/calibration |
|---|---|---|---|---|---|---|---|---|
| Mt. Blanc ASC | 11/2016 - 10/2021 | 88 | 6 days | 19:30 (UTC+02:00) | 288 | 6% | | 29/30 |
| Mt. Blanc DESC | | 66 | | 07:30 (UTC+02:00) | 287 | 6% | | 30/30 |



| Everest ASC | 11/2017 - 10/2022 | 12 | 12 days | 18:00 (UTC+05:45) | 143 | 7% | 11/2019 - 10/2020 | 14/15 |
|---|---|---|---|---|---|---|---|---|
| Everest DESC | | 121 | | 06:00 (UTC+05:45) | 147 | 4% | | 15/16 |
| Hispar ASC | 11/2017 - 10/2022 | 27 | 12 days | 18:00 (UTC+05:00) | 146 | 5% | | 14/15 |
| Hispar DESC | | 34 | | 06:00 (UTC+05:00) | 54 | 65% | | 7/7 |

In addition to the Sentinel-1 time series, we used four cloud-free Pléiades orthoimages acquired over the Mt Blanc massif with a spatial resolution of 0.5 metres. Two images were taken during winter (08/12/2020 and 19/01/2021) and the two others during summer (08/07/2020 and 09/08/2020), and they were used to derive high precision avalanche deposits to evaluate the outlines obtained with Sentinel-1.

The characteristics of the avalanche deposits (size, elevation, slope), were derived using the global AW3D30 30m DEM (Tadono et al., 2014). The avalanche time series obtained were also compared to the precipitation time series over the different study areas, as an indication of the amount of snow deposited at high elevations. For the Mt Blanc massif we used the rainfall and snowfall at 3000 m a.s.l from the SAFRAN reanalysis product (Vernay et al., 2022). For the Everest region we used precipitation measurements from the Pyramid precipitation gauge (Fig. 1a) with a Geonor sensor using a weighing device suitable to measure liquid and solid precipitation (Khadka et al., 2022) located at 5035 m a.s.l on the southern side of the survey domain. No station data was available for the Hispar region so we used precipitation from the ERA5-Land reanalysis (Muñoz Sabater, 2019).

## 3 Methods

### 3.1 Image pre-processing

All images were pre-processed in Google Earth Engine, using the S1 GRD (Ground Range Detected) library (Gorelick et al., 2017). We filtered the images per orbit and kept only one ascending and one descending orbit per survey area to have observations at regular intervals (6 days for Mt Blanc, 12 days for Everest and Hispar). We conducted all the processing steps independently for the ascending and descending acquisitions. Images were mosaiced per day in case of overlapping images. We applied a 500m high pass filter to reduce the influence of large-scale snow wetness changes and averaged the VV and VH polarizations to reduce the speckle (Leinss et al., 2020). The backscatter values were then clamped to [-25; -6] dB, a range beyond which we do not expect to observe changes in the backscatter caused by changes in the snow surface roughness, and normalized to [0, 1] (Fig. 2). The images were then combined into RGB composites, with the backscatter of the D image



(image taken on the day of interest) stored in the green channel and the D-i image (first image taken prior the day of interest, i is equal to 6 or 12 days depending on the domain) stored in the red and blue channels. This enabled the identification of increases in the backscatter as green and decreases as purple (Fig. 3). We downloaded the first GRD images of each orbit from the Alaska Satellite Facility to produce a mask of shadow and layover using the ESA SNAP software. These masks were

extended to all locations where the mean annual backscatter (brightness) was lower than 0.1 or higher than 0.82 or outside the RGI 6.0 glacier extents (RGI Consortium, 2017) plus a 200 m buffer (Fig. 1).

### 3.2 Avalanche mapping

The mapping approach that we developed is adapted from the method by Karas et al. (2022) and as such uses the RGB images converted to HSV (Hue, Saturation, Value) space. This approach uses minimum Saturation and Value thresholds to determine

if the green patches in the image (which indicate an increase in the backscatter) should be classified as avalanche deposits. By targeting the saturation and brightness of these green patches, this approach is well suited to identify avalanche deposits in RGB images (Karas et al., 2022).

In this approach, which targeted the mapping of avalanche deposits over a multi-year period, we normalised the Saturation and

Value by the mean values of the first images of the time series to improve the temporal consistency of the signal. We used a 35° slope threshold above which the increases in backscatter were not considered to be avalanches, and removed all detections smaller than 40 pixels (4000 m$^2$; Leinss et al., 2020; Eckerstorfer et al., 2019). Furthermore, in addition to the two thresholds on Saturation and Value proposed by Karas et al. (2022), we added extra constraints to reduce the effect of changes in snow wetness which would otherwise lead to a large amount of false positive detections. First, once the bright green patches had

been detected, we allowed them to expand within a vicinity of 7 pixels (70 m) to capture less bright parts of the avalanche deposit according to another threshold value $T_O$, identical for both the Saturation and Value (Table 1). Second, we directly differentiated the image at D with high pass filtered images at D and D-i. The high pass filter consisted of a 45 pixel (450 m) wide Gaussian filter. We kept only pixels for which at least one of the differences was above set thresholds ($T_{D1}$, $T_{D2}$ and $T_{D3}$, 3$^{rd}$ filtering step, Fig. 2). The idea of this additional step being that an avalanche event results in a spatial discontinuity in the

backscatter, if not with the image before, at least in the current image.



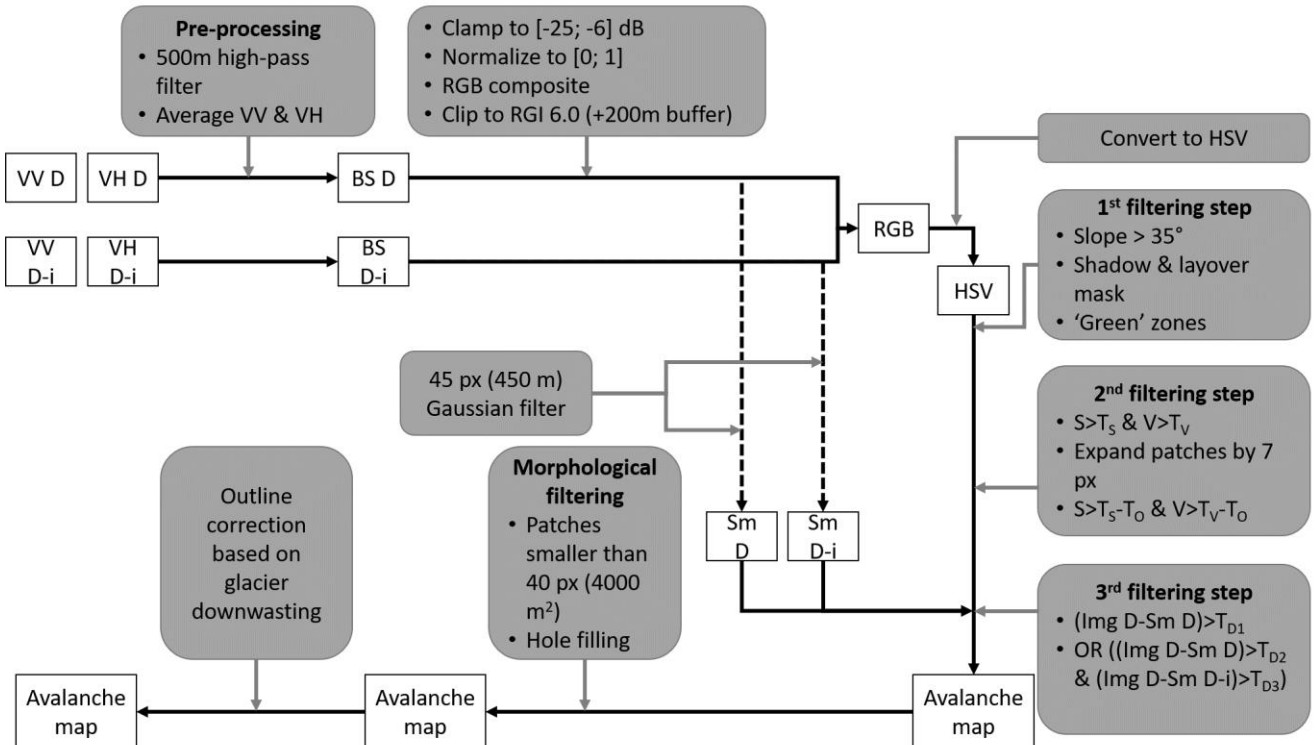

**Figure 2: Processing steps (grey) applied to the Sentinel-1 GRD data (white) to obtain avalanche maps. The polarizations of images at D and D-i are averaged to get backscatter (BS) images which are then combined into an RGB and then an HSV image. These HSV images are then filtered following three filtering steps using six different thresholds ($T_S$, $T_V$, $T_O$, $T_{D1}$, $T_{D2}$ and $T_{D3}$), before the final morphological filtering step and correction for glacier elevation change.**

### 3.3 Threshold calibration

We manually derived the avalanche deposits outlines of all images between November 2019 and October 2020 at all sites, based on the pre-processed RGB images. The main advantage of the manual mapping is that it gives the possibility to account for the shape of the events to discriminate avalanche deposits from changes in snow wetness, for example (Vickers et al., 2016; Eckerstorfer et al., 2016; Hafner et al., 2021). A single operator performed the manual detection, and to account for biases in the delineations, we compared these outlines with those of four other operators for 4 scenes (2 ascending and 2 descending) covering the Mt Blanc region and 4 scenes covering the Everest region (Kneib et al., 2021; Table S1, Fig. S1-S2).

The manual outlines were used to calibrate and validate the six free parameters ($T_S$, $T_V$, $T_O$, $T_{D1}$, $T_{D2}$ and $T_{D3}$) used for the mapping (Fig. 2). We used the Dice coefficient, also known as F1-score, as a metric to quantify the goodness-of-fit of the automated delineation (Dice, 1945):

$$Dice = \frac{2TP}{2TP+FP+FN},$$

(1)



Where TP is the number of pixels classified as true positives, FP as false positives and FN as false negatives (Fig. 3). This metric is therefore well suited when the mapping targets represent a small percentage of the total area of the scene, and a calibration based on this metric will result in finding the parameters that lead to maximising the number of true positives while also balancing the number of false positives and false negatives (Kneib et al., 2020). For a perfect classification, the Dice is equal to 1.

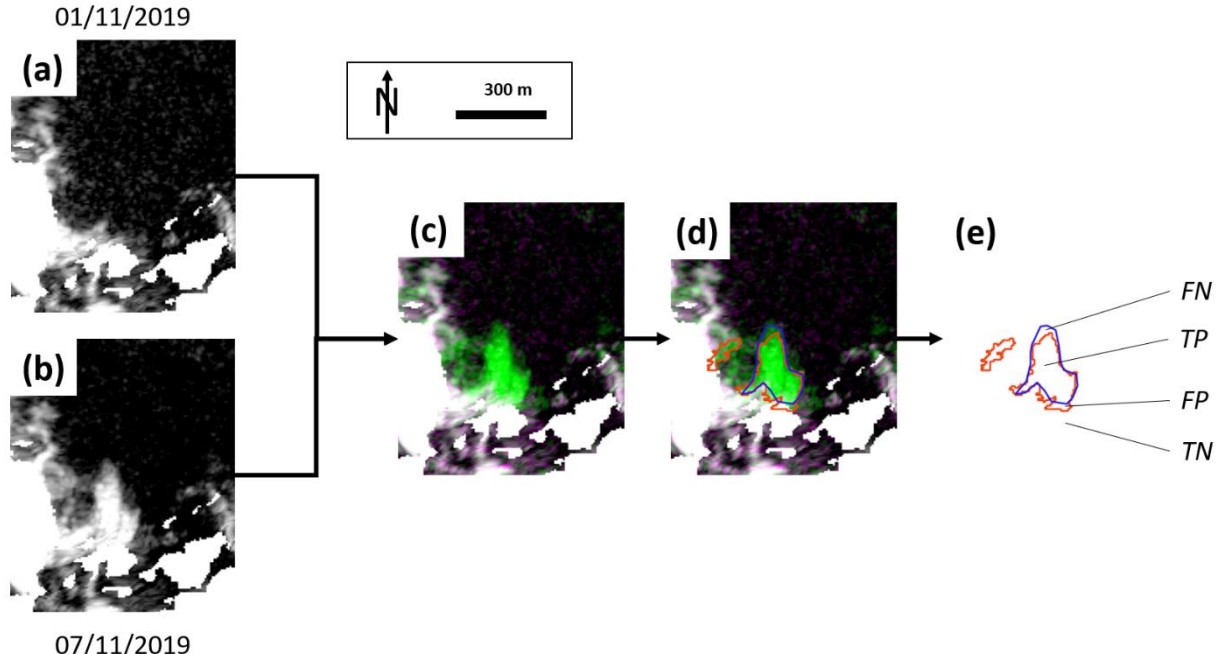

**Figure 3: Example of the different processing steps from two pre-processed Sentinel-1 images taken at a 6 days interval (a-b), combined into one RGB image for change detection (c). The different bands range between -25 and -6 dB. This image is then used for the manual (blue outlines) and automated (red outlines) mapping of the avalanche deposits that appear in green (d). These outlines are then compared based on the confusion matrix, used to compute the Dice coefficient, TN corresponding to the true negative pixels, TP to the true positive pixels, FP to the false positive pixels and FN to the false negative pixels (e).**

We used every second image pair for the calibration and the remaining half was used for validation. We split the time series into two time periods, November-April and May-October to account for lower backscatter values across large portions of the glaciers during the melt season, which we considered to be bounded by the May-October period for all survey domains (Karbou et al., 2021; Scher et al., 2021). Thus, the calibration and validation were done independently for each ascending and descending orbit of each survey domain and for each time period. We started from an initial guess of the parameter values based on trial and error and then sampled the parameter space within reasonable bounds (values within +/- 0.5 of the best set of values obtained from trial and error) using a Monte Carlo approach to randomly select the parameter sets (Fig. 2). Ultimately, for each survey area and each orbit we choose the set of parameters that maximised the Dice coefficient. This parameter selection was then evaluated against the validation set and used to automatically map avalanche deposits across the entire Sentinel-1 time series.





Of all six parameters used for the calibration, the saturation threshold $T_S$ was the only one with a defined value maximising the Dice coefficient, between 0.3 and 0.5 (Fig. 4, Fig. S3-7), and therefore also the most sensitive. The other parameters did not have a clear maximum defined and several combinations of these parameters could lead to similarly high Dice values (Fig. S3-7).

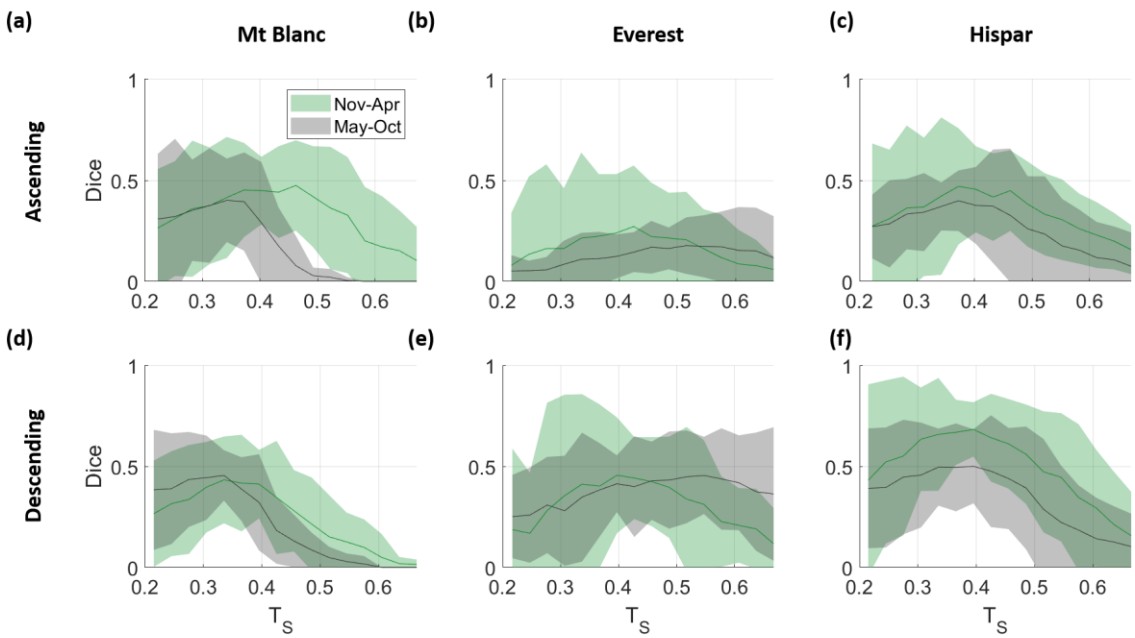

**Figure 4: Mean value +/- 3 standard deviations of Dice as a function of values of $T_S$ taken between 0.2 and 0.65 for ascending and descending orbits of all three survey areas.**

### 3.4 Application to entire Sentinel-1 time series

After calibration and validation of the mapping approach, we applied it to a five-year time series of Sentinel-1 images over the three survey domains (Table 1). All Sentinel-1 images were pre-processed in Google Earth Engine. False positive (including

from crevassed areas, changes related to snow wetness, vegetated areas, frozen supra glacial lakes) and false negative detections were corrected manually to obtain a dataset comparable to the 11/2019-10/2020 calibration/validation dataset. The Google Earth Engine Sentinel-1 images are map projected using the SRTM DEM, so we had to account for glacier elevation change by shifting the outlines based on the local elevation change rates from Hugonnet et al., (2021), as well as the Sentinel-1 look and heading angles for each orbit (Fig. S8). While these shifts were negligible in the accumulation area of most glaciers,

they reached values of 5 m yr$^{-1}$ in the lower ablation zone of the glaciers in the Mt Blanc, which had the highest surface lowering rates. The final outlines were aggregated into avalanche 'activity' maps indicating the avalanche frequency for the different avalanche deposits.





### 3.5 Comparison with optical images

We compared the Sentinel-1 outlines that occurred over given periods in the summer and in the winter with manually derived
outlines of avalanche deposits from high resolution (0.5 m) Pléiades orthoimages over part of the Mt Blanc survey area,
acquired on 08/12/2020, 19/01/2021, 08/07/2020 and 09/08/2020. We also compared the aggregation of one year (11/2019-
09/08/2020) of Sentinel-1 manual outlines from ascending and descending orbits with all the avalanche deposits identified in
a Pléiades image taken at the end of the summer season (09/08/2020), with the assumption that these end-of-summer deposits
result from the union of all individual deposits throughout the year. This comparison was made for all deposits above 2700 m
a.s.l, which was the altitude of the snow line, derived from the Pléiades orthoimage. We also restricted the comparison to
locations with slopes lower than 35° and within the ascending or descending mapping extents (Fig. 1). We attempted to do the
same over the Everest survey domain using 5 m resolution Venus multi-spectral images (Raynaud et al., 2020) but found that
the spatial resolution was not high enough to outline the deposits with a high enough confidence.

### 3.6 Characterization of avalanche activity

We quantified the avalanche influence on a given glacier using different metrics. We defined the proportion of avalanche
deposits as the area union of all detected deposits in the ascending or descending orbits relative to the glacier size, and the
avalanche activity as the cumulative area of all avalanche deposits in the ascending and descending orbits relative to the glacier
total area. We also defined a catchment for each glacier by taking all its upstream area following the D-infinity method
(Schwanghart and Scherler, 2014). We could then calculate for each glacier the ratio (R) of the area of the catchment with
slopes steeper than 30°, which stands as a proxy for the avalanche contribution area, and the glacier area (Hughes, 2008; Laha
et al., 2017).

## 4 Results

### 4.1 Avalanche mapping

#### 4.1.1 Sentinel-1 avalanche mapping potential

The comparison of the aggregation of one year of Sentinel-1 manual outlines with all the deposits identifiable in the end-of-
summer Pléiades scene above 2700 m a.s.l results in a Dice value of 0.47, with a majority of false negatives (Fig. 5). A large
amount of undetected Pléiades deposits are actually smaller than the Sentinel-1 detectability threshold of 4000 m2, although
removing them does not change the comparison (Dice value of 0.49) between the Pléiades and aggregated Sentinel-1 deposits.

The comparison of the manually derived Sentinel-1 deposits with the Pléiades deposits detected over time periods of ~1 month
in the winter (dates) and summer (dates) seasons gives more insights on the potential of Sentinel-1 images to identify particular
deposits (Fig. 5). It indicates locations of very good agreement, usually for large deposits with a lot of surface texture (Fig.





5b), but also false positive detections, for example caused by the opening of crevasses (Fig. 4c), as well as false negatives (Fig.

5d), often related to the relatively small size of the events, or to the fact that the roughness changes remained low due to the

low cohesion of the snow. This was particularly the case for the winter-season comparison, where very large (up to 60000 m$^2$)

avalanche deposits were identified in the Pléiades orthoimage, but not in the Sentinel-1 RGB pairs (Fig. 5e). This comparison

also highlights the important advantage of the Sentinel-1 detections which enable constraining the timing of the events within

a regular 6-day (for the European Alps) time window.

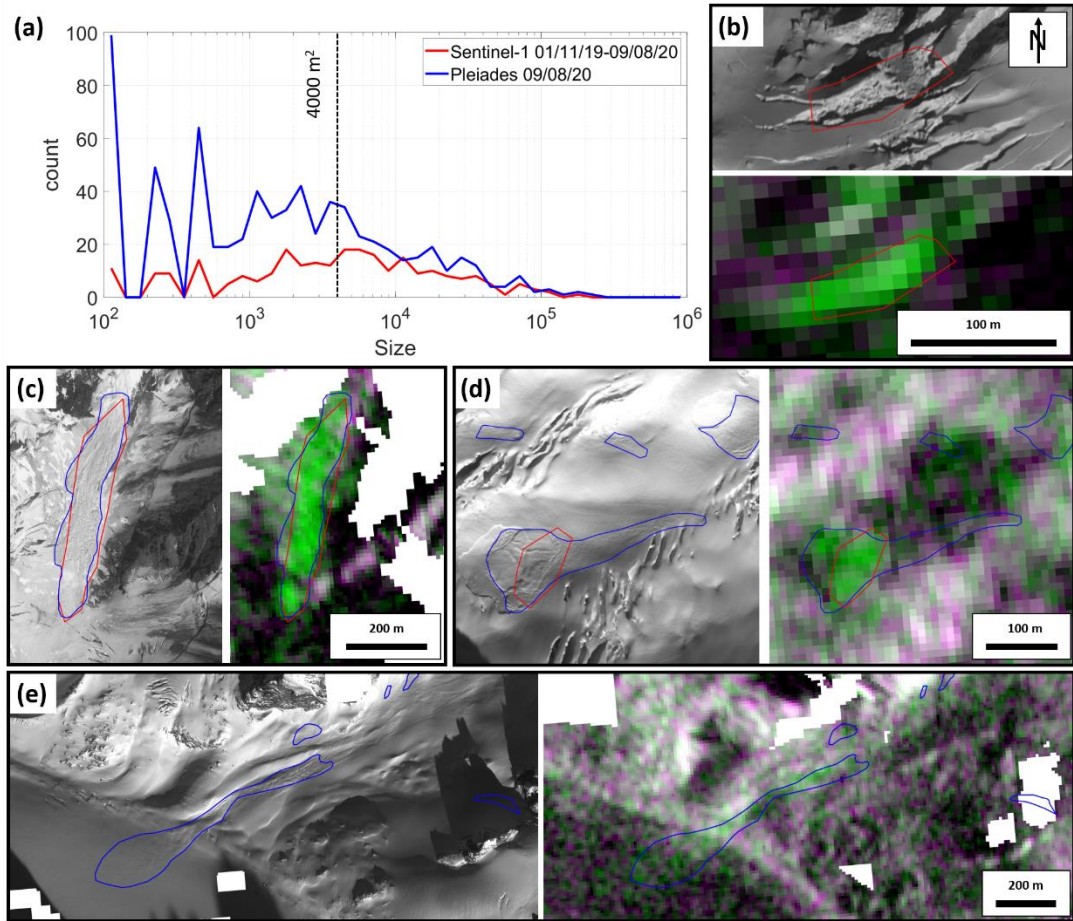

**Figure 5: (a) Size comparison of the aggregation of all the Sentinel-1 deposits (ascending and descending orbits) for the period
01/11/2019-09/08/2020 with all deposits identifiable in the 09/08/2020 Pléiades orthoimage. (b-e) examples of manual avalanche
detections in the Sentinel-1 (red) and Pléiades (blue) images (© CNES, distribution AIRBUS DS): (b) false positive detection of an
opening crevasse in Sentinel-1, (c) large avalanche deposit clearly visible in both Pléiades and Sentinel-1 imagery, (d-e) Dry snow
avalanches clearly identifiable in the Pléiades images, but only the deposits with high surface roughness are visible in the Sentinel-1**

**RGB images.**

We evaluated the presence of potential biases in the manual delineation of avalanche deposits in the Sentinel-1 images by

comparing the manual outlines from four different independent operators for eight randomly chosen Sentinel-1 image pairs

with different orbits taken during different seasons, covering both the Mt Blanc and the Everest regions (Table S1, Fig. S1-





S2). We directly compared the manual outlines from the operator who derived the manual datasets for the whole period
11/2019-10/2020 for all three sites, with the consensus outlines from the other three operators, which were the outlines for
which at least two operators agreed (Kneib et al., 2021). The outlines used for the calibration and validation of the automated
mapping approach lead to less avalanche detections (-29% +/- 36% of events detected and -46% +/- 27% of deposit areas) than
the consensus outlines, and can therefore be considered as a lower bound for the manual detection of avalanches in the Sentinel-
1 RGB pairs.

**4.1.2 Evaluation of the automated mapping approach**

We obtained Dice values ranging between 0.29 and 0.78 when calibrating the mapping parameters against the manually derived
outlines (Table 2). The Dice values are similar for both calibration and validation sets, which indicates the good transferability
of the parameters between scenes taken during the same season and with the same orbit. Dice values are generally lower for
the ascending orbits (average Dice of 0.47) compared to the descending ones (0.62) and for the warm season (0.49) compared
to the cold season (0.60). Except for the Everest ascending scenes, the Dice values obtained for the calibration were always
higher than 0.49.

**Table 2: results of the calibration and validation of Sentinel-1 avalanche outlines for the period 11/2019-10/2020 for each of the three
survey areas. The values of the calibrated parameters are indicated along with the Dice values obtained for the calibration and
validation sets. For each parameter the minimum value obtained is indicated in cyan and the maximum in magenta. Dice values are
written in blue when higher than 0.5, and in orange when lower.**

| Survey area | Path | Season | $T_O$ | $T_S$ | $T_V$ | $T_{D1}$ | $T_{D2}$ | $T_{D3}$ | Dice calibration | Dice validation |
|---|---|---|---|---|---|---|---|---|---|---|
| Mt Blanc | Descending | November-April | 0.15 | 0.31 | 0.73 | 0.08 | 0.01 | 0.30 | 0.56 | 0.53 |
| | | May-October | 0.13 | 0.33 | 0.65 | 0.07 | 0.02 | 0.39 | 0.56 | 0.54 |
| | Ascending | November-April | 0.09 | 0.36 | 0.51 | 0.11 | 0.06 | 0.41 | 0.54 | 0.51 |
| | | May-October | 0.07 | 0.31 | 0.61 | 0.10 | 0.07 | 0.42 | 0.49 | 0.36 |
| Everest | Descending | November-April | 0.07 | 0.33 | 0.67 | 0.07 | 0.06 | 0.34 | 0.67 | 0.68 |
| | | May-October | 0.01 | 0.44 | 0.26 | 0.09 | 0.01 | 0.35 | 0.53 | 0.50 |
| | Ascending | November-April | 0.17 | 0.47 | 0.68 | 0.07 | 0.04 | 0.29 | 0.39 | 0.45 |
| | | May-October | 0.15 | 0.55 | 0.38 | 0.11 | 0.08 | 0.42 | 0.29 | 0.34 |
| Hispar | Descending | November-April | 0.04 | 0.30 | 0.65 | 0.03 | 0.06 | 0.33 | 0.78 | 0.78 |
| | | May-October | 0.04 | 0.32 | 0.40 | 0.11 | 0.00 | 0.27 | 0.59 | 0.59 |



| | Ascending | November-April | 0.04 | 0.30 | 0.60 | 0.06 | 0.04 | 0.38 | 0.64 | 0.55 |
|---|---|---|---|---|---|---|---|---|---|---|
| | | May-October | 0.06 | 0.38 | 0.42 | 0.13 | 0.01 | 0.26 | 0.49 | 0.37 |

Local increases in the Sentinel-1 backscatter that are discarded in the manual delineation but that can be detected as false positives in the automated approach can be linked to snow wetness changes, especially during the May-October season (Fig. S9a) or calving into proglacial lakes (Fig. S9b). Conversely, the automated approach could miss events which had backscatter

values below the imposed thresholds but had the obvious shape of an avalanche (Fig. S9c). Such false positive or false negative detections were manually removed or added based on considerations of shape, size and location, and this manual filtering was applied to all time series of all survey domains for the results presented in sections 4.2 and 4.3. Over the entire automatically derived dataset we removed 36% of the mapped deposits and added 41% of what we considered were false negatives (Fig. S10). Furthermore, we also observed that deposits with a high avalanche activity remained with a high backscatter value for

time periods of several months during which there is not enough time, surface melt or precipitation for the surface roughness of the deposits to change significantly between two Sentinel-1 acquisitions. The only way that avalanches can be detected on such deposits is when they are large enough to have their runout zone go beyond the previous avalanche deposits (Fig. S9d). Therefore, for many deposits across the three survey domains, the frequency and size of avalanche events is likely to be underestimated.


We compared the total size and number of manually and automatically derived avalanche events for the validation image sets over the 11/2019-10/2020 period (Fig. 6, S11-S13). There is a relatively good correspondence between the two categories for the Mt Blanc as well as the Everest and the Hispar regions during the cold and warm seasons, with Pearson's correlation coefficients higher than 0.85 for the total size, 0.71 for the number of detected deposits. The automated mapping generally

underestimates the number and sizes of the avalanche deposits, especially in the May-October season, which is due to conservative thresholds to reduce the false positive detections of snow wetness changes (Fig. 6, S11-S13).



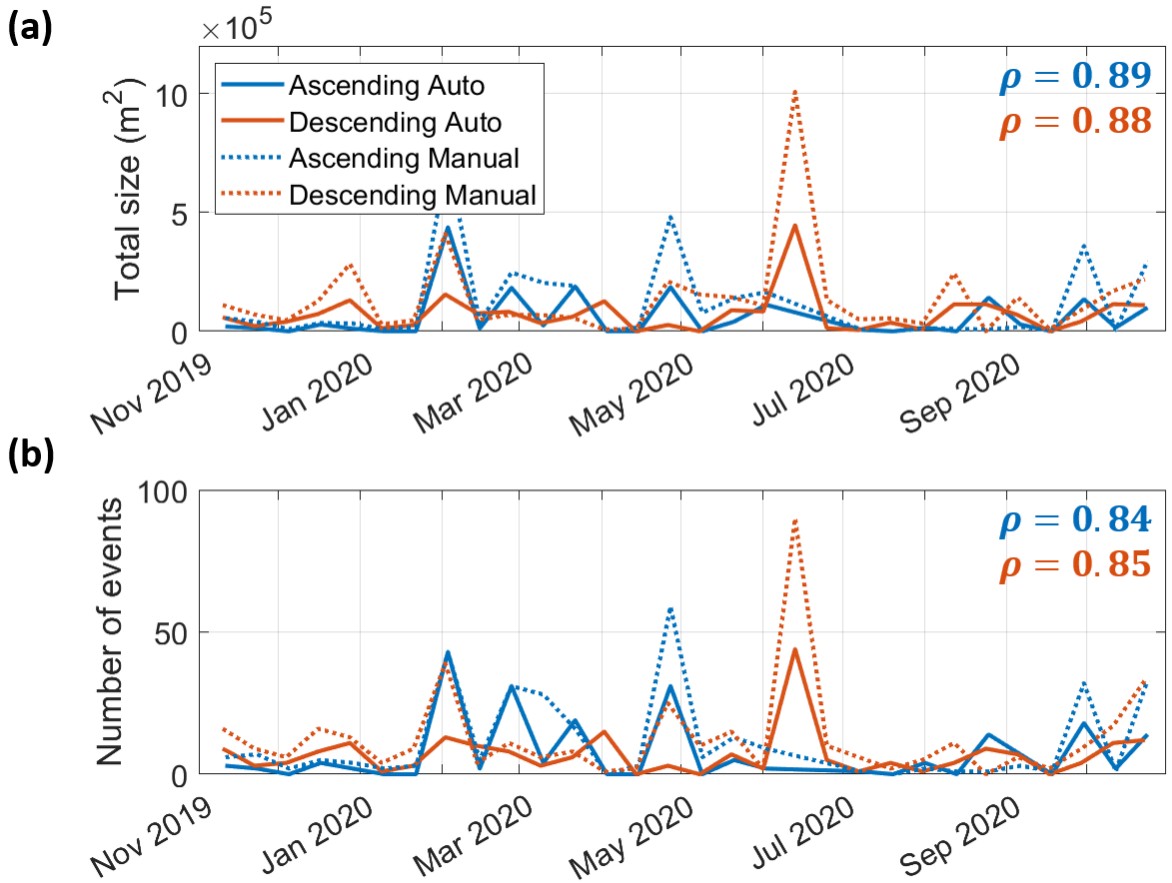

**Figure 6: Total size and number of manually and automatically detected avalanche events as a function of time for the period 11/2019-10/2020 for the validation datasets of Mt Blanc. The Pearson's correlation coefficients are indicated in blue (ascending) and red (descending).**

### 4.1.3 Transferability of the automated mapping parameters

To test the transferability of the calibrations obtained for the different orbits and periods of the different survey domains, we applied these parametrizations to the other survey domains, including to the Mt Blanc scenes with a 12-day interval (Fig. 7). Most parameterizations are well transferable to the Hispar and Everest November-April descending scenes and to the Hispar May-October descending scenes with Dice scores generally above 0.5 (0.6 for the Hispar November-April scenes). The ascending scenes present in general lower Dice values (often lower than 0.5), particularly the May-October scenes of Everest for which the Dice coefficient never exceeds 0.32. With an average Dice coefficient of 0.46, the Everest descending November-April parameter set is the most transferable, but still performs poorly (Dice <0.4) for some of the ascending and/or May-October scenes (Fig. 7).





| Dice coefficient | | | Parameters | | | | | | | | | | | | |
| --- | --- | --- | --- | --- | --- | --- | --- | --- | --- | --- | --- | --- | --- | --- | --- |
| | | | Everest | | | | Hispar | | | | Mt Blanc (6 days) | | | | Median |
| 0.4 0.5 0.6 | | | DESC | | ASC | | DESC | | ASC | | DESC | | ASC | | |
| | | | N-A | M-O | N-A | M-O | N-A | M-O | N-A | M-O | N-A | M-O | N-A | M-O | |
| Everest | DESC | Nov-Apr | 0.65 | 0.40 | 0.56 | 0.42 | 0.63 | 0.50 | 0.56 | 0.51 | 0 | 0.50 | 0.50 | 0.53 | 0.57 |
| | | May-Oct | 0.28 | 0.51 | 0.36 | 0.48 | 0.26 | 0.39 | 0.29 | 0.41 | 0.27 | 0.29 | 0.42 | 0.32 | 0.34 |
| | ASC | Nov-Apr | 0.37 | 0.22 | 0.40 | 0.19 | 0.33 | 0.23 | 0.32 | 0.25 | 0.05 | 0.29 | 0.21 | 0.29 | 0.34 |
| | | May-Oct | 0.19 | 0.25 | 0.24 | 0.32 | 0.14 | 0.13 | 0.14 | 0.15 | 0.20 | 0.14 | 0.21 | 0.18 | 0.17 |
| Hispar | DESC | Nov-Apr | 0.74 | 0.43 | 0.66 | 0.42 | 0.78 | 0.68 | 0.74 | 0.63 | 0.75 | 0.75 | 0.63 | 0.69 | 0.70 |
| | | May-Oct | 0.51 | 0.42 | 0.44 | 0.35 | 0.50 | 0.59 | 0.54 | 0.56 | 0.50 | 0.55 | 0.57 | 0.55 | 0.56 |
| | ASC | Nov-Apr | 0.59 | 0.27 | 0.44 | 0.25 | 0.65 | 0.42 | 0.60 | 0.41 | 0.36 | 0.61 | 0.42 | 0.51 | 0.53 |
| | | May-Oct | 0.36 | 0.32 | 0.34 | 0.23 | 0.37 | 0.46 | 0.38 | 0.45 | 0.36 | 0.38 | 0.39 | 0.36 | 0.39 |
| Mt Blanc (12 days) | DESC | Nov-Apr | 0.55 | 0.13 | 0.29 | 0.13 | 0.50 | 0.37 | 0.50 | 0.32 | 0.59 | 0.52 | 0.43 | 0.53 | 0.53 |
| | | May-Oct | 0.46 | 0.11 | 0.17 | 0.02 | 0.39 | 0.36 | 0.42 | 0.33 | 0.37 | 0.41 | 0.43 | 0.45 | 0.45 |
| | ASC | Nov-Apr | 0.39 | 0.38 | 0.49 | 0.37 | 0.32 | 0.27 | 0.35 | 0.33 | 0.26 | 0.31 | 0.42 | 0.38 | 0.38 |
| | | May-Oct | 0.40 | 0.05 | 0.05 | 0 | 0.33 | 0.27 | 0.35 | 0.25 | 0.29 | 0.32 | 0.39 | 0.43 | 0.39 |
| Mean | | | 0.46 | 0.29 | 0.37 | 0.27 | 0.43 | 0.39 | 0.43 | 0.38 | 0.33 | 0.42 | 0.42 | 0.44 | 0.45 |


**Figure 7: Dice coefficients obtained when applying different sets of parameters to sets of images for which they were not calibrated. The parameters in the last column correspond to the median parameters calibrated over Mt. Blanc (6 days intervals), Everest and Hispar. The values on the diagonal correspond to the calibrated parameter sets for the given study area, orbit and period.**

The Dice coefficients obtained for the Mt Blanc with a 12-day interval are maximised by the Mt Blanc 6 days parameters, but
with generally lower Dice values than the ones obtained for the Mt Blanc scenes with a 6-day interval (Table 2). The application
of the different parameters sets to the Mt Blanc 12-day scenes results in more false positive detections than false negatives
(Table S2).

**4.2 Characteristics of avalanche deposits**

After manually updating the automated outlines, we detect 1801 (2761) avalanche events in the Mt Blanc, 1192 (2808) in the
Everest and 4323 (3417) in the Hispar regions with the ascending (descending) scenes, corresponding to $3.6 \times 10^{-2}$, $1.0 \times 10^{-2}$
and $3.2 \times 10^{-2}$ avalanches $m^{-2}$ $yr^{-1}$ in the ascending and $5.9 \times 10^{-2}$, $2.0 \times 10^{-2}$ and $4.8 \times 10^{-2}$ avalanches $m^{-2}$ $yr^{-1}$ in the descending
orbits, respectively to the three above-mentioned regions.

Due to the time frequency of images, there appears to be more avalanches detected over Mt Blanc than over the two HMA
domains (Fig. 8a). The size distribution of the avalanches follows a similar distribution for the different regions, at least beyond
the 4000 $m^2$ detectability thresholds (Fig. 8b). These distributions followed an exponential decrease, with slopes between -
$1.1 \times 10^{-5}$ $m^{-2}$ for Hispar and $-2.6 \times 10^{-5}$ $m^{-2}$ for Mt Blanc, with an $R^2$ between 0.44 and 0.89 (Table S3). Some of the largest



events (up to $1.0\times10^6$ m²) are found in the Hispar region, which is also the region with the highest number of detected avalanches relative to the number of image pairs. The distribution of avalanches is tightly related to the hypsometry of the
surveyed areas (which correspond to the buffered glacierized areas minus the shadow and layover masks) although for all three survey domains, and for the Mt Blanc region especially, the peak in avalanche activity is generally slightly lower than the peak in hypsometry (Fig. 8c). The elevation range over which avalanches are actively detected is narrower than the catchments' hypsometry for Everest and Hispar, where proportionally avalanches affect the upper elevations less, which are also the steepest (Fig. 1e), and where there are extensive and relatively flat glacier tongues with no visible avalanche activity. This is
not the case for the Mt Blanc massif where avalanches are the most frequent at lower elevations, relative to the hypsometry.

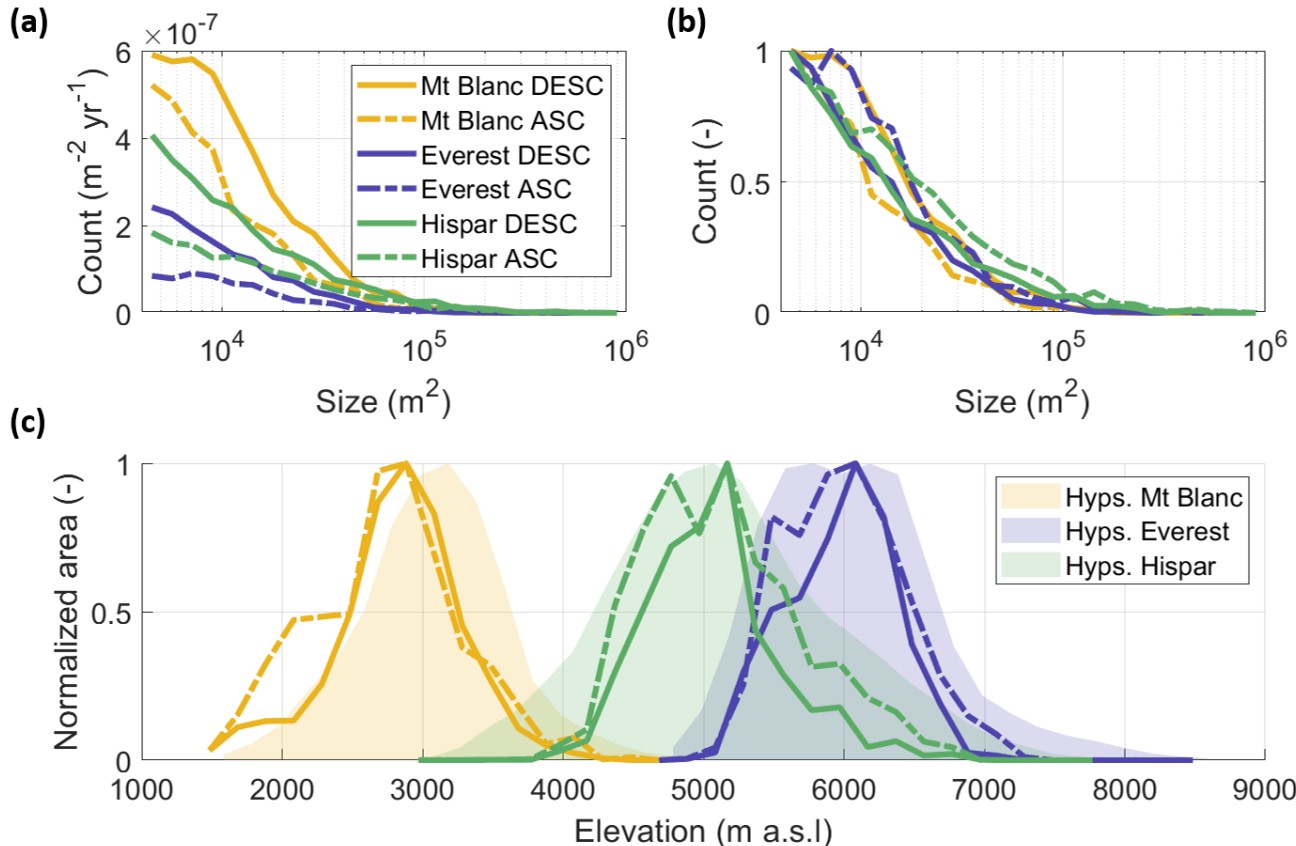

**Figure 8: Size distribution of avalanche events at the three different sites, with (a) and without (b) normalisation. (c) Total size of avalanche events expressed as a fraction of the surveyed area segmented in 200m elevation bins.**

For all that follows, we take the union of all detected avalanche detections and define the resulting connected features as
avalanche deposits. We define the avalanche activity as the sum of the total area (for a glacier) or number (for a deposit) of all avalanche events over a given time period and given area. We also define the proportion of deposits as the total area of deposits for a given location. Avalanche deposits have a maximum activity of 3.8 events per year for the Mt Blanc massif, and up to 4.6 events per year for the Hispar and Everest regions, where Sentinel-1 image pairs are acquired at a 12-day interval (Fig. 9a).



These maxima are likely an underestimation of the actual deposit activity given that deposits with a frequent avalanche activity
remain for long periods of time with high surface roughness and therefore high backscatter values preventing the detection of
further avalanches (Fig. S9c-d). Despite these limitations, distributed deposit activity maps are indicators of where the most
active avalanche deposits are located, which is generally at the base of steep headwalls and in some cases below large hanging
glaciers (Fig. 9b-e).

**Figure 9: (a) Avalanche activity for all avalanche deposits. Examples of avalanche activity maps (number of avalanches over the
five-year study period) at various locations across the three survey domains, on Argentière Glacier (b) and Talèfre Glacier (c) in the
Mt. Blanc, on Khumbu Glacier (d) in the Everest region, and on Mulungutti Glacier (e) in the Hispar region. Contour lines are from
the AW3D30 DEM and are taken every 50 m. Background images from © Google Earth.**

We compared the avalanche activity and proportion of avalanche deposits on the different glaciers of the three survey domains
with the proportion of slopes steeper than 30° in the glaciers' catchments (R index, Hughes, 2008; Laha et al., 2017). We found
that for a given proportion of steep slopes, the maximum avalanche activity and proportion of avalanche deposits per glacier
is generally around one order of magnitude smaller than this R index (Fig. 10, S14). It is also noteworthy that a number of





(generally smaller) glaciers have an avalanche activity and proportion of avalanche deposits smaller than this maximum value, indicating that while a high R index value is a necessary condition for a high avalanche activity, it is not sufficient.

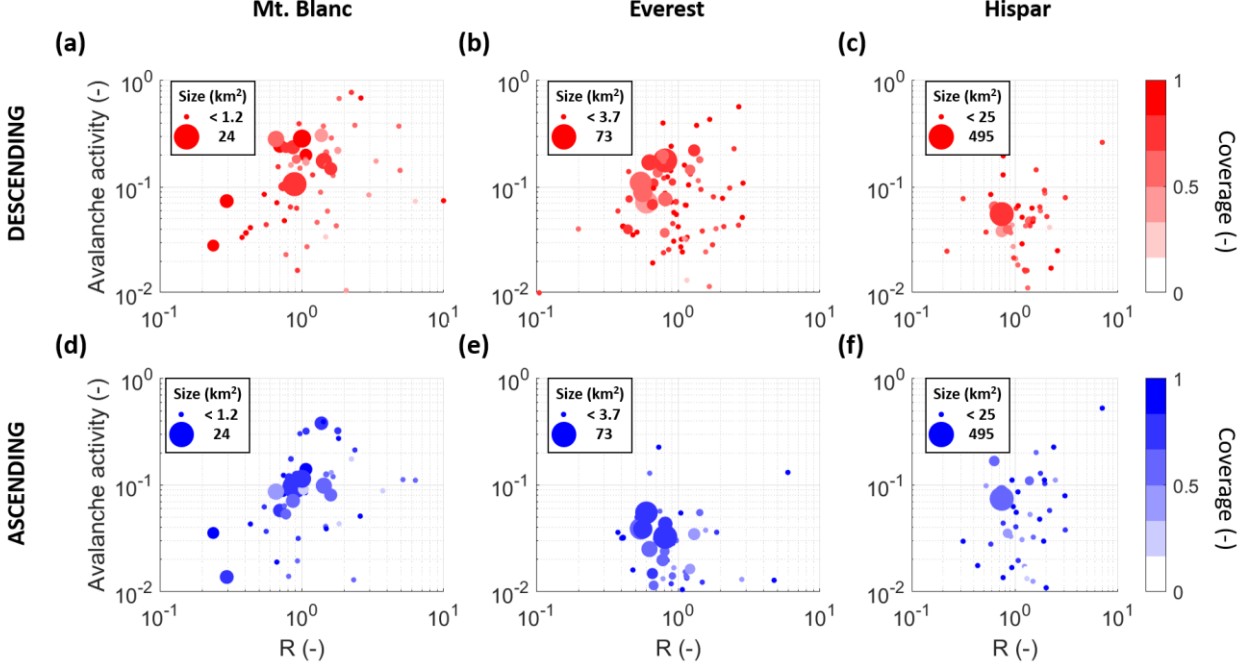


**Figure 10: Avalanche activity per glacier as a function of the proportion of slopes steeper than 30° in the glaciers' catchments (R index, Hughes, 2008). The size of the dots indicates the size of the glaciers and their colour corresponds to the proportion of glacier area that is free of shadow and layover.**

### 4.3 Spatio-temporal evolution of avalanches

The avalanche activity varies seasonally and with elevation. While there are pronounced seasonal differences (Fig. 11-13, S16-S18, Table S4), the interannual variability of deposit activity is not very strong (Fig. S15). Interestingly, only a minority of deposits are active every year, which indicates that the detected yearly avalanche activity at a given location is not very regular (Fig. S15).

At all three sites, the spatio-temporal patterns of number and size of detected avalanche events are similar from year to year (Fig. 11-13, S16-S18). There are avalanches all year round over the Mt Blanc massif, but with a higher activity between January and July (Fig. 11). Between January and April there are well individualised peaks in avalanche activity which correspond to peaks in solid precipitation and are well captured by the avalanche forecast (Fig. S19-S23). From mid-April to July, despite the lower amount of precipitation (Table S5), there are longer periods of avalanche activity with similar number

and size of events as in the colder January-March months, but which are not captured by the avalanche forecast (Fig. S19-S23, Table S4). From mid-November to mid-April, avalanches are mostly identified at elevations lower than 3500 m a.s.l, and as





low as 1500 m a.s.l, which is the lowest elevation reached by glaciers in this survey domain (Fig. 1).This lower limit of avalanche detections rises from 1500 m to 2700 m a.s.l between April and July, and from mid-June the avalanche activity reduces and all events take place between 2700 m and 4300 m a.s.l. The avalanche activity increases again from December

onwards and the elevation of detected avalanches lowers again to 1500 m a.s.l by January. There is therefore a clear seasonality in the avalanche activity, driven by temperature and snow conditions, as indicated by the seasonality of snow and rain events at 3000 m a.s.l (Fig. 11c). However, regarding specific events there is a strong control from precipitation, with peaks in avalanche activity corresponding generally to peaks in precipitation, including during the warmer months of April-July (Fig. S16-S20).

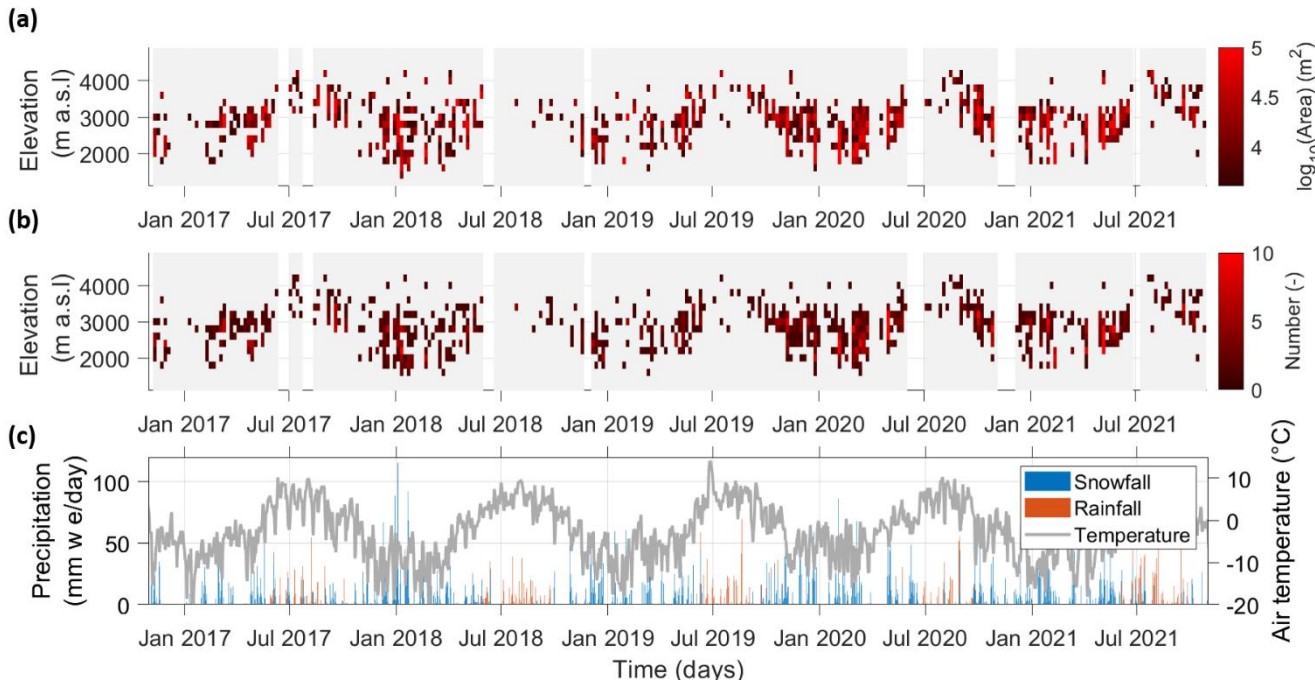

**Figure 11: Five years (11/2016-10/2021) of avalanche time series over the Mt Blanc massif in the descending orbits. (a) Total area and (b) number of avalanches as a function of time and elevation. Frequency of acquisitions is 6 days. White rectangles indicate data gaps. (c) Total precipitation and mean daily air temperature at 3000 m a.s.l over the Mt Blanc massif according to the SAFRAN reanalysis product (Vernay et al., 2022).**

A seasonality is also apparent for the Everest region, with the highest avalanche activity occurring in the monsoon months, between June 21st and September 21st (45-53% of the annual avalanche activity, Table S4) with a ~1 month lag relative to the start of the monsoonal precipitation events (Fig. 12, S17, Table S4-S5), with some high pre-monsoon avalanche events such as at the end of May 2021 seemingly not affecting the avalanche activity. This is also when avalanches are detected at higher elevations, between 5300 m and 7100 m a.s.l. During the periods from October to April avalanches range between 5100 and

6300 m a.s.l and are much less frequent, with periods with no detected avalanches at all.



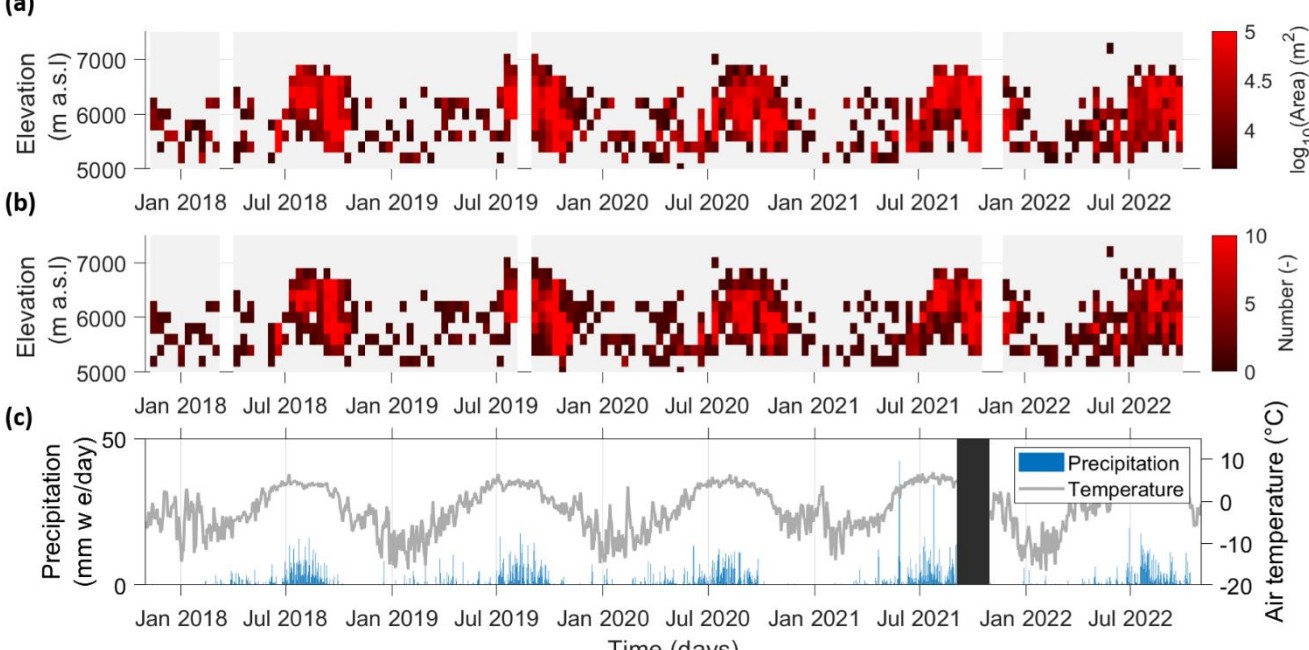

**Figure 12: Five years (11/2017-10/2022) of avalanche time series over the Everest region in the descending orbits. (a) Total area and (b) number of avalanches as a function of time and elevation. Frequency of acquisitions is 12 days. White rectangles indicate data gaps. (c) Daily precipitation and mean air temperature recorded at the Pyramid precipitation gauge (5035 m a.s.l). The black rectangle indicates a data gap.**


There is also a seasonal signal visible for the Hispar domain, mostly linked to temperature and snow conditions as precipitation occurs all year round without a clear seasonality (Fig. 13, Table S5). The avalanche activity is highest between May and October, which is also when air temperatures are higher and avalanches are detected at higher elevations, between 4500 and 6700 m a.s.l. This lower elevation bound does not vary much during the year, however, the upper elevation bound lowers down

to 5300 m a.s.l during the cold period between October and May, even if it is less defined as for the two other survey domains.



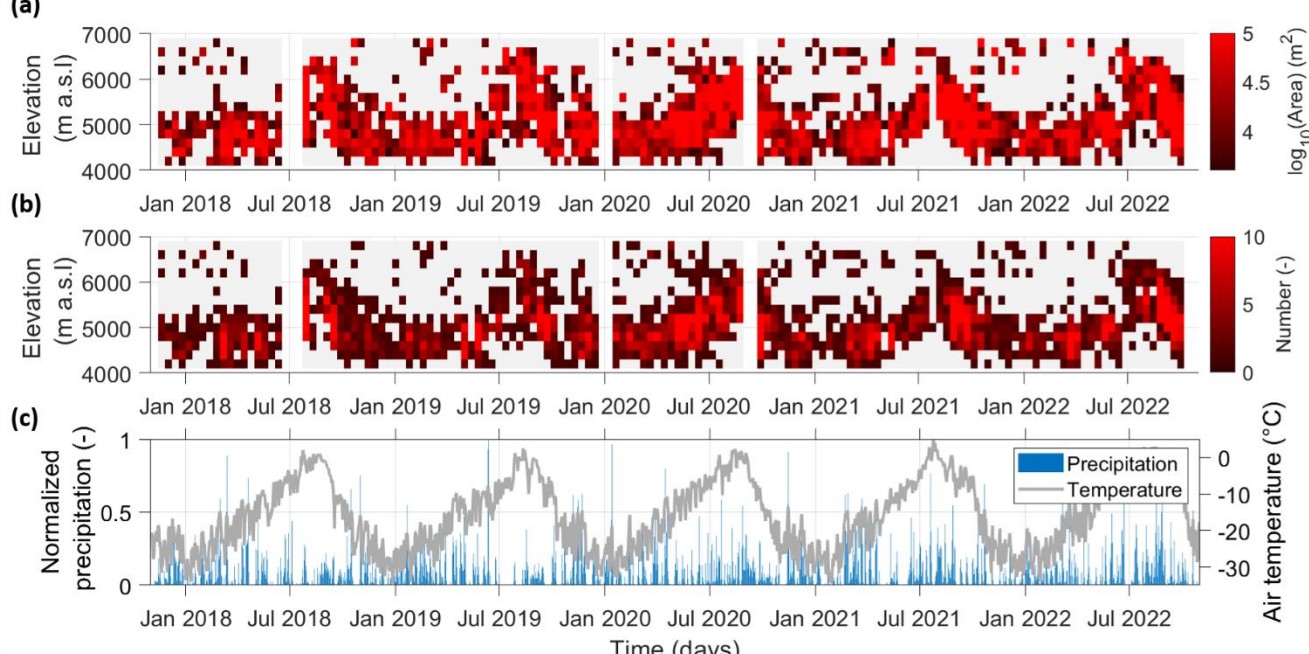

**Figure 13: Five years (11/2017-10/2022) of avalanche time series over the Hispar region in the ascending orbits. (a) Total area and (b) number of avalanches as a function of time and elevation. Frequency of acquisitions is 12 days. White rectangles indicate data gaps. (c) Daily precipitation and mean air temperature over the region from the ERA5-Land reanalysis product (Muñoz Sabater, 2019). Daily precipitation values were normalised due to potential biases (Khadka et al., 2022).**

## 5 Discussion

### 5.1 Suitability of Sentinel-1 for detecting avalanches in remote glacierized regions

We have applied a semi-automated approach to obtain a long-term (five years) time series of avalanche deposits in remote glacierized areas of the European Alps and High Mountain Asia, locations where no data on such events existed.

We used Sentinel-1 images to detect avalanche events, which enabled us to obtain a massif-wide distributed dataset, at least for the zones unaffected by shadow and layover, therefore less spatially biased than ground-based inventories in populated valleys (Eckert et al., 2010; Schweizer et al., 2020). The performance metrics obtained from our automated mapping approach are similar to that of other studies following similar threshold-based approaches (Leinss et al., 2020; Eckerstorfer et al., 2019; Karas et al., 2022; Wesselink et al., 2017). The performance of such approaches is generally very good in dry snow conditions, with high precision (>0.7) and low false positive rates (<0.4), which correspond to Dice values above 0.6-0.7 (Leinss et al., 2020; Eckerstorfer et al., 2019). The few studies that targeted extensive periods rather than a specific event also encountered the most difficulties for periods with wet snow conditions, leading to extensive false positive detections which had to be removed manually (Eckerstorfer et al., 2019). Such false positive detections can be discarded manually based on size and





texture considerations, which indicates that deep learning approaches based on convolutional neural networks, for example, offer a promising way to improve these classifications (Tompkin and Leinss, 2021; Waldeland et al., 2018; Yang et al., 2020; Bianchi et al., 2021; Kapper et al., 2023). Such approaches trained with large enough datasets (Hafner et al., 2021) would likely also improve the transferability of the mapping to other sites with different topo-climatic conditions and frequency of acquisitions. Indeed, scenes unaffected by snow wetness changes (descending/morning acquisitions during the cold season)

are well mapped regardless of the parameter set (Fig. 7). However, the scenes acquired in other conditions require specific calibration and the mapping performs less well.

Our comparison of the Sentinel-1 with the Pléiades avalanche outlines indicate that avalanches detected with Sentinel-1 are of relatively large size (>4000 m$^2$ deposits) with high surface roughness, which limits the detectability to avalanches with high

enough snow temperatures to form granular deposits (Steinkogler et al., 2015), or which are formed from cohesive wind slabs (Fig. 5d) or that entrain rock or ice debris, for instance from serac falls (Fig. 5c). Therefore, cold, low density snow progressively redistributed down steep rock faces or snow gullies (Sommer et al., 2015) is likely to be missed by this method, which likely also explains the upper elevation limits to avalanche detections, especially during the cold season (Fig. 11-13). Similarly, the detection of the avalanche events requires the previous deposits to have regained lower backscatter values for

the signal to be visible, meaning that the surface of the deposit needs to have been smoothed by additional precipitation or melt for the next events to be visible at this location. We have observed this smoothing to require several weeks and even months before avalanches can be detected at the location of old deposits, while avalanche events are still occurring in the meantime (Fig. S9d). The avalanche activity that is detected is therefore a lower bound value of the actual avalanche activity, and the aggregation of all Sentinel-1 deposits is still an underestimation of all the glacierized areas affected by gravitational

snow redistribution (Fig. 5a). Nevertheless, this semi-automated approach is promising to explore the temporal and spatial variability of avalanches in remote areas, especially in glacierized regions of HMA, where close to no data exists on the occurrence of such events (Ballesteros-Cánovas et al., 2018; Caiserman et al., 2022; Acharya et al., 2023; Singh et al., 2022).

For future implementation of SAR detection of avalanches, we therefore recommend prioritising the use of morning scenes.

Although scenes acquired in the afternoon may help fill spatial and temporal gaps, it is important to note that they will require additional work to separate actual avalanche events from false positive detections caused by snow wetness changes, and will likely not considerably change the long-term spatio-temporal patterns of avalanche activity (Fig. 11-13). At this stage, automated outlines require a careful manual check even during the cold season and for the morning scenes, with up to 36% of detected false positive detections and 41% of identified false negative detections for our survey domains (Fig. S10). Similarly,

the parameters used for the automated mapping are likely not directly transferable to other locations. However, using the median of all our parameter sets (Table 2) is likely a good first guess to apply our mapping approach to other survey areas (Fig. 7), either for the calibration of new parameters or to obtain a first reasonable avalanche map which can then easily be updated manually.





### 5.2 Characteristics of on-glacier avalanches

The size distribution of avalanches with Sentinel-1 RGB pairs shows a peak around 4000 m², with less detections for smaller avalanche sizes. Beyond this 4000 m² value the frequency of avalanches decreases with size following a similar exponential decrease for all survey areas (Fig. 8, Table S3). Similar observations have been made for snow avalanches in the European Alps or North America based on field inventories (Faillettaz et al., 2004; Birkeland, 2002; Schweizer et al., 2020). The lower number of small avalanches in these inventories is generally interpreted as an observation bias, with small events being difficult

to detect visually and not consistently inventoried (Schweizer et al., 2020), unless automatically recorded by seismic sensors (Reuter et al., 2022). This is also the case for detections of avalanches (or any other features) using remote sensing products, that are constrained in this case by the spatial resolution of the images (Hafner et al., 2021; Miles et al., 2017; Kneib et al., 2020). The 4000 m² threshold was therefore interpreted as a size detectability threshold below which avalanches are likely to be missed. This value is consistent with other studies that have used Sentinel-1 images for the detection of avalanches

(Eckerstorfer et al., 2019).

    During the periods with the highest avalanche activity in the three survey domains we detected between 2 (Everest) and 8 (Mt Blanc) avalanches/day/100 km², which is relatively low compared to the value of 10-20 avalanches/day/100 km² suggested by Schweizer et al. (2020) for days with a high avalanche level (4) in the Davos region of Switzerland. This difference is likely

due to the detectability threshold, as well as the fact that recurring avalanches are likely to be missed if the surface roughness does not change between two events (Fig. S9c-d). More avalanches are detected in the Mt Blanc massif, which is likely at least partly due to the higher temporal frequency of Sentinel-1 acquisitions over this range. Indeed, manual mapping of avalanches with images with a 12-day interval results in 4 to 62% less avalanche area detected than with images with a 6-day interval (Table S6). As a result, the activity of the deposits in the Mt Blanc massif is also higher than in the two other regions (Fig. 9a).

The activity of the deposits on Everest and Hispar is similar, with the Hispar deposits being generally more active than in the Everest region, which could be due to precipitation events in the westerlies-influenced Karakoram being more distributed throughout the year, while the avalanche activity in the Everest region is low outside of the monsoon (Fig. 12-13). Some deposits appear to be much more active (up to 4.6 avalanches/year, Fig. 9) than what has previously been observed for snow avalanches in the European Alps (<0.6 avalanches/year, Eckert et al., 2013), which could be related to the fact that at higher

elevations the deposits remain active for longer periods of time, if not throughout the year, due to snow accumulation and the presence of hanging glaciers that may break off on a more or less regular basis, irrespective of the season (Pralong and Funk, 2006). Indeed, snow avalanches cannot be distinguished from serac falls in the Sentinel-1 images.

    Avalanches tend to be more concentrated at low elevations for all three survey domains, and we observed a shift between the

hypsometry of the glacierized catchments and the avalanche activity (Fig. 8c). This is likely related to the slope distribution with regards to elevation, as for all survey domains the proportion of slopes higher than 30° increases with elevation, from 0



to close to 100% (Fig. 1). Slope has been identified as one of the main controls for avalanche release (Voellmy, 1955), and numerous algorithms use this as a primary criterion to classify avalanche-prone slopes, along with secondary criteria such as terrain curvature, roughness or the presence of vegetation (Maggioni and Gruber, 2003; Bühler et al., 2018; Duvillier et al.,
2023). Avalanche deposits are thus expected to occur only for slopes lower than 35°, which is generally the criteria taken for the automated detection of avalanche deposits in remote sensing images (Leinss et al., 2020; Eckerstorfer et al., 2019). Avalanche deposits therefore preferentially occur in the lower half of the catchments, thus highlighting the redistribution of snow from higher altitudes. Interestingly however, contrary to the Mt Blanc massif where avalanching events are frequent at the lowest elevations of glaciers (Fig. 8c) and especially in winter (Fig. 11), the large ablation zones of the Hispar and the
Everest regions are less affected in proportion by avalanching (Fig. 9c), which is likely due to the fact that most avalanches occur in the summer months, when the snow-rain transition and snowline elevation is higher (Fig 1, 12-13; Racoviteanu et al., 2019; Girona-Mata et al., 2019).

We could outline a clear seasonality of the avalanche activity at each domain, with contrasting patterns between the three sites
(Fig. 11-13). The avalanche activity is more important in winter and spring in the Mt Blanc massif (21-35% and 32-44% of the avalanche activity, respectively, Table S4), and the avalanche peaks coincide with high precipitation events, following what is typically observed at lower elevations in the European Alps (Baggi and Schweizer, 2009; Schweizer et al., 2020). The number and size of avalanches decreases and their minimum elevation increases in Spring with rising temperatures and their dependence on precipitation is less strong (Fig. 11, S19-23), highlighting the transition from dry to wet avalanches (Baggi and
Schweizer, 2009). This also hints towards a delay of a few months for the redistribution of part of the snow from the mountain headwalls down to the glaciers. Avalanche deposits are still detected in the summer months at high elevation, related either to snow and ice avalanches, but also to rock avalanches from de-glacierized headwalls (Legay et al., 2021). The Everest region, characterised by a monsoon-dominated climate with very little precipitation in winter (Sherpa et al., 2017) reaches its peak avalanche activity between July and September, with avalanches then mostly occurring at high elevations relative to the
hypsometry of the study area (Fig. 8c, 12). There again, there appears to be a 1-2 months delay between the occurrence of precipitation and the avalanche activity, both for at the start and at the end of the monsoon. The avalanche activity is also higher in the summer in the Hispar region (37-51% of the annual avalanche activity), although the seasonality of the precipitation is much less strong than for Everest (27% of the annual precipitation, Table S5). This seasonality in avalanche activity could partly be explained by the presence of cold and dry snow at high elevations in the winter, leading to high
backscatter values that may reduce the detectability of avalanches, and especially slab avalanches (Fig. 5), in these upper reaches.

The three survey domains are characterised by many hanging glaciers located on numerous headwalls of the studied glaciers (Kaushik et al., 2022). We expect these hanging glaciers to sporadically release large avalanches, well visible in the Sentinel-
1 images due to the presence of ice blocks in the deposit area. However, the avalanche activity at the scale of the three survey



domains seemed to be mainly driven by temperature and precipitation, which are unlikely to influence ice detachments from these glaciers (Pralong and Funk, 2006). This indicates that mass redistribution is dominated by snow avalanches. A complementary explanation is that ice detachments from hanging glaciers are more likely to trigger large deposits when they can entrain snow that has accumulated along the avalanche flow path, and they therefore enhance the avalanche signal during periods of already high snow avalanche activity (Fujita et al., 2017).

### 5.3 Implications for glacier mass balance

While the Sentinel-1 images do not give any indication on the volume or mass of the redistributed snow, we obtained from these products key information related to the spatial extents of the avalanche deposits and the spatio-temporal variability of the avalanche activity (Fig. 11-13). Avalanches are indeed important contributors to the mass balance of glaciers, and with no prior knowledge of the location of the main avalanche deposits, this contribution has to date been estimated only indirectly (Laha et al., 2017) and on the basis of topographical characteristics (Hughes, 2008; Brun et al., 2019). Avalanche extents derived from remote sensing images have been used at a handful of locations to calibrate simple mass redistribution routines based on excess snow to be redistributed from pixels where the snow height exceeds a certain threshold that decreases exponentially with the slope (Bernhardt and Schulz, 2010; Ragettli et al., 2015). Avalanche outlines from the Sentinel-1 images therefore provide a much more detailed and consistent dataset to calibrate such parametrizations to adapt them to different topo-climatic settings.

The Sentinel-1 time series also enabled the identification of avalanche 'hotspots', i.e. locations at the surface of the glaciers with a high avalanche activity. At the glacier scale, we could therefore show that the presence of steep slopes within the glacier catchments is a clear necessary condition for avalanches to occur (Fig. 10, S14; Hughes, 2008; Laha et al., 2017), although not a sufficient one to detect these events. Ultimately, at the scale of a glacierized massif we could extract a clear seasonal and altitudinal signal in avalanche activity, controlled mainly by precipitation events, thus indicating that at this scale the mass redistribution after a snowfall can be considered to occur almost instantaneously, with a time lag of 1-2 months at most (Fig. 11-13, S16-S23).

### 6 Conclusion

Our study explored a five-year time series of avalanches across three distinct remote glacierized areas. These regions were expected to be strongly affected by avalanching, yet lacked consistent avalanche observation records. Leveraging the capabilities of repeat Sentinel-1 SAR images, we successfully established a semi-automated framework for identifying avalanche deposits within intervals of 6 to 12 days. Notably, the devised automated method exhibited strong performance, particularly for the morning and cold-season scenes, although certain limitations required manual refinements of parts of the outlines.

The semi-automated mapping of avalanche deposits enabled the characterization of avalanche events in terms of size, frequency and spatio-temporal evolution. We could use this dataset to identify avalanche hotspots at various locations of the survey domains and to link the on-glacier avalanche activity with the proportion of steep slopes in the glaciers' catchments. Our analysis revealed that the exponential decline in size distribution of avalanche deposits was consistent across all three surveyed domains, with the Hispar region displaying a somewhat gentler slope. Importantly, the distribution of avalanches shows a bias towards lower elevations, with however minimal impact on the expansive glacier tongues of the Hispar and Everest regions. This altitudinal distribution varies seasonally, with avalanche deposits expanding at lower elevations during the colder periods. This temporal variability is also strongly controlled by precipitation, with the snow redistribution occurring almost immediately after a snowfall, albeit with some time lags of approximately 1-2 months in the Mt Blanc and Everest regions.

Ultimately, our approach enables the mapping of avalanche deposits over long time periods at the scale of a small mountain range, thus providing crucial information on the timing and spatial distribution of avalanche characteristics, to better account for this mass redistribution in glacier models.

**Code availability**

The Google Earth Engine and Matlab scripts for the processing of the Sentinel-1 images to automatically derive avalanche outlines will be made available on Zenodo and GitHub upon acceptance of the manuscript.

**Data availability**

Avalanche outlines for all three sites and avalanche metrics per glacier will be made available on Zenodo upon acceptance of the manuscript.

**Competing interests**

The authors declare that they have no conflict of interest.

**Acknowledgements**

This project has received funding from the Swiss National Science Foundation (SNSF) under the Postdoc.Mobility programme, grant agreement P500PN_210739, CAIRN, "Contribution of avalanches to glacier mass balance". The Pléiades images used



in this study were obtained through the Kalideos-Alpes project (https://alpes.kalideos.fr) funded by the French Space Agency

(*Centre National d'Etudes Spatiales*, CNES) and the DINAMIS initiative through the research infrastructure DATA TERRA (https://dinamis.data-terra.org/). Authors from IGE acknowledge the support from the LabEx OSUG@2020 (Investissements d'Avenir - ANR10 LABX56). Finally, we would like to thank Anna Karas and Benjamin Reuter at the Centre d'Etude de la Neige, Grenoble, for their useful inputs in the preparation stages of the manuscript.

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
