# Peer review of "Mapping and characteristics of avalanches on mountain glaciers with Sentinel-1"

_EGUsphere, 2023_

## Author Comment (AC1)

**Reviewer 1**

Dear authors,

Thank you for the interesting work combining avalanche detection with glacier mass balance. I agree with you that remote sensing has great potential for avalanche identification that goes beyond what has been done so far. However, I have identified several points that need to be addressed before publication.

We would like to thank Reviewer 1 for his/her high quality review, and his/her very relevant and constructive comments.

Below you find a collection of the most relevant and general comments followed by specific comments referring to lines in your manuscript:

- In your processing chain for the Sentinel-1 data you state that you have averaged the polarization modes VV and VH. Various studies however have observed a difference in backscatter based on the polarization of around 5 dB (e.g., Liu et al., 2022). Why did you still choose to treat the two polarizations the same and did you check if treating polarizations differently would affect your results? (see also comment to 147ff)

  We averaged the VV and VH polarizations in order to reduce the radar speckle, as has been done in previous studies (e.g. Leinss et al., 2020). However, you are correct in the sense that VV and VH have a difference in backscatter and therefore should in theory be treated separately. We had conducted some initial tests to check if it really made a difference for the automated mapping of avalanches, but results were inconclusive. Some studies have indicated that VH polarization is more suited for avalanche detection (Hafner et al., 2021), but this difference is likely most important for low incidence angles (e.g. Fig. 7 of Tompkin and Leinss, 2021), which were regions that were in part masked out when applying our shadow/layover and brightness masks (Section 3.1). When calibrating our method separately to the VV and VH RGB images for the Mt Blanc massif for the period 2019-2020, we find generally lower values than those obtained with the averaged polarizations (Table R1). This indicates that our approach is better suited to the combined VV and VH polarizations. We will now mention this interesting point in the discussion section and add Table R1 as Table S7 to the SI:

  *'Future method developments could also benefit from separating the VV and VH polarizations, particularly for regions of the SAR images with low incidence angles (Tompkin and Leinss, 2021). While in our case we obtained better results by averaging the two (Table S7), other machine learning-based approaches would likely benefit from the additional information provided by the two polarizations (Liu et al., 2022).'*

  *Table R1: F1-scores obtained for the calibration of our method to VV and VH RGB triplets for the period 2019-2020 over the Mt. Blanc massif.*

| *Polarisation* | *Path* | *Season* | *F1-score calibration* |
|---|---|---|---|
| | | | |

| | | | |
|---|---|---|---|
| *VV* | *Descending* | *November-April* | *0.29* |
| | | *May-October* | *0.40* |
| | *Ascending* | *November-April* | *0.47* |
| | | *May-October* | *0.31* |
| *VH* | *Descending* | *November-April* | *0.17* |
| | | *May-October* | *0.18* |
| | *Ascending* | *November-April* | *0.14* |
| | | *May-October* | *0.30* |
| *(VV+VH)/2* | *Descending* | *November-April* | *0.56* |
| | | *May-October* | *0.56* |
| | *Ascending* | *November-April* | *0.54* |
| | | *May-October* | *0.49* |

- The (description of the) process of manual avalanche mapping for parameter estimation followed by automatic detection but then again extensive manual corrections is confusing. How good was your automatic detection, why did you need to manually correct in the first place? How much time is gained by including the automatic mapping compared to going fully manual? Additionally, it is not always clear from which processing step the results presented in the result section came from.

We apologise for the lack of clarity in the overall description of our approach. It is important to note that developing a mapping approach for avalanches was only one of the focus points of this study, and that our main objective was the study of the spatio-temporal variability of these events for the three studied regions, as mentioned in the introduction (L96-99):

*'To this end we (1) calibrate and evaluate our automated mapping approach at each site and assess its transferability to other sites, (2) extract the size-frequency characteristics of avalanches at various spatial scales over a period of five years and (3) evaluate the implications for the glacier mass balance.'*

While the fully automated approach did lead to relatively high scores, as described in sections 4.1.2 (Table 2) and 4.1.3, there were still a number of false positive and false negative detections, which we feared may influence the final results related to the spatio-temporal variability. That is why the manual refinement was applied. Considering that 36% of the avalanche area was removed and 41% was added in this second manual step which mostly affected the descending and May-October scenes, but also that the mapping is considerably simplified when the regions of interests are already partly mapped, we consider that applying the automated approach first still reduced the mapping time by at least half (more for morning and winter scenes, which required less manual updates).

We will make sure to clarify this point in the discussion section:

*'Our automated mapping therefore still requires manual edits, although we consider that applying the automated mapping approach and then updating the outlines by hand has reduced the mapping time by at least half relative to a fully manual mapping, more if only morning scenes were to be considered.'*

We will also make sure to explicitly state what set of outlines is used in the different results sections:

*'Here, we first compare our manually derived outlines with high-resolution Pléiades images and evaluate the results and transferability of the automated mapping approach (Section 4.1). We then use the manually updated set of outlines to obtain the characteristics of avalanche deposits (Section 4.2) and their spatio-temporal variability (Section 4.3) for all three survey domains.'*

Furthermore, in the methods we will put the paragraph on the application of the mapping to the entire Sentinel-1 time series after the paragraph on the comparison with Pléiades, and explicitly state the reason for the manual editing:

*'We required highly accurate maps of avalanche deposits for the analysis of their spatio-temporal characteristics of avalanche deposits.'*

- There is a lack of clarity in your discussion regarding the detectability of dry snow avalanches (line 440ff, 456ff). You state that "the performance of such approaches is generally very good under dry snow conditions […] most difficulties for periods with wet snow conditions", this contradicts (other) previous research (e.g., Eckerstorfer et al., 2022, Abermann et al., 2019). But you also contradict yourself by writing later that "cold, low density snow […] is likely to be missed by this method, which likely also explains the upper elevation limits to avalanche detections, especially during the cold season." Please carefully reexamine the passages and literature concerning this topic and adapt your discussion.

  These apparently contradictory statements refer to two different things:

  - When avalanches are manually detected in Sentinel-1 scenes in dry snow conditions, they are usually also well mapped by the automated approach, as indicated by the high F1-scores, and there are less false positive detections than in wet snow conditions.
  - However, when we make the comparison with Pléiades optical images (Fig. 5d), we can also note that cold snow avalanches are not always detectable in

the Sentinel-1 images, as they do not always change the roughness of the snowpack much, as mentioned in the discussion section 5.1 (L453-456):

*'Our comparison of the Sentinel-1 with the Pléiades avalanche deposit outlines indicate that avalanches detected with Sentinel-1 are of relatively large size (>4000 $m^2$ deposits) with high surface roughness, which limits the detectability to avalanches with high enough snow temperatures to form granular deposits (Steinkogler et al., 2015), or which are formed from cohesive wind slabs (Fig. 5d) or that entrain rock or ice debris, for instance from serac falls (Fig. 5c)'*

As such, these observations are in line with Eckerstorfer et al., (2019; 2022) and Abermann et al., (2019).

However, we do understand that this difference may not be obvious in the current version of the manuscript and will therefore move the paragraph concerning the comparison with the Pléiades images at the beginning of this discussion section, and highlight the fact that the discussion of the second paragraph only applies to the comparison of the automated versus manually delineated outlines in the Sentinel-1 images:

*'The performance metrics obtained from our automated mapping approach compared to the manual detections in the Sentinel-1 outlines are similar to that of other studies following similar threshold-based approaches'*

- Concerning the structure of your manuscript: There are 21 sections, subsections and subsubsections which make it hard for the reader to orient. Please try to simplify and keep an eye on descriptive headings for your sections.

  Such a structure and number of sections is consistent with other articles published in The Cryosphere. We felt that it was better to have more subsections with explicit titles rather than larger sections with less obvious structure. We do however agree that some of the headings could be more explicit. We will make the following changes:

  - '3.3 Parameter calibration' instead of 'Threshold calibration'
  - '4.1 Sentinel-1 avalanche mapping' instead of 'avalanche mapping'
  - '4.1.1 Comparison of Sentinel-1 and Pléiades manual detections' instead of 'Sentinel-1 avalanche mapping potential.'

- You have included many figures, please keep only those illustrating your main points/results (see also specific comments).

  In the specific comments reviewer 1 suggests removing Fig. 9a, Fig. 10, 12 and 13 from sections 4.2 and 4.3 of the results related to the characteristics and spatio-temporal variability of detected avalanche deposits, and questions the readability/relevance of some parts of figures 1, 4 and 5 which are more related to the evaluation of the mapping approach. For the latter, we do agree that some elements may be confusing and may impact the overall readability of the overall figure. As per our answers to the specific comments, we will therefore:

  - Move panels e and d of figure 1 to the SI

○ Move figure 4 to the SI
○ Move panel a of figure 5 to the SI

Regarding the removal from the main text of Fig. 9a, Fig. 10, 12 and 13, which describe the characteristics and spatio-temporal variability of detected avalanche deposits, we note that while a portion of our manuscript is meant to focus on the methodological aspects of avalanche mapping with Sentinel-1 images, the other main objective was the study of the spatio-temporal variability of these events for the three studied regions, as stated in the introduction (L96-99):

*'To this end we (1) calibrate and evaluate our automated mapping approach at each site and assess its transferability to other sites, (2) extract the size-frequency characteristics of avalanches at various spatial scales over a period of five years and (3) evaluate the implications for the glacier mass balance.'*

We feel that these figures are important elements highlighting these characteristics and spatio-temporal variability and therefore do illustrate our main points/results. However, we also understand the reviewer's concern related to the overall high number of figures in the manuscript and to the readability of figures 9-13. We will therefore:

○ Improve the readability of Fig. 9-13 following the reviewer's specific comments
○ Move Fig. 10 to the SI along with Fig. 4, which will bring the number of figures down from 13 to 11

- There is a lot of spoken instead of written language used, for example "this is even more true", "indeed", "ultimately". Please stick to scientific written language, especially for starting and connecting your sentences.

  We will replace 'This is even more true' with 'This is particularly the case' and remove or replace the adverbs 'ultimately' and 'indeed'.

- Lastly, you have a mix of methodology, discussion and results in those three sections. Could you please go over this again and try to disentangle. If you want to keep discussion with results you could also do a section "Results and Discussion".

  We will make sure to clearly separate results from discussions, following specific comments from both reviewers. Specifically, we will:

  - Re-organize and streamline section 4.1.1 to stick to the comparison between Pléiades and Sentinel-1 outlines
  - Remove sentences repeated from the methods in sections 4.2 and 4.3

**Detailed comments with reference to line numbers:**

39: You make it sound like rock(fall) contributes to mass balance. Debris cover may indirectly contribute by preventing surface melt, but since you want to help calibrate glacier mass models it is weird to mention rockfall with snow/ ice avalanches as if it's the same process. I assume they are not calibrated the same way in the models. Not being able to

differentiate between those mass movements in my opinion is a limitation of the chosen method (which should be mentioned in the discussion/limitations).

If we define the mass balance of a glacier as everything that comes in minus everything that comes out of a glacier, then we would argue that rockfalls do contribute to the glacier mass balance. It is true however that this is something very distinct from snow/ice avalanches (although you could imagine that these snow/ice avalanches may also transport with them some amount of rocks). We will therefore keep this term here but will add more details about the limitations that it poses in the discussion:

*'It is also noteworthy that this mapping approach with Sentinel-1 will likely not differentiate large rockfalls on glaciers from snow avalanches, which could explain some of the activity in the summer and autumn in the Mt Blanc massif. '*

70ff: The citations on previous avalanche work in the introduction do not account for most recent work while you mention most of the recent work in the discussion/436ff. Could you please try to paint a more complete picture of recent work in the introduction and not bring "new" work only in the discussion without mentioning it before.

We will make sure that the new work also appears in the introduction. Specifically we will:

- refer to the work by Sartori and Darbiri (2023) and Guiot et al. (2023), as suggested by Reviewer 2.
- refer to recent work on the use of machine learning approaches for the automated mapping of avalanches:

  *'More recently a number of studies have also trained machine learning approaches to improve the mapping of avalanches (Tompkin and Leinss, 2021; Waldeland et al., 2018; Yang et al., 2020; Bianchi et al., 2021; Kapper et al., 2023; Liu et al., 2021), but they have been limited by the lack of large training datasets for this application.'*

- directly refer to the work by Hafner et al. (2022, 2023) for the use of optical images to detect avalanches.

84: Could you please quickly explain how validity was proven and what challenges were found?

We will reformulate this sentence as follows:

*'The validity of these approaches has been demonstrated by quantifying the overlap between outlines from Sentinel-1 images and those obtained from high-resolution optical and field observations (Leinss et al., 2020; Hafner et al., 2021).'*

For the limitations, we refer to the end of the paragraph, where we wrote:

*'There remain limitations to these approaches, especially as they fail to detect smaller events (<4000 $m^2$) or have a high rate of false detections in the case of transitions from wet to dry snow that also result in increasing the SAR backscatter (Eckerstorfer et al., 2019).'*

101: To me it is unclear how large your three study areas are. Please specify.

The size of the study areas is indicated for both ascending and descending scenes in Figure 1 (see also response to comment below). We will make sure to specify this in the caption (see response to next comment).

Figure 1: Connected to 101: are the numbers in the upper right corner your added up area of interest? Or is it the percentage of the added up?

The percentage given corresponds to the portion of glacier area covered either by ascending or by descending scenes. We selected the Sentinel-1 images so that the ascending and descending scenes always covered the entire survey area. We will make sure to specify this in the caption:

*'The numbers in the upper right corner indicate the total area of interest covered by the ascending and descending scenes, respectively, and the third number indicates the percentage of glacierized area covered by ascending or descending scenes.'*

Please add an overview map so that every reader can get an idea where your areas of interest are located. Additionally, you need to explain RGI 6.0 at first mention and as you cannot assume that every reader knows what it is.

Agreed, we will add an overview map. And we will specify here what the RGI 6.0 stands for and cite the source:

*'Randolph Glacier Inventory (RGI) 6.0 outlines (RGI Consortium, 2017) are shown in black'*

(d) and (e): these two graphs are quite hard to read, and I am not sure the content is essential for understanding your work. Maybe replace them with the overview map.

We will move these panels to the SI and put the overview map instead.

128: It would be interesting to know if all Pleiades imagery was taken on a day Sentinel-1 was acquired also, or (and if how long) the time gap was?

Agreed, we will specify this here:

*'The winter and August Pléiades scenes were acquired on the same day as a Sentinel-1 acquisition, while the July scene was acquired two days before the nearest Sentinel-1 acquisition.'*

147/172: Here you mention a 500m AND a 450m high pass filter? Did you filter twice? Or if these sentences describe the same operation why is the kernel size different? Additionally, a 450/500m kernel is quite large- did you perform an investigation of the effects of different filter kernel sizes on visibility of avalanches and results?

Apologies for this misleading part. These are indeed two different filters, but the 45 px one is actually a low pass filter (not a high pass, as we wrongly wrote). We will make sure to correct 'high pass' to 'low pass' there.

We did check the influence of kernel sizes on the visibility of avalanches. We actually needed our kernel to be at least of the same order of magnitude as the largest avalanche

events (up to $10^6$ m$^2$), to be able to smooth out all avalanche deposits, even the largest ones. This will be specified in the text:

*'We selected this kernel size to be able to smooth even the largest avalanche deposits'*

This is consistent with the overall objective of this additional filtering step:

*'The idea of this additional step was that an avalanche event results in a spatial discontinuity in the backscatter, if not with the image before, at least in the current image.'*

147ff: You have averaged the polarization modes VV and VH. Various studies however have observed a difference in backscatter based on the polarization of around 5 dB (e.g., Liu et al., 2022). Furthermore, the different polarization modes do not provide information about the same objects (e.g., HV results in a higher noise floor, see also Howell et al., 2019) and a combination of this dual-polarization data should hence be treated as an index rather than a 'normal' dB scale image as suggested in Nagler et al. (2021). Why did you treat the two polarizations the same and did you check if treating polarizations differently affects your (manual and automatic) avalanche detections.

Please refer to our response to the corresponding general comment.

154ff: How much of your area of interest was masked out when excluding shadow and layover? Could you please specify how much of your area was at the end classified glacierized and considered for analyses (and consequently is the area you are referring to in the rest of your manuscript).

We will specify it here:

*'As a result, 35%, 28% and 43% of the considered area was masked out for the Everest, Mt Blanc and Hispar regions, resulting in a total area available for mapping of 492, 140 and 762 km$^2$, respectively (Fig. 1).'*

156: You cannot assume the reader to know what RGI 6.0 is, please explain this abbreviation.

We will specify here what this abbreviation stands for.

Figure 3(e): Given recent work comparing manually mapped outlines (Hafner et al., 2023), I am wondering if the blue outline is "more correct" and how dependent it is on the operator (see also comment to 224/186).

This is a valid question and we acknowledge that manual outlines will always hold some bias, no matter how expert the operator is. This is the reason why we compared the outlines from multiple independent operators to make a first estimate of the underlying uncertainties (Fig. S1-S2, Sections 3.3, 4.1.1). However, in the absence of ground truth, as is the case for these three regions, the manual outlines had to be used as a reference for the automated mapping approach. We do agree that this point deserves more attention and we will:

- mention in the results the F1-score of the different operators:

*'The outlines of the operators had F1-scores between 0.54 and 0.66 relative to the outlines of the operator who derived the entire manual dataset for all three sites. We also directly compared the manual outlines from this operator with the consensus outlines from the other three operators, which were the outlines for which at least two operators agreed (Kneib et al., 2021).'*

- add some explanations in the discussion and refer to the work by Hafner et al. (2023) on this topic:

*'The performance metrics obtained from our automated mapping approach compared to the manual detections in the Sentinel-1 outlines, have a wide range of values (F1-score between 0.29 and 0.78) depending on the season and acquisition time. For most scenes, the F1-score was actually similar to those obtained by manual outlines from independent operators (Table S1, Hafner et al., 2023).'*

Figure 4: What exactly do you want to show with this?

We found this plot interesting as it showed the sensitivity of the automated mapping to the threshold values used. We however agree that it is not a major figure of this manuscript and will move it to the SI.

224/186: Could you please give a number for the agreement between the experts (e.g., Intersection over Union, IoU) to understand how much uncertainty is introduced if one operator manually corrects predictions (see also comment to Figure 3).

We agree that this is a crucial point, and the results of this comparison are given in the Results section L277-285. We will however specify the F1-score (rather than IoU, to be in line with the metric used in the rest of the study), which was calculated in Table S1, there as well:

*'The outlines of the operators had F1-scores between 0.54 and 0.66 relative to the outlines of the operator who derived the entire manual dataset for all three sites. We also directly compared the manual outlines from this operator with the consensus outlines from the other three operators, which were the outlines for which at least two operators agreed (Kneib et al., 2021).'*

234: How did you determine a match between Pleiades and Sentinel-1? How much overlap was needed?

We are unsure whether you refer to a spatial or temporal overlap, so we answer for both:

**Spatial overlap**

For this comparison and all the calibration and validation of the methods, we directly made a pixel-by-pixel comparison, which meant that we did not need to consider the overlap of individual avalanche events. We will specify it here:

*'We compared on a pixel-by-pixel basis the Sentinel-1 outlines that occurred over given periods in the summer and in the winter with manually derived outlines of avalanche*

*deposits from high resolution (0.5 m) Pléiades orthoimages over part of the Mt Blanc survey area'*

And in Section 3.3:

*'We used the F1-score, also known as the Dice coefficient, as a metric to quantify the goodness-of-fit of the automated delineation on a pixel-by-pixel basis (Dice, 1945)'*

**Temporal overlap:**

To be able to compare deposits from a Sentinel-1 RGB pair with deposits from a Pléiades image, the Pléiades image needs to have been acquired as close as possible to the second Sentinel-1 image of the pair. This was the case for most Pléiades acquisitions, and we will specify this in the data section 2:

*'The winter and August Pléiades scenes were acquired on the same day as a Sentinel-1 acquisition, while the July scene was acquired two days before the nearest Sentinel-1 acquisition.'*

There is of course a risk that the deposits detected in the Pléiades image are anterior to the first image of the Sentinel-1 pair so for all the examples that we presented for this qualitative comparison (Fig. 5) we also checked for older Sentinel-1 deposits. We will highlight that these were just qualitative examples in section 4.1.1:

*'The qualitative comparison of the manually derived Sentinel-1 deposits with the Pléiades deposits detected over time periods of ~1 month in the winter and summer seasons gives more insights on the potential of Sentinel-1 images to identify particular deposits (Fig. 5). '*

4.1.1: This chapter would fit better into the discussion, except for some specific results.

Agreed, we will re-organize and streamline this section to stick to the direct comparison of Pléiades and Sentinel-1 outlines:

*'The comparison of the manually derived Sentinel-1 deposits with the Pléiades deposits detected over time periods of ~1 month in the winter and summer seasons gives more insights on the potential of Sentinel-1 images to identify particular deposits (Fig. 5). It indicates locations of very good agreement, usually for large deposits with a lot of surface texture (Fig. 5b). But there are also false positive detections, for example caused by the opening of crevasses (Fig. 5c), as well as false negatives (Fig. 5d), that could reach large sizes (up to 60000 $m^2$, Fig. 5e). The comparison of the aggregation of one year of Sentinel-1 manual outlines with all the deposits identifiable in the end-of-summer Pléiades scene above 2700 m a.s.l results in a F1-score value of 0.47, with a majority of false negatives (Fig. 5). A large amount of undetected Pléiades deposits are actually smaller than the Sentinel-1 detectability threshold of 4000 $m^2$. Nevertheless, excluding  them does not change the comparison (F1-score value of 0.49) between the Pléiades and aggregated Sentinel-1 deposits.'*

264: As mentioned in the general comments, discussion is mixed into the results, for example here as you try to give reasons for results.

As mentioned above, we will remove here any redundancy with the discussion section 5.1.

Figure 5: Even though it is known that avalanche deposits, especially large ones remain visible for a long time, it seems weird to me to compare avalanches from 1.11.2019 Sentinel-1 imagery to 9.8.2020 Pleiades. I would be very careful with this comparison as I believe a comparison of what was visible around the same time is a lot more plausible. If you compare "everything with everything" it becomes hard for the reader to follow. Hence, could you go over your analyses again and leave those not very relevant out.

In this figure, we compared the aggregation of avalanche deposits over one full year. For Sentinel-1 this was easily conducted by taking all outlines since 01/11/19 and until 09/08/20. For Pléiades, it appeared to us that in an end-of-melt season image (09/08/2020), avalanche deposits that had not already completely melted were well visible (rougher, darker surface) and likely corresponded to the accumulation of all avalanche deposits of the previous year, thus the comparison with the aggregated Sentinel-1 outlines. The comparison between the 2 (Fig. 5a) actually shows a relatively good agreement, which comforts our hypothesis and gives an interesting perspective on the mapping of on-glacier avalanche deposits. We however also agree that it doesn't fit so well with the other results shown in figure 5, so we will:

- move panel 5a to the SI.
- shift the first paragraph of 4.1.1 at the end of the second paragraph to give it less importance

276: As mentioned in the general comments, methodology is mixed into the results, like here where you mention (again) how you compared operators. The overall IoU (comment to 224) would be very nice here in addition to the consensus numbers. Additionally, how do you define an avalanche event and how do you separate it? Hafner at al. (2021) found that avalanches in Sentinel-1 might be detected in more than one blob, so just taking connected pixels might be problematic/ lead to one avalanche being counted twice or more times.

We will indicate here the F1-scores for the comparison between operators. As mentioned above, for this entire study the comparisons of outlines are made on a pixel-basis as we are not so much interested in the events, but rather in the spatial and temporal variability of avalanches. We will make sure to clarify this in the methods:

*'A single operator performed the manual detection, and to account for biases in the delineations, we compared on a pixel-by-pixel basis these outlines with those of four other operators for 4 scenes (2 ascending and 2 descending) covering the Mt Blanc region and 4 scenes covering the Everest region (Kneib et al., 2021; Table S1, Fig. S1-S2). '*

We will remove from these results the repetition of the description of the methods and indicate the F1-scores of the different operators:

*'The manual outlines of the three external operators had F1-scores between 0.54 and 0.66 relative to the outlines of the operator who derived the entire manual dataset for all three sites (Table S1, Fig. S1-S2). '*

298: The references to S9b &Co are of different style than in other places and seem to be a mixture between page number and Figure caption. There is quite a few of those throughout your manuscript, please check them all and correct them (e.g., 388, 391, 474).

These are references to figures in the Supplementary Information. In this case, panel b of supplementary Figure 9. They follow the style recommendations of the journal The Cryosphere.

312: What do you mean by referencing to S11-13? Is this supposed to be page numbers? Maybe it should be refereeing to a chapter (then it should be the chapter numbering)? There is quite a few of those throughout your manuscript, please check them all and correct them (e.g., 377, 388, 391).

As explained above, these are references to figures in the Supplementary Information.

302: Removing 36% and adding 41% is quite a lot. I already mentioned this in the general comments, could you please add a section to the discussion where you discuss the benefits of your approach despite it not being transferable and needing quite a bit of manual work.

Please refer to our answer to the general comments.

312/Figure 6: It is unclear for which variables correlation was calculated exactly. Please make this clear.

We will specify this in the figure caption:

*'The Pearson's correlation coefficients characterizing the correlation between the validation set and the outlines from the automated mapping approach are indicated in blue (ascending) and red (descending).'*

4.1.3: Did you use the automated mapping prior to manual corrections for this analysis? Or after the step described in 302?

This section still refers to outlines from the automated mapping approach. We will specify this at the start of the results section (see response to comments above) and in the caption of Figure 7:

*'F1-score obtained when applying different sets of parameters to sets of images for which they were not calibrated, without any manual modifications.'*

Figure 7: What is N-A and M-O for the Ascending and Descending?

This stands for November-April and for May-October. We will specify this in the caption:

*'N-A and M-O stand for the November-April and May-October periods, respectively'*

347: What is R2?

This stands for coefficient of determination. We will spell it here.

Figure 8: It is not clear at first sight that the legend in (a) is true for all panels. Please move the legend outside the panels as it applies to all.

We will change 'Total size of avalanche events' to 'Normalized area of all avalanche events' for the legend of panel c for clarity. But the legend of panel a only applies to panels a and b, which we find to be self-explanatory as it is. We will therefore keep it this way:

*'Figure 8: Size distribution of avalanche events at the three different sites, with (a) and without (b) normalisation. (c) Normalized area of all avalanche events expressed as a function of the surveyed area segmented in 200m elevation bins. '*

359ff: Here you are mixing methodology and results again, please disentangle.

We will remove the first three sentences that are repeated from the methods.

362: What is meant by activity is not clear, I assume it is repeated occurrence of an avalanche in the same place. Since it is not possible to detect whole outlines in Sentinel-1, how did you determine deposits to be "the same", in other words how much overlap did you require?

Following the comment above we will remove this sentence that is repeated from the Methods. The definition is given in Section 3.6 of the Methods, which we will modify for clarity:

*'The union of all avalanche pixels over time indicates individual deposits affected by more or less avalanche activity. We estimated the influence of avalanches on a given glacier, independently for ascending and descending orbits, with two metrics: area affected by avalanches and avalanche activity. The area affected by avalanches is estimated by taking the union of all individual avalanche deposits, and expressed relative to glacier area. The avalanche activity is calculated for each pixel as the number of deposits affecting this pixel over a given time period. It is then calculated on a per-deposit basis by taking the maximum activity and on a per-glacier or per-elevation band basis by taking the area of the glacier affected by avalanches divided by glacier area or area of elevation band, respectively.'*

365: You are using Change detection with a D and D-i image, with change appearing green in your data. The green hue vanishes when moving forward in time even if the deposit remains well visible/ the backscatter of a single image high. Did you analyze backscatter separately or how did you come to that conclusion?

This is a very nice way of summing up our approach. What we meant here was that when there is a new avalanche on an old deposit, it doesn't necessarily lead to an increase in surface roughness (and therefore backscatter), which means that the mapping approach does not work to detect this new avalanche. This is shown by examples in Fig. S9c-d.

Figure 9: I do not see a benefit in (a), the graph is hard to read. I believe a simple table could be a better choice for bringing your point across. Furthermore, the choice of color makes the differentiation between ASC and DESC hard (it is also not color blind safe). Additionally, areas detected in both ASC and DESC cannot be identified. To get the full picture, it would

also be nice to be able to see the area that was masked as outside the glacierized or in radar shadow/layover.

For a more general answer on the value of Figs. 9a, 10-13, we refer the reviewer to our answer to the general comment on this matter. More precisely for this panel (a), we feel that it is particularly important to summarise the avalanche activity of the different individual deposits in each survey area in terms of number of events per unit area, and such information would be difficult to fit in a table. This panel also fits well with the other, more qualitative panels, that indicate the number of repeated avalanches on different avalanche deposits. We will however put a log scale for the y axis to make it more readable. We will also indicate the count in number of deposits per square kilometres to make it more comprehensive. To improve readability of the maps, we will show the descending deposits in blue, indicate in the caption that the ascending and descending deposits overlap (this will ensure that it is colour blind safe) and indicate in the figure the shadow/layover and glacier mask.

Figure 10: This figure is hard to read. Overlapping circles cannot be distinguished and the absolute values of circles of the same size in (a), (b) und (c) differ, conveying that values are the same, though they are not. I would remove that Figure.

For a more general answer on the value of Figs. 9a, 10-13, we refer the reviewer to our answer to the general comment on this matter. We still find this figure interesting as it specifically targets the quite loose definition of what is an 'avalanche-fed' glacier. However, given the reviewer's concern about the high number of figures in our manuscript, we will shift it to the SI. To improve the readability we will put the circle edge colour in black so that overlapping circles are visible. We will also scale the circles to the same size using a log scale to make them comparable between regions.

4.3: Please carefully reexamine for mixture of results with discussion.

We will remove any explanation of the linkages between avalanche activity and meteorological variables for the Mt Blanc massif.

Figure 11/12/13: I would suggest displaying only one of those in the manuscript and moving the other two to the Appendix. Furthermore, you should not use the same color range for number of avalanches and for avalanche area. Additionally, the number of avalanches is discrete (I assume you are displaying per acquisition day), while the way you display it implies a continuous scale. Lastly, the x-axis is the same for all panels, I think they would be better readable/comparable if the time would only be displayed once at the bottom of all three panels. In (c) you could improve readability by adding a thin grey line at 0° Celsius. Additionally, you should indicate data gaps in Fig 12 the same for all panels and not once white and once black.

For a more general answer on the value of Figs. 9a, 10-13, we refer the reviewer to our answer to the general comment on this matter. For Fig. 11-13, we believe that the comparison of the avalanche activity at the three sites is one of the main results of this paper and will therefore keep all three figures in the main text, to allow visual comparison between regions. We will however improve the readability by:

- using a different colour for area and number
- using a discrete colour scale for the number of avalanches and specify in the caption that these numbers are 'for each Sentinel-1 pair'
- using one x-axis for all three panels
- adding a thin grey line at 0°C
- indicating the data gap in T & P data in white
- adding the region title at the top of the figure

440ff: "the performance is generally very good in dry snow conditions"- this contradicts findings by Eckerstorfer et al. (2022) who found a low Probability of detection for solely dry snow avalanches and you also contradict yourself in 456ff where you state that "cold, low density snow avalanches are likely to be missed". I suppose cold low density snow avalanches make up a good proportion of avalanches occurring under dry snow conditions. You also state that a rough surface is detected by its backscatter, generally wet snow avalanches tend to have a rougher surface. Could it be that the changes in overall snow wetness, wet to dry or dry to wet from D-I to D are one of the main drivers affecting detectability. For example, in Figure 11 it seems that the avalanche activity was very low for rain on snow (after a period of low temperatures) events which are generally known for critical avalanche situations and remained low until temperature conditions were stable again over a period.

Please refer to our response to the general comment for the first part of the comment.

You are absolutely correct that wet snow conditions lead to false positive detections in the case of a dry to wet snow transition. We will explicitly state this in the discussion:

*'The few studies that targeted extensive periods rather than a specific event also encountered the most difficulties for periods with wet snow conditions, leading to extensive false positive detections which had to be removed manually in situations of dry to wet snow transitions (Eckerstorfer et al., 2019)'*

447: "such approaches", please be specific and precise.

We will specify 'such machine learning approaches'

469: "we therefore recommend"- therefore refers to previous reasoning and discussion which is absent here. Please elaborate why you believe morning scenes are better suitable before giving a recommendation. Following your argumentation- wouldn't it be possible to mitigate the effects of snow wetness changes (caused by a rise of temperatures during the day) by comparing morning to morning and evening to evening scenes?

On the contrary, 'therefore' refers to the limitations of the automated approach described in the previous paragraph:

*'Indeed, scenes unaffected by snow wetness changes (descending/morning acquisitions during the cold season) are well mapped regardless of the parameter set (Fig. 7). '*

And yes, as indicated in Table 1 (acquisition time) the scenes acquired on the same orbit are always taken at the same time of the day, so choosing morning-to-morning scenes is therefore the obvious choice. We will specify this here:

*'For future implementation of SAR detection of avalanches, we therefore recommend prioritising the use of morning-to-morning scenes'*

472: I am a bit skeptical of you being so sure about a good manual check/ correction. Could you please elaborate a bit on why you are so sure to be able to detect (true) false positives/ false negatives without additional information.

This is a good point. We will mention that even with manual edition, these scenes will likely lead to more uncertainties:

*'Although scenes acquired in the afternoon may help fill spatial and temporal gaps, it is important to note that they will require additional work to separate actual avalanche events from false positive detections caused by snow wetness changes. This is a difficult task leading to higher uncertainties for the mapping, and will likely not considerably change the long-term spatio-temporal patterns of avalanche activity (Fig. 11-13). '*

480: Didn't you exclude all deposits smaller than 4000 m2?

Good point. We will remove the second part of this sentence and remind the reader that deposits smaller than 4000 m$^2$ were filtered out i.e.:

*'The size distribution of avalanches with Sentinel-1 RGB pairs reaches a maximum around 4000 m$^2$ (avalanches smaller than 4000 m2 have been filtered and therefore not considered in this study).'*

491: How did you determine an overlap? See also comment to 362.

Please refer also to the  response to comment L. 362.

509-522: Not all discussion is related and relevant to your work, I suggest to significantly shorten this section. Additionally, I wonder if avalanches being more concentrated at lower elevations is mostly related to wetter snow conditions and better detectability (see also Eckerstorfer et al., 2022, Abermann et al., 2019). See also comments to 440ff.

We will remove the discussion on slope control on avalanching. You also have a good point regarding the detectability at lower elevations, which we will indicate here:

*'In addition, the detection at these lower elevations could also be aided by the wetter snow conditions (Eckerstorfer et al., 2022; Abermann et al., 2019). '*

538: You contradict yourself here regarding to 440ff.

We've addressed this comment in the General Points.

5.3: Generally, since one of your research questions is to "evaluate implications for the glacier mass balance" I expect you to be a bit more specific and elaborate on this a bit more.

Please see response to specific comments below.

556: Did they use whole outlines for parametrization or just deposits/part of the avalanche as can be detected from Sentinel-1?

These studies have calibrated their parametrization in a qualitative way, based on the general shape and extents of avalanche deposits visible in relatively coarse optical images. We will specify this:

*'Such calibration has been conducted in a qualitative way based on comparing the deposits from the model and the general shape and extents of deposits in a few optical images.'*

562: This sections content does not really fit the chapter headline.

On the contrary this section addresses two important points in relation to glacier mass balance:

- Glaciers with avalanches have steep headwalls, but glaciers with steep headwalls do not necessarily have avalanches. Therefore one cannot only use topographic arguments to characterize a glacier as 'avalanche-fed'. We will simplify this sentence to improve readability:

  *'At the glacier scale, we could therefore show that the presence of steep slopes within the glacier catchments is a clear necessary condition for avalanches to occur (Fig. 10, S14; Hughes, 2008; Laha et al., 2017), although not a sufficient one'*

- Avalanches are well correlated with precipitation, indicating that there is little to no snow retention from surrounding headwalls at the scale of ~1 month, which is something important to consider for the rerouting of the snow.

We will move this paragraph to the start of 5.3 for the link with the calibration of the avalanche parametrization to be made clearer.

5.3/6: I am missing a section with a throughout discussion of the limitations of your method. That would for example include that even though your methodology can detect the approximate area and frequency of avalanches, it does not give you any information on the mass of snow involved.

We argue that in 5.3 this point is apparent in the first sentence of paragraph 1:

*'While the Sentinel-1 images do not give any indication on the volume or mass of the redistributed snow, we obtained from these products key information related to the spatial extents of the avalanche deposits and the spatio-temporal variability of the avalanche activity'*

We will add a similar sentence at the end of the conclusion:

*'While it does not give any information on the mass redistributed by avalanches, our approach enables the mapping of avalanche deposits over long time periods at the scale of a small mountain range'*

573/578/589: "we successfully established a semi-automated framework"- with the manual identification of thresholds, the automatic detection and the extensive manual correction (requiring extensive domain knowledge) it remains unclear to the reader how large this benefit is. What is the gain (e.g., time) compared to full manual mapping and how much of your methodology may be reused and saves whoever uses it time (this is linked to limitations, see comment to 5.3/6). Furthermore, are there ways to translate the avalanche area into mass without additional measurements (e.g., Hynek et al., 2023) that in your case are not available.

*We have mentioned above that applying the automated approach reduces mapping time by at least half, more if only morning scenes are considered. Another major output of this study is a quality-controlled dataset of 16302 avalanche deposits in data-scarce regions, which constitutes an excellent training dataset for future method development. We will add this point to section 5.1:*

*'In the end, this study resulted in a manually checked dataset of 16,302 avalanche deposits. This dataset will be highly beneficial for the training of future mapping approaches.'*

*Regarding the translation of avalanche area into mass, we believe that using avalanche outlines to calibrate simple avalanche redistribution parametrizations is the most straightforward way to get to mass, as direct volume-area scaling without accounting for the actual precipitation amounts is unlikely to work. We agree that the best would be direct measurements of this contribution, and are working on such approach on Argentière Glacier combining high-resolution DEMs and field measurements (similar to Hynek et al., 2023), but such efforts are extremely time and resource consuming and will only ever work for 1 or 2 sites so for larger scale modelling, such combined approaches of remote sensing & modelling are better suited.*

*We will highlight these elements at the end of the conclusion:*

*'While still requiring manual checks, this approach considerably reduces the mapping effort, and the large dataset obtained will help train future mapping approaches, and calibrate mass redistribution parametrizations to be applied in the surface mass balance routines of glacio-hydrological models. '*

Could you please give an outlook on what is still needed for including avalanches into glacier models large scale with your methodology.

*This will be indicated at the end of section 5.3:*

*'Once calibrated, such avalanche redistribution parametrization can be coupled to the mass balance routine of a glacier model, for a more accurate representation of accumulation processes (Bernhardt and Schulz, 2010; Ragettli et al., 2015; Quéno et al., 2023).'*

References:

Abermann, J., Eckerstorfer, M., Malnes, E., and Hansen, B. U.: A large wet snow avalanche cycle in West Greenland quantified using remote sensing and in situ observations, Nat. Hazards, 97, 517–534, https://doi.org/10.1007/s11069-019-03655-8, 2019.

Liu, C., Li, Z., Zhang, P., Huang, L., Li, Z., and Gao, S.: Wet snow detection using dual-polarized Sentinel-1 SAR time series data considering different land categories, Geocarto International, 37, 10 907–10 924, https://doi.org/10.1080/10106049.2022.2043450, 2022.

Eckerstorfer, M., Oterhals, H. D., Müller, K., Malnes, E., Grahn, J., Langeland, S., and Velsand, P.: Performance of manual and automatic detection of dry snow avalanches in Sentinel-1 SAR images, Cold Regions Science and Technology, 198, 103 549, https://doi.org/https://doi.org/10.1016/j.coldregions.2022.103549, 2022.

Hafner, E. D., Techel, F., Leinss, S., and Bühler, Y.: Mapping avalanches with satellites – evaluation of performance and completeness, The Cryosphere, 15, 983–1004, https://doi.org/10.5194/tc-15-983-2021, 2021.

Hafner, E. D., Techel, F., Daudt, R. C., Wegner, J. D., Schindler, K., and Bühler, Y.: Avalanche size estimation and avalanche outline determination by experts: reliability and implications for practice, Natural Hazards and Earth System Sciences, 23, 2895–2914, https://doi.org/10.5194/nhess-23-2895-2023, 2023.

Hynek, B., Binder, D., Citterio, M., Larsen, S. H., Abermann, J., Verhoeven, G., Ludewig, E., and Schöner, W.: Accumulation by avalanches as significant contributor to the mass balance of a High Arctic mountain glacier, The Cryosphere Discuss. [preprint], https://doi.org/10.5194/tc-2023-157, in review, 2023.

Howell, S. E., Small, D., Rohner, C., Mahmud, M. S., Yackel, J. J., and Brady, M.: Estimating melt onset over Arctic sea ice from time series multi-sensor Sentinel-1 and RADARSAT-2 backscatter, Remote Sensing of Environment, 229, 48–59, https://doi.org/https://doi.org/10.1016/j.rse.2019.04.031, 2019.

Nagler, T., Schwaizer, G., Keuris, L., Rott, H., Luojus, K., Moisander, M., Small, D., Metsämäki, S., Malnes, E. and Eckertorfer, M.: SEOM S1-4Sci Snow: Development of Pan-European Multi-Sensor Snow Mapping Methods Exploiting Sentinel-1, Final Report, Deliverable 4.2, https://eo4society.esa.int/wp-content/uploads/2021/06/S14SciSnow.D4.2_v1_2_FR.pdf, 2021.

---

## Author Comment (AC2)

**Reviewer 2**

In this article, the authors apply methods to manually and semi-automatically map avalanche deposits across the Mt. Blanc, Everest, and Hispar regions in Sentinel-1 Synthetic Aperture Radar (SAR) imagery over a five-year period. By applying their technique, they mapped 16,302 avalanche deposits across multiple glaciers, enabling the quantification of their activity and spatio-temporal variability, thus offering vital insights into mass redistribution processes affecting glacier mass balance. The approach shows enhanced performance for images taken in winter mornings, and it indicates that avalanche deposits are mostly situated at lower elevations within glacier catchments.

I found this article to be interesting and written in polished, articulate English. The topic appears to hold significant relevance for the avalanche/glacier research community and promises to be a valuable reference for future work. The article offers a comprehensive account of the of the significant work accomplished by the authors. However, I recommend some major and minor improvements in the methods, results, and discussion sections, which I will detail and justify in the following text. Consequently, I advise a major revision of this article prior to its publication. Additional specific recommendations and corrections are outlined in the attached PDF.

We would like to thank Reviewer 2 for their thorough review and their very relevant and constructive comments.

**Major Comments**

1. **References and literature review:** The article currently relies - particularly in the introduction but also throughout the whole article - on many outdated references and lacks recent studies, notably in the context of avalanche detection using satellite data. For instance, a recent paper by Thu Trang Lê et al. (2023) demonstrates deep semantic fusion of Sentinel-1 and Sentinel-2 for snow monitoring in mountainous regions, which is highly relevant to this research. The inclusion of more current references, such as this study, is essential to validate and contextualize the findings. Some further examples to incorporate could be Sartori & Darbiri (2023) for the comparison of the methods, Guiot et al. (2023) for avalanche data from the French Alps, Liu et al. (2021) as example of avalanche detection in Asia. I highly recommend to add some more recent references.

   Thanks for these suggestions. We will make sure to add them in the text, specifically at the following locations:

   ● Refer to Guiot et al. (2023) and Sartori and Darbiri (2023) in the introduction
   ● Refer to Liu et al. (2021) and Lê et al. (2023) in the discussion relative to the use of machine learning approaches for the automated mapping of avalanches in Sentinel-1 images.

2. **Data validation with ground truth records:** The comparison between detected avalanches and actual recorded events in the three studied regions has not been sufficiently addressed. While acknowledging the limited availability of data in some areas, the integration of ground truth avalanche records, where possible, could substantially improve the credibility and reliability of the findings. Possible references

could be Guiot et al. (2023), Acharya et al. (2023) and respective regional avalanche warning services. Please consider adding a comparison or at least a thorough investigation of available ground truth avalanche records in relation to the detected avalanches.

This is a very good point, thanks for bringing it up. Data on avalanches is particularly scarce in remote glacierized regions, which is one of the reasons we decided to go for this automated mapping with Sentinel-1 images. For example, the French historical avalanche maps (CLPA: Carte des limites probables des avalanches, map of probable avalanche limits in English) do not cover the glaciers of the Mt Blanc massif due to a lack of data. We also note that while the work by Acharya et al. (2023) is tremendous and brings a nice perspective on avalanche hazard in HMA, it is biased by populated regions where avalanches were visually witnessed or caused damages. We have checked their dataset and they did not identify any avalanches in the Hispar region and only identified two avalanches in the Everest region, dating from 1997 and 1980. We therefore consider that this comparison is not really relevant to evaluate our outlines.

We did make the comparison with avalanche warning services, as shown in the Supplementary Information, figures S21-S23 for the Mt Blanc massif (shown below), although we forgot to mention it in the main text. There was indeed a good correspondence between this avalanche warning and the detected avalanche activity, at least in the winter months. We will explicitly mention this comparison in the discussion section:

*'There is also a good correspondence between the avalanche activity and the predicted avalanche danger level in the winter months (Fig. S21-23). The number and size of avalanches decreases and their minimum elevation increases in spring with rising temperatures and their dependence on precipitation and correspondence with the avalanche danger level is less strong (Fig. 11, S19-23), highlighting the transition from dry to wet avalanches (Baggi and Schweizer, 2009).'*

[Figure]

*Figure S21: One year (11/2018-10/2019) of avalanche time series over the Mt Blanc massif in the ascending and descending orbits. (a) Total area and (b) number of avalanches as a function of time across all elevations. (c) Total daily precipitation and mean daily air temperature at 3000 m a.s.l over the Mt Blanc massif according to the SAFRAN reanalysis product (Vernay et al., 2022). The red shaded areas indicate days with a predicted avalanche danger level higher than or equal to 3 (Source: Météo-France).*

[Figure]

*Figure S22: One year (11/2019-10/2020) of avalanche time series over the Mt Blanc massif in the ascending and descending orbits. (a) Total area and (b) number of avalanches as a function of time across all elevations. (c) Total daily precipitation and mean daily air temperature at 3000 m a.s.l over the Mt Blanc massif according to the SAFRAN reanalysis product (Vernay et al., 2022). The red shaded areas indicate days with a predicted avalanche danger level higher than or equal to 3 (Source: Météo-France).*

[Figure]

*Figure S23: One year (11/2020-10/2021) of avalanche time series over the Mt Blanc massif in the ascending and descending orbits. (a) Total area and (b) number of avalanches as a function of time across all elevations. (c) Total daily precipitation and mean daily air*

*temperature at 3000 m a.s.l over the Mt Blanc massif according to the SAFRAN reanalysis product (Vernay et al., 2022). The red shaded areas indicate days with a predicted avalanche danger level higher than or equal to 3 (Source: Météo-France).*

3. **Clarity in methods section:** The Methods section requires further detail and a more coherent structure to improve readability and comprehension. Presently, the steps lack information, making it challenging to follow the methodology applied. For example, it is unclear which images were used for comparison to detect avalanches. Sentinel-1 provides daily images but with different geometry (track number). However, the geometric configuration recurs every 6 or 12 days, depending on the specific region. Clarification is needed on whether only two consecutive images or a series was analysised and if daily images were taking into account. Providing, e.g., the track number would give clarity. Related to this context it should be clarified if avalanches were observed beyond 6 (or 12) days in the Sentinel-1 images.

We apologise for the lack of clarity in the methods section. Relative orbits (what you refer to as track numbers) are indicated in Table 1 of the data section. We always used the same track numbers to keep the same geometric configuration, thus the revisit times of 6 and 12 days obtained for the different regions (Table 1). This will be highlighted in the Data section:

*'We used the same orbits for each survey domain to guarantee that the incidence angles remained the same throughout the study periods.'*

In paragraph 3.1 it is specified at two occasions that the images are at 6 day intervals for the Mt Blanc and 12 days for the HMA regions.

We will specify it again in 3.4:

*'After calibration and validation of the mapping approach, we applied it to a five-year time series of Sentinel-1 images over the three survey domains (Table 1), using 6-day intervals for the Mt Blanc region and 12-day intervals for the Everest and Hispar regions.'*

4. **Performance metrics - Dice Coefficient/F1 Score:** The reported F1 score (Dice coefficient) of 0.47 for manual detection appears to be very low in comparison to the automatic detection. In general, automatic detection still lacks the manual detection behind. In addition, the F1 scores of the automatic detection are lower than F1 scores in the literature. Both points should be explained in detail in the discussion.

The F1-score of 0.47 corresponds to 2 things:

- the comparison of the aggregated Sentinel-1 manual outlines for the period 01/11/2019-09/08/2020 and the end-of-season Pléiades manual outlines from 09/08/2020. While we find this comparison interesting, it is not a central part of the analysis, and can be misleading, as indicated by reviewer 1. We will therefore remove panel a of Fig. 5 from the main text and move it to the SI. Similarly, in the text we will shift this comparison after the description of the scene-by-scene comparison.

- the average score of the ascending orbits. We note that the score of the descending orbits is much higher (average score of 0.62), as detailed in Section 4.1.2. These lower scores for afternoon scenes are discussed in Section 5.1 of the Discussion. While 0.47 is relatively low, 0.62 is quite a high score relative to the F1-scores obtained in other studies. We will make this comparison more explicit in the Discussion:

  *'The performance metrics obtained from our automated mapping approach compared to the manual detections in the Sentinel-1 outlines, have a wide range of values (F1-score between 0.29 and 0.78) depending on the season and acquisition time. These results are similar to that of other studies following similar threshold-based approaches (Leinss et al., 2020; Eckerstorfer et al., 2019; Karas et al., 2022; Wesselink et al., 2017).'*

Following these variable scores, and particularly the low ones for the ascending scenes, we manually updated our dataset to analyse the characteristics and spatio-temporal variability of avalanches. We will insist on this aspect at the start of the Results section:

*'Here, we first compare our manually derived outlines with high-resolution Pléiades images and evaluate the results and transferability of the automated mapping approach (Section 4.1). We then use the manually updated set of outlines to obtain the characteristics of avalanche deposits (Section 4.2) and their spatio-temporal variability (Section 4.3) for all three survey domains. '*

5. **Explanation of results:** The explanation of results in section 4.2 lacks clarity. Further elaboration is required to adequately convey the findings. Please refer to specific comments in the PDF.

   There are no specific comments in 4.2 in the PDF and overall very few comments in the results sections. Following the specific comments in the results, we will specify wherever necessary that only the Sentinel-1 outlines were used for the analysis, the Pléiades only being used as a qualitative check. We will thoroughly check the results sections and update the text where more clarity is needed. Specifically we will:

   - Add a sentence at the start of the results to describe the overall organisation of this section:

     *'Here, we first compare our manually derived outlines with high-resolution Pléiades images and evaluate the results and transferability of the automated mapping approach (Section 4.1). We then use the manually updated set of outlines to obtain the characteristics of avalanche deposits (Section 4.2) and their spatio-temporal variability (Section 4.3) for all three survey domains.'*

   - Remove from 4.2 any explanations that belong to the methods
   - Remove from 4.3 any interpretations that belong to the discussion

6. **Discussion:** The discussion does not address several critical issues, including the impact of radar shadow, the differences between SAR and optical data (Sentinel-1 and Pleiades images), and the low F1 scores, as mentioned before. Moreover, the comparison with actual avalanche records, although little in number, is missing and should also be added. Additionally, it is important to discuss the effects of radar shadow and layover, especially in regions located in HMA that are significantly impacted by these phenomena. A quantification of the area not taken into account due to radar shadow and layover in relation to the total investigated area should be added.

As mentioned in our answers to the previous general comments:

- We have added a comparison of the avalanche activity and the avalanche danger level in the Mt Blanc massif. No such comparison is possible for the Everest and Hispar regions.
- The F1-scores that we obtained are low for the afternoon scenes, but are in-line (or even higher) than the scores obtained by other studies. We will outline this comparison in the discussion section:

  *'The performance metrics obtained from our automated mapping approach compared to the manual detections in the Sentinel-1 outlines, have a wide range of values (F1-score between 0.29 and 0.78) depending on the season and acquisition time. These results are similar to that of other studies following similar threshold-based approaches (Leinss et al., 2020; Eckerstorfer et al., 2019; Karas et al., 2022; Wesselink et al., 2017). The performance of such approaches is generally very good in dry snow conditions, with high precision (>0.7) and low false positive rates (<0.4), which correspond to F1-scores above 0.6-0.7 (Leinss et al., 2020; Eckerstorfer et al., 2019). '*

Regarding the other points raised:

- The comparison of the Pléiades and Sentinel-1 is already discussed in detail in section 5.1 of the Discussion. We will therefore keep it as is:

  *'Our comparison of the Sentinel-1 with the Pléiades avalanche outlines indicate that avalanches detected with Sentinel-1 are of relatively large size (>4000 $m^2$ deposits) with high surface roughness, which limits the detectability to avalanches with high enough snow temperatures to form granular deposits (Steinkogler et al., 2015), or which are formed from cohesive wind slabs (Fig. 5d) or that entrain rock or ice debris, for instance from serac falls (Fig. 5c). Therefore, cold, low density snow progressively redistributed down steep rock faces or snow gullies (Sommer et al., 2015) is likely to be missed by this method, which likely also explains the upper elevation limits to avalanche detections, especially during the cold season (Fig. 11-13). Similarly, the detection of the avalanche events requires the previous deposits to have regained lower backscatter values for the signal to be visible, meaning that the surface of the deposit needs to have been smoothed by additional precipitation or melt for the next events to be visible at*

*this location. We have observed this smoothing to require several weeks and even months before avalanches can be detected at the location of old deposits, while avalanche events are still occurring in the meantime (Fig. S9d). The avalanche activity that is detected is therefore a lower bound value of the actual avalanche activity, and the aggregation of all Sentinel-1 deposits is still an underestimation of all the glacierized areas affected by gravitational snow redistribution (Fig. 5a). Nevertheless, this semi-automated approach is promising to explore the temporal and spatial variability of avalanches in remote areas, especially in glacierized regions of HMA, where close to no data exists on the occurrence of such events (Ballesteros-Cánovas et al., 2018; Caiserman et al., 2022; Acharya et al., 2023; Singh et al., 2022).'*

- Radar shadows and layover were removed from the surveyed areas (Section 3.1). We will however insist in the discussion that as a result only 57 to 72% of the surveyed areas were actually covered (as indicated by the numbers in Fig. 1):

  *'We used Sentinel-1 images to detect avalanche events, which enabled us to obtain a massif-wide distributed dataset, at least for the zones unaffected by shadow and layover (57-72% of our survey domains characterized by steep topographies), therefore less spatially biased than ground-based inventories in populated valleys (Eckert et al., 2010; Schweizer et al., 2020).'*

**Minor Comments**

1. **Figures:** Please add latitude/longitude to all figures showing details of Sentinel-1 images. Especially Fig. 1 needs a map context with an overview map showing the location of the respective insets a,b, and c. In addition, the boundaries of the used Sentinel-1 and Pleiades scenes should be added to Fig. 1a,b,and c. Country borders would be also useful addition.

   We will add latitude & longitude to all the different panels of Fig. 1, 5 (for the Pléiades panels) and 9. We will also add a context map to figure 1 (following the recommendations from reviewer 1, this context map will replace panels d and e). These regions are a small subset of Sentinel-1 scenes, and the boundaries of the Sentinel-1 tiles will not be visible in this small subset. We will also add the Pléiades boundaries to the figure. We usually refrain from indicating country borders in scientific figures, particularly for such regions which all have disputed borders.

2. **Consistency in abbreviations:** Once introduced, abbreviations should be consistently used throughout the document to ensure clarity and reduce redundancy. The parameters of the threshold calibration have not been introduced at all.

   We will consistently use the abbreviations SAR and RGI throughout the text.

   The parameters of the threshold calibration were introduced in section 3.2 and Figure 2. We will indicate this more clearly by directly referring to TS and TV for the saturation and value threshold, and by directly referring to the 2nd filtering step in Figure 2 when introducing TO.

3. **Clarification on Dice coefficient/F1 score:** I recommend using the term 'F1 score' instead of Dice coefficient due to its definition in the article. Please refer to the article of Chicco et al. (2020) for a short summary of its history.

   Agreed. We will change this in the main text, SI, and figures.

4. **Detection coverage by different sensors:** It should be noted, e.g., in the results and/or discussion, that Pleiades imagery captures the entire avalanche area, whereas SAR images may only capture part of it. Understanding this difference is critical for evaluating the outcomes of manual detection accurately.

   Thanks for pointing this out. While it is possible to map the avalanche path and rupture zone with Pléiades, it is important to note that, as mentioned in section 3.5 of the methods and section 4.1.1 of the results, we only mapped the avalanche deposits in these images. We will explicitly mention this term in the discussion as well:

   *'Our comparison of the Sentinel-1 with the Pléiades avalanche deposit outlines indicate that…'*

5. **Pearsons's correlation coefficient**: should be introduced in the methods section with formula and reference.

   This statistical coefficient is widely used throughout all scientific fields (en.wikipedia.org/wiki/Pearson_correlation_coefficient). We will add a reference to the original 1895 article where it was first described:

Pearson, K. (1895). Note on Regression and Inheritance in the Case of Two Parents. *Proceedings of the Royal Society of London Series I*, *58*, 240–242.

**Line-by-line comments**

Title: change to 'Mapping  and characterization of mountain glacier avalanches using Sentinel-1 satellite imagery'

Will be changed as suggested.

L14: 'They' -> The avalanches

Will be to 'Avalanches'.

L21-22: 'at the surface of' -> 'on'

Will be changed as suggested.

L33: I suggest to change to: Additionally, the mass balance of a glacier is traditionally expected to increase with elevation, as higher altitudes typically have colder temperatures leading to less melting and more snow accumulation (Reference).

Agreed, we will add it as suggested:

*'Mountain glaciers usually gain mass via solid precipitation falling in their accumulation area that is then advected downstream with ice flow. The mass balance of a glacier is traditionally expected to increase with elevation, as higher altitudes typically have colder temperatures leading to less melting and more snow accumulation (Benn and Lehmkuhl, 2000)'*

L46: Here I would restructure the sentence - if you are talking about avalanches- because there are observations as you state in the paragraph below. Now it sounds like there are no records of avalanches.

We will remove this sentence as it is repeated in the next paragraph.

L46: 'these events': Do you mean 'avalanches'? Please substitute if so.

Will be changed as suggested.

L59-61: Maybe the Enquête Permanente sur les Avalanches (EPA) or somethoing similar is worth mentioning here. as additional reference for avalanches in general in the Mont Blanc region

Agreed, we will add a reference to the work by Eckert et al. (2013)

L64: Some -> For example, some

Will be changed as suggested.

L72: I would add here Hafner et al. (2022)

Will be added as suggested.

L75: Sentinel-1 images are provided daily and not on demand

By near real-time we mean that avalanches can be extracted almost immediately once the images are released. We will keep it here.

L79: add Bianchi et al. (2021)

Will be added as suggested.

L84: 'high repeat frequency' -> in middle Europe it would be an image every 3 days in the same geometry. SO i would remove this part.

In our opinion this still falls within the definition of 'high repeat frequency'. Will keep it as is and mention that these are 6-12 days repeat cycles.

Maybe change to: Sentinel1 satellites are independent of light, free of charge and.. .. or sth similar

We will add 'free of charge'

L86-88: Avalanches located in areas affected by radar shadow, layover etc are also difficult to detect. Especially in regions with high mountains and steep topography this can affect the

detection results and should be mentioned, here as well as taken into account in the analysis/discussion.

We will mention it as suggested: *'or will not work in areas affected by radar shadow or layover'*

L86-88: here i would rather refer to Eckerstorfer et al. (2022) showing the descrepancy of the detection of wet-snow avalanches in SAR images

We will add the reference

L94: remove 'full'

Will be modified as suggested

L104: 'This' - PLease specify: e.g.: The steep topography/Slopes >30% etc.

Will be modified as suggested

Fig. 1: PLease add lat and lon values to (a)-(c) or a bigger map to show the location of the 3 areas as insets. It would be nice to see the overall extent of the SAR scenes int the image.

Please refer to response to general comment

L121: 'were applied the avalanche mapping' -> 'the avalanche mapping was applied'

Will be modified as suggested

L147-148: Do you mean that you took the average between the VV and VH images? In some studies VV and VH were treated separately for the different information they hold. For different application one of the two used to be more useful and avalanches can be more visible in one of the two. Did you try to detect avalanches in VV and VH separately?

This is a good point and was also pointed out by reviewer 1. We reproduce our answer to reviewer 1 below:

> We averaged the VV and VH polarizations in order to reduce the radar speckle, as has been done in previous studies (e.g. Leinss et al., 2020). However, you are correct in the sense that VV and VH have a difference in backscatter and therefore should in theory be treated separately. We had conducted some initial tests to check if it really made a difference for the automated mapping of avalanches, but results were inconclusive. Some studies have indicated that VH polarization is more suited for avalanche detection (Hafner et al., 2021), but this difference is likely most important for low incidence angles (e.g. Fig. 7 of Tompkin and Leinss, 2021), which were regions that were in part masked out when applying our shadow/layover and brightness masks (Section 3.1). When calibrating our method separately to the VV and VH RGB images for the Mt Blanc massif for the period 2019-2020, we find generally lower values than those obtained with the averaged polarizations (Table R1). This indicates that our approach is better suited to the combined VV and VH

polarizations. We will now mention this interesting point in the discussion section and add Table R1 as Table S7 to the SI:

*'Future method developments could also benefit from separating the VV and VH polarizations, particularly for regions of the SAR images with low incidence angles (Tompkin and Leinss, 2021). While in our case we obtained better results by averaging the two (Table S7), other machine learning-based approaches would likely benefit from the additional information provided by the two polarizations (Liu et al., 2022).'*

*Table R1: F1-scores obtained for the calibration of our method to VV and VH RGB triplets for the period 2019-2020 over the Mt. Blanc massif.*

| Polarisation | Path | Season | F1-score calibration |
|---|---|---|---|
| VV | Descending | November-April | 0.29 |
| | | May-October | 0.40 |
| | Ascending | November-April | 0.47 |
| | | May-October | 0.31 |
| VH | Descending | November-April | 0.17 |
| | | May-October | 0.18 |
| | Ascending | November-April | 0.14 |
| | | May-October | 0.30 |
| (VV+VH)/2 | Descending | November-April | 0.56 |
| | | May-October | 0.56 |
| | Ascending | November-April | 0.54 |
| | | May-October | 0.49 |

L154-156: Please assess the amount of area that is removed from the total area. and mention it here.

We will add it as suggested:

*'As a result, 35%, 28% and 43% of the considered area was masked out for the Everest, Mt Blanc and Hispar regions, resulting in a total area available for mapping of 492, 140 and 762 km$^2$, respectively (Fig. 1).'*

L161: Can you cite the F1 score or similar measure of he results of Karas here.

We will add it as suggested: *'this approach is well suited to identify avalanche deposits in RGB images, with a true positive rate between 0.36 and 0.58 (Karas et al., 2022)'*

L171-172: Here it is not clear what you did. You filtered the images at 2 different time steps and then differentiated the D image how exactly with the high-pass filtered images?

It is in general easier to understand to talk about activity and reference images instead of D and D-i

We apologize for the confusion, the filter was not a high pass filter but a low pass filter (smoothing). These low-pass filtered images at D and D-i correspond to Sm D and Sm D-i in the figure. We will specify it here:

*'Second, we directly differentiated the image at D with low pass filtered images at D and D-i (Sm D and Sm D-i)'*

We however argue that the notations D and D-i, introduced in the previous subsection, are well understandable and we will keep them as such.

L174-175: being -> was

Will be modified as suggested

Fig. 2: What does Sm indicate? Please mention in the caption.

Will be added to the caption: *'Sm indicates the smoothed images after application of the 45 pixel low-pass filter.'*

L177: 'images' -> VV and VH

Will be modified as suggested

L178: PLease explain the meaning of D and D-i

These were introduced in 3.1. We will add the meaning in the caption as well.

L188: (TS… TD3): Please specify he range of these values and why you chose these values.. PLease indicate what the abbreviations mean, T_S.. saturation threshold etc.

These thresholds were first introduced in 3.2 and we will indicate there the meaning of the abbreviations.

We will indicate a bit lower in the text the range of values and how they were obtained:

*'using the following ranges of value obtained from trial-and-error tests: [0.20; 0.65], [0.20; 0.65], [0.01; 0.16], [0.05; 0.11], [0.01; 0.09] and [0.31; 0.43].'*

L190: change to F1-score

Will be modified as suggested

L190: remove 'the'

Disagree. Will keep original.

L191: F1 score (also change in equation). Change throughout the text.

Will be changed throughout the text.

L191: PLease add Sørensen T. A method of establishing groups of equal amplitude in plant sociology based on similarity of species and its application to analyses of the vegetation on Danish commons. K Dan Vidensk Sels. 1948; 5(4):1–34.

Will be added as suggested

L195: true positives -> TP-

Will be modified as suggested

L196: false positives -> FP false negatives -> FN

Will be modified as suggested

L202-203: Could you please clarify in figure (e) which parts are TN, TP, FP, FN f.e. with different colors. Increasing the figure (e) would also help.

We will update the figure as suggested

L204: can you state here maybe how many pairs there were in the end?

We will specify this here: *'(~28 pairs for the Mt Blanc, ~14 for the Hispar and Everest regions for ascending and for descending scenes).'*

L210: 'Monte Carlo approach': Could you please add some information here, e.g., a reference, more specific details.

We will remove this sentence to prevent any confusion.

L210: Which parameter set are you referring to in FIgure 2? PLease specify.

All of them. We will specify 'all parameter values'

L213: What do the Sm boxes indicate in Figure 2? this should be mentioned in this paragraph as well.

We will define this in the caption and section 3.2

L215-216: Why did you not consider T_v? It has higher

We are not sure what you mean here. In any case the same plots for Tv are available in the SI. All thresholds were used in the method, we just thought we'd highlight the fact that Ts is the most sensitive here. However, we agree that this might be confusing and will therefore move Figure 4 to the SI.

L215: As a result of the threshold calibration, then saturation threshold T_s was the only ....

We believe the current version is clearer

L224-232: This part should rather be in the Data section or in 3.1

In our opinion, this is rather a methodological point and fits well here, after having focused on the automated mapping with Sentinel-1.

L242: What about the Hispar region?

No such high-resolution optical images were available for this region. We will mention it in the text:

'*For the Hispar region also, no such high-resolution (<5m) optical images were available for the study period*'

L250: Why nor 35 degrees? you use it as threshold.

30° is the value used by the two papers we refer to here and is a classic threshold value used for avalanche risk assessment (e.g. https://www.data-avalanche.org/cristal). For consistency we will keep this value.

L254: I'd rather change the section title to: Evaluation results of the manual mapping approach and 4.1 instead to: Sentinel-1 avalanche mapping or similar.

We agree that these titles could be a bit more explicit, we will make the following changes:

- '4.1 Sentinel-1 avalanche mapping' instead of 'avalanche mapping'
- '4.1.1 Comparison of Sentinel-1 and Pléiades manual detections' instead of 'Sentinel-1 avalanche mapping potential.'

L261: season without 's'

Disagree. Will keep it as is.

L261: remove (dates)

Agreed, will be removed as suggested.

L263: Do you mean 5?

All the figure numbers will be updated and carefully checked

L282: +/-: please improve this symbol: (-29 \pm 36)%

We are not using latex. This will be updated in the final typesetting stages before publication.

L286-287: These were the outlines obtained from the Sentinel-1 or the Pleiades images? Please specify!

Good point. We will specify here that these are the Sentinel-1 outlines.

L298: Was this checked with the precipitation time series or how can you link it to to snow wetness changes?

If yes, please mention it, otherwise, there can be other reasons for this false positive!

Furthermore, inferences of the results should be made in the discussion and not in the results section.

This comes from the observation of wide-spread snow backscatter changes, as shown in figure S9a. We will specify this:

*'can in some cases be linked to widespread snow backscatter increases likely due to wetness changes, especially during the May-October season (Fig. S9a)'*

We consider this to be an observation result, rather than an interpretation, and will leave it here.

L311: 'number of manually': where the Pleiades detection taken into account? Please indicate this here or earlier in the text.

This section and the following ones only refer to Sentinel-1 images. We will specify it here and will mention it also at the start of the results:

*'Here, we first compare our manually derived outlines with high-resolution Pléiades images and evaluate the results and transferability of the automated mapping approach (Section 4.1). We then use the manually updated set of outlines to obtain the characteristics of avalanche deposits (Section 4.2) and their spatio-temporal variability (Section 4.3) for all three survey domains. '*

L313: 'Pearson's correlation coefficient': This coefficient should be introduced in the Methods section with forrmula and reference!

Please refer to response to general comments

L325: 'generally above 0.5': here it would be better to judge the results if average values plus/minus stdev are reported.

Agreed. Will change as suggested.

L326: also here average values plus/minus stdev

Agreed. Will change as suggested.

Fig. 9: The color choice is not convenient. Ascending and descending are both in the red spectrum. E.g. red and blue spectra would be better distinguishable.

We will change one of the plots to blue as suggested

**References**

Acharya, A., Steiner, J.F., Walizada, K.M., Ali, S., Zakir, Z.H., Caiserman, A. and Watanabe, T., 2023. Snow and ice avalanches in high mountain Asia–scientific, local and indigenous knowledge. Natural Hazards and Earth System Sciences, 23(7), pp.2569-2592.

Chicco, D. and Jurman, G., 2020. The advantages of the Matthews correlation coefficient (MCC) over F1 score and accuracy in binary classification evaluation. BMC genomics, 21(1), pp.1-13.

Guiot, A., Karbou, F., James, G. and Durand, P., 2023. Insights into Segmentation Methods Applied to Remote Sensing SAR Images for Wet Snow Detection. Geosciences, 13(7), p.193.

Lê, T.T., Atto, A., Trouvé, E. and Karbou, F., 2023, July. Deep Semantic Fusion of Sentinel-1 and Sentinel-2 Snow Products for Snow Monitoring in Mountainous Regions. In IGARSS 2023-2023 IEEE International Geoscience and Remote Sensing Symposium (pp. 6286-6289). IEEE.

Liu, Y., Chen, X., Qiu, Y., Hao, J., Yang, J. and Li, L., 2021. Mapping snow avalanche debris by object-based classification in mountainous regions from Sentinel-1 images and causative indices. Catena, 206, p.105559.

Sartori, M. and Dabiri, Z., 2023. Assessing the Applicability of Sentinel-1 SAR Data for Semi-automatic Detection of Snow-avalanche Debris in the Southern Tyrolean Alps. GI_Forum 2023, 11, pp.59-68

---

## Author Response (AR1)

Dear Prof. Schweizer,

Many thanks for the consideration of our manuscript. We have received two well-informed reviews which are overall very positive on the quality of the analysis and relevance of the science presented in this manuscript. Both reviewers also raised some valid points especially related to the use of averaged VV and VH polarizations, the low F1-scores of the mapping approach, the lack of structure and clarity in the results or the representation of the spatio-temporal characteristics of avalanches in the main text.

We'd like to thank both reviewers for their very valuable inputs and thorough reviews which have contributed to improve the manuscript. In response to the main comments we have conducted the following changes:

- **Use of averaged VV and VH polarizations:** This choice was conducted based on performance considerations of our mapping approach, which we have now explicitly mentioned in the main text. We have also added a table to the SI showing the performance of the approach for the averaged VV and VH polarizations and for each polarization separately.
- **Low F1-scores and use of manual mapping:** We have added a number of discussion points related to the limits of the automated mapping and insisted on the need for manual mapping to analyse the spatio-temporal characteristics of avalanches. We note that the performance of our approach is aligned with that of other automated mapping methods, and that the automated mapping step still considerably reduces the manual mapping effort.
- **Organisation of the methods, results and discussions:** We have considerably streamlined the Results section to disentangle methods/results/discussions. We have also rearranged the Methods and Discussions sections to be clearer and more logical.
- **Figures:** We have reduced the number of figures from 13 to 11 and moved figures 4 and 10, as well as some of the panels from other figures to the SI. We have however kept the last three figures showing the spatio-temporal characteristics of avalanches for the three regions as we considered them to be an important outcome of this work. All figures were carefully checked and updated following the reviewers' comments.

Our answers to each specific comment are further indicated in blue below and the line numbers indicated correspond to the Manuscript with tracked changes.

We think that the manuscript has been strengthened by these revisions, but none of our main results or conclusions have changed.

Thank you for your consideration of our revised manuscript, which we hope is now acceptable for publication.

Kind regards,

Marin Kneib and Co-authors

**Reviewer 1**

Dear authors,

Thank you for the interesting work combining avalanche detection with glacier mass balance. I agree with you that remote sensing has great potential for avalanche identification that goes beyond what has been done so far. However, I have identified several points that need to be addressed before publication.

We would like to thank Reviewer 1 for his/her high quality review, and his/her very relevant and constructive comments.

Below you find a collection of the most relevant and general comments followed by specific comments referring to lines in your manuscript:

- In your processing chain for the Sentinel-1 data you state that you have averaged the polarization modes VV and VH. Various studies however have observed a difference in backscatter based on the polarization of around 5 dB (e.g., Liu et al., 2022). Why did you still choose to treat the two polarizations the same and did you check if treating polarizations differently would affect your results? (see also comment to 147ff)

  We averaged the VV and VH polarizations in order to reduce the radar speckle, as has been done in previous studies (e.g. Leinss et al., 2020). However, you are correct in the sense that VV and VH have a difference in backscatter and therefore should in theory be treated separately. We had conducted some initial tests to check if it really made a difference for the automated mapping of avalanches, but results were inconclusive. Some studies have indicated that VH polarization is more suited for avalanche detection (Hafner et al., 2021), but this difference is likely most important for low incidence angles (e.g. Fig. 7 of Tompkin and Leinss, 2021), which were regions that were in part masked out when applying our shadow/layover and brightness masks (Section 3.1). When calibrating our method separately to the VV and VH RGB images for the Mt Blanc massif for the period 2019-2020, we find generally lower values than those obtained with the averaged polarizations (Table S7). This indicates that our approach is better suited to the combined VV and VH polarizations. We have now mentioned this interesting point in the discussion section and added Table S7 to the SI (L554-557):

  *'Future method developments could also benefit from separating the VV and VH polarizations, particularly for regions of the SAR images with low incidence angles (Tompkin and Leinss, 2021). While in our case we obtained better results by averaging the two (Table S7), other machine learning-based approaches would likely benefit from the additional information provided by the two polarizations (Liu et al., 2022).'*

  *Table S7: F1-scores obtained for the calibration of our method to VV and VH RGB triplets for the period 2019-2020 over the Mt. Blanc massif.*

| *Polarisation* | *Path* | *Season* | *F1-score calibration* |
|---|---|---|---|
| | | | |

| | | | |
|---|---|---|---|
| *VV* | *Descending* | *November-April* | *0.29* |
| | | *May-October* | *0.40* |
| | *Ascending* | *November-April* | *0.47* |
| | | *May-October* | *0.31* |
| *VH* | *Descending* | *November-April* | *0.17* |
| | | *May-October* | *0.18* |
| | *Ascending* | *November-April* | *0.14* |
| | | *May-October* | *0.30* |
| *(VV+VH)/2* | *Descending* | *November-April* | *0.56* |
| | | *May-October* | *0.56* |
| | *Ascending* | *November-April* | *0.54* |
| | | *May-October* | *0.49* |

- The (description of the) process of manual avalanche mapping for parameter estimation followed by automatic detection but then again extensive manual corrections is confusing. How good was your automatic detection, why did you need to manually correct in the first place? How much time is gained by including the automatic mapping compared to going fully manual? Additionally, it is not always clear from which processing step the results presented in the result section came from.

We apologise for the lack of clarity in the overall description of our approach. It is important to note that developing a mapping approach for avalanches was only one of the focus points of this study, and that our main objective was the study of the spatio-temporal variability of these events for the three studied regions, as mentioned in the introduction (L103-106):

'To this end we (1) calibrate and evaluate our automated mapping approach at each site and assess its transferability to other sites, (2) extract the size-frequency characteristics of avalanches at various spatial scales over a period of five years and (3) evaluate the implications for the glacier mass balance.'

While the fully automated approach did lead to relatively high scores, as described in sections 4.1.2 (Table 2) and 4.1.3, there were still a number of false positive and false negative detections, which we feared may influence the final results related to the spatio-temporal variability. That is why the manual refinement was applied. Considering that 36% of the avalanche area was removed and 41% was added in this second manual step which mostly affected the descending and May-October scenes, but also that the mapping is considerably simplified when the regions of interests are already partly mapped, we consider that applying the automated approach first still reduced the mapping time by at least half (more for morning and winter scenes, which required fewer manual updates).

We have now clarified this point in the discussion section (L549-552):

*'Our automated mapping therefore still requires manual edits, although we consider that applying the automated mapping approach and then updating the outlines by hand has reduced the mapping time by at least half relative to a fully manual mapping, more if only morning scenes were to be considered.'*

We have also made sure to explicitly state what set of outlines is used in the different results sections (L298-301):

*'Here, we first compare our manually derived outlines with high-resolution Pléiades images and evaluate the performance and transferability of the automated mapping approach (Section 4.1). We then use the manually updated set of outlines to obtain the characteristics of avalanche deposits (Section 4.2) and their spatio-temporal variability (Section 4.3) for all three survey domains.'*

Furthermore, in the methods we have put the paragraph on the application of the mapping to the entire Sentinel-1 time series after the paragraph on the comparison with Pléiades, and explicitly stated the reason for the manual editing (L262-263):

*'We required highly accurate maps of avalanche deposits for the analysis of their spatio-temporal.'*

- There is a lack of clarity in your discussion regarding the detectability of dry snow avalanches (line 440ff, 456ff). You state that "the performance of such approaches is generally very good under dry snow conditions […] most difficulties for periods with wet snow conditions", this contradicts (other) previous research (e.g., Eckerstorfer et al., 2022, Abermann et al., 2019). But you also contradict yourself by writing later that "cold, low density snow […] is likely to be missed by this method, which likely also explains the upper elevation limits to avalanche detections, especially during the cold season." Please carefully reexamine the passages and literature concerning this topic and adapt your discussion.

  These apparently contradictory statements refer to two different things:

  - When avalanches are manually detected in Sentinel-1 scenes in dry snow conditions, they are usually also well mapped by the automated approach, as indicated by the high F1-scores, and there are less false positive detections than in wet snow conditions.
  - However, when we make the comparison with Pléiades optical images (Fig. 4d), we can also note that cold snow avalanches are not always detectable in

the Sentinel-1 images, as they do not always change the roughness of the snowpack much, as mentioned in the discussion section 5.1 (L505-509):

*'Our comparison of the Sentinel-1 with the Pléiades avalanche deposit outlines indicate that avalanches detected with Sentinel-1 are of relatively large size (>4000 m2 deposits) with high surface roughness, which limits the detectability to avalanches with high enough snow temperatures to form granular deposits (Steinkogler et al., 2015), or which are formed from cohesive wind slabs (Fig. 4a) or that entrain rock or ice debris, for instance from serac falls (Fig. 4c).'*

As such, these observations are in line with Eckerstorfer et al., (2019; 2022) and Abermann et al., (2019).

However, we do understand that this difference may not have been obvious in the previous version of the manuscript and we have therefore moved the paragraph concerning the comparison with the Pléiades images at the beginning of this discussion section, and highlighted the fact that the discussion of the second paragraph only applies to the comparison of the automated versus manually delineated outlines in the Sentinel-1 images (L524-525):

*'The performance metrics obtained from our automated mapping approach compared to the manual detections in the Sentinel-1 outlines, have a wide range of values…'*

- Concerning the structure of your manuscript: There are 21 sections, subsections and subsubsections which make it hard for the reader to orient. Please try to simplify and keep an eye on descriptive headings for your sections.

  Such a structure and number of sections is consistent with other articles published in The Cryosphere. We felt that it was better to have more subsections with explicit titles rather than larger sections with less obvious structure. We do however agree that some of the headings could be more explicit. We will make the following changes:

  - '3.3 Parameter calibration' instead of 'Threshold calibration'
  - '4.1 Sentinel-1 avalanche mapping' instead of 'avalanche mapping'
  - '4.1.1 Comparison of Sentinel-1 and Pléiades manual detections' instead of 'Sentinel-1 avalanche mapping potential.'

- You have included many figures, please keep only those illustrating your main points/results (see also specific comments).

  In the specific comments reviewer 1 suggests removing Fig. 9a, Fig. 10, 12 and 13 from sections 4.2 and 4.3 of the results related to the characteristics and spatio-temporal variability of detected avalanche deposits, and questions the readability/relevance of some parts of figures 1, 4 and 5 which are more related to the evaluation of the mapping approach. For the latter, we do agree that some elements may be confusing and may impact the overall readability of the overall figure. As per our answers to the specific comments, we have therefore:

  - Moved panels e and d of figure 1 to the SI as figure S1.
  - Moved figure 4 to the SI as figure S4
  - Moved panel a of figure 5 (now figure 4) to the SI as figure S11

Regarding the removal from the main text of Fig. 9a, Fig. 10, 12 and 13, which describe the characteristics and spatio-temporal variability of detected avalanche deposits, we note that while a portion of our manuscript is meant to focus on the methodological aspects of avalanche mapping with Sentinel-1 images, the other main objective was the study of the spatio-temporal variability of these events for the three studied regions, as stated in the introduction (L103-106):

*'To this end we (1) calibrate and evaluate our automated mapping approach at each site and assess its transferability to other sites, (2) extract the size-frequency characteristics of avalanches at various spatial scales over a period of five years and (3) evaluate the implications for the glacier mass balance.'*

We feel that these figures are important elements highlighting these characteristics and spatio-temporal variability and therefore do illustrate our main points/results. However, we also understand the reviewer's concern related to the overall high number of figures in the manuscript and to the readability of figures 9-13. We have therefore:

- ○ Improved the readability of Fig. 9-13 (Now Fig. 8-11) following the reviewer's specific comments
- ○ Moved Fig. 10 to the SI as figure S17

● There is a lot of spoken instead of written language used, for example "this is even more true", "indeed", "ultimately". Please stick to scientific written language, especially for starting and connecting your sentences.

We have replaced 'This is even more true' with 'This is particularly the case' and removed or replaced the adverbs 'ultimately' and 'indeed' throughout the text.

● Lastly, you have a mix of methodology, discussion and results in those three sections. Could you please go over this again and try to disentangle. If you want to keep discussion with results you could also do a section "Results and Discussion".

We have made sure to clearly separate results from discussions, following specific comments from both reviewers. Specifically, we have:

- ● Re-organized and streamlined section 4.1.1 to stick to the comparison between Pléiades and Sentinel-1 outlines
- ● Removed sentences repeated from the methods in sections 4.2 and 4.3

**Detailed comments with reference to line numbers:**

39: You make it sound like rock(fall) contributes to mass balance. Debris cover may indirectly contribute by preventing surface melt, but since you want to help calibrate glacier mass models it is weird to mention rockfall with snow/ ice avalanches as if it's the same process. I assume they are not calibrated the same way in the models. Not being able to differentiate between those mass movements in my opinion is a limitation of the chosen method (which should be mentioned in the discussion/limitations).

If we define the mass balance of a glacier as everything that comes in minus everything that comes out of a glacier, then we would argue that rockfalls do contribute to the glacier mass balance. It is true however that this is something very distinct from snow/ice avalanches (although you could imagine that these snow/ice avalanches may also transport with them

some amount of rocks). We have therefore kept this term here but added more details about the limitations that it poses in the discussion (L518-520):

*'It is also noteworthy that this mapping approach with Sentinel-1 will likely not differentiate large rockfalls on glaciers from snow avalanches, which could explain some of the activity in the summer and autumn in the Mt Blanc massif.'*

70ff: The citations on previous avalanche work in the introduction do not account for most recent work while you mention most of the recent work in the discussion/436ff. Could you please try to paint a more complete picture of recent work in the introduction and not bring "new" work only in the discussion without mentioning it before.

We have made sure that the new work also appears in the introduction. Specifically we have:

- referred to the work by Sartori and Darbiri (2023) and Guiot et al. (2023), as suggested by Reviewer 2 (L81).
- referred to recent work on the use of machine learning approaches for the automated mapping of avalanches (L87-90):

  *'More recently a number of studies have also trained machine learning approaches to improve the mapping of avalanches (Tompkin and Leinss, 2021; Waldeland et al., 2018; Yang et al., 2020; Bianchi et al., 2021; Kapper et al., 2023; Liu et al., 2021), but they have been limited by the lack of large training datasets for this application.'*

- directly referred to the work by Hafner et al. (2022, 2023) for the use of optical images to detect avalanches (L75, 527).

84: Could you please quickly explain how validity was proven and what challenges were found?

We have reformulated this sentence as follows (L85-87):

*'The validity of these approaches has been demonstrated by quantifying the overlap between outlines from Sentinel-1 images and those obtained from high-resolution optical and field observations (Leinss et al., 2020; Hafner et al., 2021).'*

For the limitations, we refer to the end of the paragraph, where we wrote (L92-94):

*'There remain limitations to these approaches, especially as they fail to detect smaller events (<4000 $m^2$) or have a high rate of false detections in the case of transitions from wet to dry snow that also result in increasing the SAR backscatter (Eckerstorfer et al., 2019; 2022; Hafner et al., 2021).'*

101: To me it is unclear how large your three study areas are. Please specify.

The size of the study areas is indicated for both ascending and descending scenes in Figure 1 (see also response to comment below). We have made sure to specify this in the caption (see response to next comment).

Figure 1: Connected to 101: are the numbers in the upper right corner your added up area of interest? Or is it the percentage of the added up?

The percentage given corresponds to the portion of glacier area covered either by ascending or by descending scenes. We selected the Sentinel-1 images so that the ascending and descending scenes always covered the entire survey area. We have made sure to specify this in the caption (L132-134):

*'The numbers in the upper right corner indicate the total area of interest covered by the ascending and descending scenes, respectively, and the third number indicates the percentage of glacierized area covered by ascending or descending scenes.'*

Please add an overview map so that every reader can get an idea where your areas of interest are located. Additionally, you need to explain RGI 6.0 at first mention and as you cannot assume that every reader knows what it is.

Agreed, we have added an overview map. And we have specified here what the RGI 6.0 stands for and cited the source (L134-135):

*'Randolph Glacier Inventory (RGI) 6.0 outlines (RGI Consortium, 2017) are shown in black'*

(d) and (e): these two graphs are quite hard to read, and I am not sure the content is essential for understanding your work. Maybe replace them with the overview map.

We have moved these panels to the SI as figure S1 and put the overview map instead.

128: It would be interesting to know if all Pleiades imagery was taken on a day Sentinel-1 was acquired also, or (and if how long) the time gap was?

Agreed, we have specified this here (L146-147):

*'The winter and August Pléiades scenes were acquired on the same day as a Sentinel-1 acquisition, while the July scene was acquired two days before the nearest Sentinel-1 acquisition.'*

147/172: Here you mention a 500m AND a 450m high pass filter? Did you filter twice? Or if these sentences describe the same operation why is the kernel size different? Additionally, a 450/500m kernel is quite large- did you perform an investigation of the effects of different filter kernel sizes on visibility of avalanches and results?

Apologies for this misleading part. These are indeed two different filters, but the 45 px one is actually a low pass filter (not a high pass, as we wrongly wrote). We have made sure to correct 'high pass' to 'low pass' there (L190-191).

We did check the influence of kernel sizes on the visibility of avalanches. We actually needed our kernel to be at least of the same order of magnitude as the largest avalanche events (up to $10^6$ m$^2$), to be able to smooth out all avalanche deposits, even the largest ones. This has been specified in the text (L191-192):

*'We selected this kernel size to be able to smooth even the largest avalanche deposits'*

This is consistent with the overall objective of this additional filtering step (L193-194):

*'The idea of this additional step was that an avalanche event results in a spatial discontinuity in the backscatter, if not with the image before, at least in the current image.'*

147ff: You have averaged the polarization modes VV and VH. Various studies however have observed a difference in backscatter based on the polarization of around 5 dB (e.g., Liu et al., 2022). Furthermore, the different polarization modes do not provide information about the same objects (e.g., HV results in a higher noise floor, see also Howell et al., 2019) and a combination of this dual-polarization data should hence be treated as an index rather than a 'normal' dB scale image as suggested in Nagler et al. (2021). Why did you treat the two polarizations the same and did you check if treating polarizations differently affects your (manual and automatic) avalanche detections.

Please refer to our response to the corresponding general comment.

154ff: How much of your area of interest was masked out when excluding shadow and layover? Could you please specify how much of your area was at the end classified glacierized and considered for analyses (and consequently is the area you are referring to in the rest of your manuscript).

We have specified it here (L172-174):

*'As a result, 35%, 28% and 43% of the considered area was masked out for the Everest, Mt Blanc and Hispar regions, resulting in a total area available for mapping of 492, 140 and 762 km$^2$, respectively (Fig. 1).'*

156: You cannot assume the reader to know what RGI 6.0 is, please explain this abbreviation.

We have specified here what this abbreviation stands for (L172).

Figure 3(e): Given recent work comparing manually mapped outlines (Hafner et al., 2023), I am wondering if the blue outline is "more correct" and how dependent it is on the operator (see also comment to 224/186).

This is a valid question and we acknowledge that manual outlines will always hold some bias, no matter how expert the operator is. This is the reason why we compared the outlines from multiple independent operators to make a first estimate of the underlying uncertainties (Fig. S1-S2, Sections 3.3, 4.1.1). However, in the absence of ground truth, as is the case for these three regions, the manual outlines had to be used as a reference for the automated mapping approach. We do agree that this point deserves more attention and we have:

- mentioned in the results the F1-score of the different operators (L335-337):

  *'The comparison of the manual outlines from four independent operators provide some insights on potential biases of the manual delineation. The F1-scores of the*

*three external operators relative to the main operator who derived the entire manual dataset for all three sites range between 0.54 and 0.66 (Table S1, Fig. S2-S3).'*

- added some explanations in the discussion and referred to the work by Hafner et al. (2023) on this topic (L524-527):

*'The performance metrics obtained from our automated mapping approach compared to the manual detections in the Sentinel-1 outlines, have a wide range of values (F1-score between 0.29 and 0.78) depending on the season and acquisition time. For most scenes, the F1-score was actually similar to those obtained by manual outlines from independent operators (Table S1, Hafner et al., 2023).'*

Figure 4: What exactly do you want to show with this?

We found this plot interesting as it showed the sensitivity of the automated mapping to the threshold values used. We however agree that it is not a major figure of this manuscript and have moved it to the SI as figure S4.

224/186: Could you please give a number for the agreement between the experts (e.g., Intersection over Union, IoU) to understand how much uncertainty is introduced if one operator manually corrects predictions (see also comment to Figure 3).

We agree that this is a crucial point, and the results of this comparison are given in the Results section L335-342. We have however specified the F1-score (rather than IoU, to be in line with the metric used in the rest of the study), which was calculated in Table S1, there as well(L335-337):

'*The comparison of the manual outlines from four independent operators provide some insights on potential biases of the manual delineation. The F1-scores of the three external operators relative to the main operator who derived the entire manual dataset for all three sites range between 0.54 and 0.66 (Table S1, Fig. S2-S3).*'

234: How did you determine a match between Pleiades and Sentinel-1? How much overlap was needed?

We are unsure whether you refer to a spatial or temporal overlap, so we answer for both:

**Spatial overlap**

For this comparison and all the calibration and validation of the methods, we directly made a pixel-by-pixel comparison, which meant that we did not need to consider the overlap of individual avalanche events. We have specified it here (L248-250):

*'We compared on a pixel-by-pixel basis the Sentinel-1 outlines that occurred over given periods in the summer and in the winter with manually derived outlines of avalanche deposits from high resolution (0.5 m) Pléiades orthoimages over part of the Mt Blanc survey area'*

And in Section 3.3 (L212-213):

*'We used the F1-score, also known as the Dice coefficient, as a metric to quantify the goodness-of-fit of the automated delineation on a pixel-by-pixel basis (Dice, 1945; Sørensen, 1948)'*

**Temporal overlap:**

To be able to compare deposits from a Sentinel-1 RGB pair with deposits from a Pléiades image, the Pléiades image needs to have been acquired as close as possible to the second Sentinel-1 image of the pair. This was the case for most Pléiades acquisitions, and we have specified this in the data section 2 (L146-147):

*'The winter and August Pléiades scenes were acquired on the same day as a Sentinel-1 acquisition, while the July scene was acquired two days before the nearest Sentinel-1 acquisition.'*

There is of course a risk that the deposits detected in the Pléiades image are anterior to the first image of the Sentinel-1 pair so for all the examples that we presented for this qualitative comparison (Fig. 5) we also checked for older Sentinel-1 deposits. We have highlighted that these were just qualitative examples in section 4.1.1 (L309-311):

*'The qualitative comparison of the manually derived Sentinel-1 deposits with the Pléiades deposits detected over time periods of ~1 month in the winter and summer seasons gives more insights on the potential of Sentinel-1 images to identify particular deposits (Fig. 4). '*

4.1.1: This chapter would fit better into the discussion, except for some specific results.

Agreed, we have re-organized and streamlined this section to stick to the direct comparison of Pléiades and Sentinel-1 outlines (L309-322):

*'The qualitative comparison of the manually derived Sentinel-1 deposits with the Pléiades deposits detected over time periods of ~1 month in the winter and summer seasons gives more insights on the potential of Sentinel-1 images to identify particular deposits (Fig. 4). It indicates locations of very good agreement, usually for large deposits with a lot of surface texture (Fig. 4c). But there are also false positive detections, for example caused by the opening of crevasses (Fig. 4b), as well as false negatives (Fig. 4a), that could reach large sizes (up to 60000 $m^2$, Fig. 4d). The comparison of the aggregation of one year of Sentinel-1 manual outlines with all the deposits identifiable in the end-of-summer Pléiades scene above 2700 m a.s.l results in a F1-score value of 0.47, with a majority of false negatives (Fig. S11). A large amount of deposits identified in Pléiades but not Sentinel-1 are actually smaller than the Sentinel-1 detectability threshold of 4000 $m^2$. Nevertheless, excluding them does not change the comparison (F1-score value of 0.49) between the Pléiades and aggregated Sentinel-1 deposits.'*

264: As mentioned in the general comments, discussion is mixed into the results, for example here as you try to give reasons for results.

As mentioned above, we have removed here any redundancy with the discussion section 5.1.

Figure 5: Even though it is known that avalanche deposits, especially large ones remain visible for a long time, it seems weird to me to compare avalanches from 1.11.2019 Sentinel-1 imagery to 9.8.2020 Pleiades. I would be very careful with this comparison as I believe a comparison of what was visible around the same time is a lot more plausible. If you compare "everything with everything" it becomes hard for the reader to follow. Hence, could you go over your analyses again and leave those not very relevant out.

In this figure, we compared the aggregation of avalanche deposits over one full year. For Sentinel-1 this was easily conducted by taking all outlines since 01/11/19 and until 09/08/20. For Pléiades, it appeared to us that in an end-of-melt season image (09/08/2020), avalanche deposits that had not already completely melted were well visible (rougher, darker surface) and likely corresponded to the accumulation of all avalanche deposits of the previous year, thus the comparison with the aggregated Sentinel-1 outlines. The comparison between the 2 (Fig. 5a) actually shows a relatively good agreement, which comforts our hypothesis and gives an interesting perspective on the mapping of on-glacier avalanche deposits. We however also agree that it doesn't fit so well with the other results shown in figure 5, so we have:

- moved panel 5a to the SI as figure S11
- shifted the first paragraph of 4.1.1 at the end of the second paragraph to give it less importance (L318-322)

276: As mentioned in the general comments, methodology is mixed into the results, like here where you mention (again) how you compared operators. The overall IoU (comment to 224) would be very nice here in addition to the consensus numbers. Additionally, how do you define an avalanche event and how do you separate it? Hafner at al. (2021) found that avalanches in Sentinel-1 might be detected in more than one blob, so just taking connected pixels might be problematic/ lead to one avalanche being counted twice or more times.

We have indicated here the F1-scores for the comparison between operators. As mentioned above, for this entire study the comparisons of outlines are made on a pixel-basis as we are not so much interested in the events, but rather in the spatial and temporal variability of avalanches. We have made sure to clarify this in the methods (L206-209):

'A single operator performed the manual detection, and to account for biases in the delineations, we compared on a pixel-by-pixel basis these outlines with those of four other operators for 4 scenes (2 ascending and 2 descending) covering the Mt Blanc region and 4 scenes covering the Everest region (Kneib et al., 2021; Table S1, Fig. S2-S3). '

We have removed from these results the repetition of the description of the methods and indicated the F1-scores of the different operators (L335-337):

'The comparison of the manual outlines from four independent operators provide some insights on potential biases of the manual delineation. The F1-scores of the three external operators relative to the main operator who derived the entire manual dataset for all three sites range between 0.54 and 0.66 (Table S1, Fig. S2-S3).'

298: The references to S9b &Co are of different style than in other places and seem to be a mixture between page number and Figure caption. There is quite a few of those throughout your manuscript, please check them all and correct them (e.g., 388, 391, 474).

These are references to figures in the Supplementary Information. In this case, panel b of supplementary Figure 9. They follow the style recommendations of the journal The Cryosphere.

312: What do you mean by referencing to S11-13? Is this supposed to be page numbers? Maybe it should be refereeing to a chapter (then it should be the chapter numbering)? There is quite a few of those throughout your manuscript, please check them all and correct them (e.g., 377, 388, 391).

As explained above, these are references to figures in the Supplementary Information.

302: Removing 36% and adding 41% is quite a lot. I already mentioned this in the general comments, could you please add a section to the discussion where you discuss the benefits of your approach despite it not being transferable and needing quite a bit of manual work.

Please refer to our answer to the general comments.

312/Figure 6: It is unclear for which variables correlation was calculated exactly. Please make this clear.

We have specified this in the figure caption (L378-379):

*'The Pearson's correlation coefficients characterizing the correlation between the validation set and the outlines from the automated mapping approach are indicated in blue (ascending) and red (descending).'*

4.1.3: Did you use the automated mapping prior to manual corrections for this analysis? Or after the step described in 302?

This section still refers to outlines from the automated mapping approach. We have specified this at the start of the results section (see response to comments above) and in the caption of Figure 6 (L392-393):

*'F1-score obtained when applying different sets of parameters to sets of images for which they were not calibrated, without any manual edits.'*

Figure 7: What is N-A and M-O for the Ascending and Descending?

This stands for November-April and for May-October. We have specified this in the caption (L395):

*'N-A and M-O stand for the November-April and May-October periods, respectively'*

347: What is R2?

This stands for coefficient of determination. We have spelled it here (L409).

Figure 8: It is not clear at first sight that the legend in (a) is true for all panels. Please move the legend outside the panels as it applies to all.

We have mentioned this at the end of the caption (L422):

'*The legend in panel (a) applies to all three panels.*'

Additionally, we have changed 'Total size of avalanche events' to 'Normalized area of all avalanche events' for the legend of panel c for clarity (L420-421).

359ff: Here you are mixing methodology and results again, please disentangle.

We have removed the first three sentences that are repeated from the methods (L423-426).

362: What is meant by activity is not clear, I assume it is repeated occurrence of an avalanche in the same place. Since it is not possible to detect whole outlines in Sentinel-1, how did you determine deposits to be "the same", in other words how much overlap did you require?

Following the comment above we have removed this sentence that is repeated from the Methods. The definition is given in Section 3.6 of the Methods, which we have modified for clarity (L284-290):

'*The union of all avalanche pixels over time indicates individual deposits affected by more or less avalanche activity. We estimated the influence of avalanches on a given glacier, independently for ascending and descending orbits, with two metrics: area affected by avalanches and avalanche activity. The area affected by avalanches is estimated by taking the union of all individual avalanche deposits, and expressed relative to glacier area. The avalanche activity is calculated for each pixel as the number of avalanches affecting this pixel over a given time period. It is then calculated on a per-deposit basis by taking the maximum activity and on a per-glacier or per-elevation band basis by taking the area of the glacier affected by avalanches divided by glacier area or area of elevation band, respectively.*'

365: You are using Change detection with a D and D-i image, with change appearing green in your data. The green hue vanishes when moving forward in time even if the deposit remains well visible/ the backscatter of a single image high. Did you analyze backscatter separately or how did you come to that conclusion?

This is a very nice way of summing up our approach. What we meant here was that when there is a new avalanche on an old deposit, it doesn't necessarily lead to an increase in surface roughness (and therefore backscatter), which means that the mapping approach does not work to detect this new avalanche. This is shown by examples in Fig. S12c-d.

Figure 9: I do not see a benefit in (a), the graph is hard to read. I believe a simple table could be a better choice for bringing your point across. Furthermore, the choice of color makes the differentiation between ASC and DESC hard (it is also not color blind safe). Additionally, areas detected in both ASC and DESC cannot be identified. To get the full picture, it would also be nice to be able to see the area that was masked as outside the glacierized or in radar shadow/layover.

For a more general answer on the value of Figs. 8a, 9-11, we refer the reviewer to our answer to the general comment on this matter. More precisely for this panel (a), we feel that it is particularly important to summarise the avalanche activity of the different individual deposits in each survey area in terms of number of events per unit area, and such information would be difficult to fit in a table. This panel also fits well with the other, more qualitative panels, that indicate the number of repeated avalanches on different avalanche deposits. We have however put a log scale for the x axis to make it more readable. We have also indicated the count in number of deposits per square kilometres to make it more comprehensive. To improve readability of the maps, we have shown the descending deposits in blue, indicated in the caption that the ascending and descending deposits overlap (this will ensure that it is colour blind safe) and indicated in the figure the shadow/layover and glacier mask.

Figure 10: This figure is hard to read. Overlapping circles cannot be distinguished and the absolute values of circles of the same size in (a), (b) und (c) differ, conveying that values are the same, though they are not. I would remove that Figure.

For a more general answer on the value of Figs. 8a, 9-11, we refer the reviewer to our answer to the general comment on this matter. We still find this figure interesting as it specifically targets the quite loose definition of what is an 'avalanche-fed' glacier. However, given the reviewer's concern about the high number of figures in our manuscript, we have shifted it to the SI. To improve the readability, we have put the circle edge colour in black so that overlapping circles are visible. We have also scaled the circles to the same size using a log scale to make them comparable between regions.

4.3: Please carefully reexamine for mixture of results with discussion.

We have removed any explanation of the linkages between avalanche activity and meteorological variables for the Mt Blanc massif.

Figure 11/12/13: I would suggest displaying only one of those in the manuscript and moving the other two to the Appendix. Furthermore, you should not use the same color range for number of avalanches and for avalanche area. Additionally, the number of avalanches is discrete (I assume you are displaying per acquisition day), while the way you display it implies a continuous scale. Lastly, the x-axis is the same for all panels, I think they would be better readable/comparable if the time would only be displayed once at the bottom of all three panels. In (c) you could improve readability by adding a thin grey line at 0° Celsius. Additionally, you should indicate data gaps in Fig 12 the same for all panels and not once white and once black.

For a more general answer on the value of Figs. 8a, 9-11, we refer the reviewer to our answer to the general comment on this matter. For former Fig. 11-13, we believe that the comparison of the avalanche activity at the three sites is one of the main results of this paper and have therefore kept all three figures in the main text, to allow visual comparison between regions. We have however improved the readability by:

- using a different colour for area and number
- using a discrete colour scale for the number of avalanches and specify in the caption that these numbers are 'for each Sentinel-1 pair'

- using one x-axis for all three panels
- adding a thin grey line at 0°C
- indicating the data gap in T & P data in white
- adding the region title at the top of the figure

440ff: "the performance is generally very good in dry snow conditions"- this contradicts findings by Eckerstorfer et al. (2022) who found a low Probability of detection for solely dry snow avalanches and you also contradict yourself in 456ff where you state that "cold, low density snow avalanches are likely to be missed". I suppose cold low density snow avalanches make up a good proportion of avalanches occurring under dry snow conditions. You also state that a rough surface is detected by its backscatter, generally wet snow avalanches tend to have a rougher surface. Could it be that the changes in overall snow wetness, wet to dry or dry to wet from D-I to D are one of the main drivers affecting detectability. For example, in Figure 11 it seems that the avalanche activity was very low for rain on snow (after a period of low temperatures) events which are generally known for critical avalanche situations and remained low until temperature conditions were stable again over a period.

Please refer to our response to the general comment for the first part of the comment.

You are absolutely correct that wet snow conditions lead to false positive detections in the case of a dry to wet snow transition. We have explicitly stated this in the discussion (L531-534):

*'The few studies that targeted extensive periods rather than a specific event also encountered the most difficulties for periods with wet snow conditions, leading to extensive false positive detections which had to be removed manually in situations of dry to wet snow transitions (Eckerstorfer et al., 2019)'*

447: "such approaches", please be specific and precise.

We will specify 'such machine learning approaches' (L537)

469: "we therefore recommend"- therefore refers to previous reasoning and discussion which is absent here. Please elaborate why you believe morning scenes are better suitable before giving a recommendation. Following your argumentation- wouldn't it be possible to mitigate the effects of snow wetness changes (caused by a rise of temperatures during the day) by comparing morning to morning and evening to evening scenes?

On the contrary, 'therefore' refers to the limitations of the automated approach described in the previous paragraph (L539-540):

*'Indeed, scenes unaffected by snow wetness changes (descending/morning acquisitions during the cold season) are well mapped regardless of the parameter set (Fig. 6). '*

And yes, as indicated in Table 1 (acquisition time) the scenes acquired on the same orbit are always taken at the same time of the day, so choosing morning-to-morning scenes is therefore the obvious choice. We have specified this here (L543-544):

*'For future implementation of SAR detection of avalanches, we therefore recommend prioritising the use of morning-to-morning scenes'*

472: I am a bit skeptical of you being so sure about a good manual check/ correction. Could you please elaborate a bit on why you are so sure to be able to detect (true) false positives/ false negatives without additional information.

This is a good point. We have mentioned that even with manual edition, these scenes will likely lead to more uncertainties (L544-547):

*'Although scenes acquired in the afternoon may help fill spatial and temporal gaps, it is important to note that they will require additional work to separate actual avalanche events from false positive detections caused by snow wetness changes. This is a difficult task leading to higher uncertainties for the mapping, and will likely not considerably change the long-term spatio-temporal patterns of avalanche activity (Fig. 9-11). '*

480: Didn't you exclude all deposits smaller than 4000 m2?

Good point. We have removed the second part of this sentence and reminded the reader that deposits smaller than 4000 $m^2$ were filtered out (L562-563):

*'The size distribution of avalanches with Sentinel-1 RGB pairs reaches a maximum around 4000 $m^2$ (avalanches smaller than 4000 m2 have been filtered and therefore not considered in this study).'*

491: How did you determine an overlap? See also comment to 362.

Please refer also to the response to comment L. 362.

509-522: Not all discussion is related and relevant to your work, I suggest to significantly shorten this section. Additionally, I wonder if avalanches being more concentrated at lower elevations is mostly related to wetter snow conditions and better detectability (see also Eckerstorfer et al., 2022, Abermann et al., 2019). See also comments to 440ff.

We have removed the discussion on slope control on avalanching. You also have a good point regarding the detectability at lower elevations, which we have indicated here (L602-603):

*'In addition, the detection at these lower elevations could also be aided by the wetter snow conditions (Eckerstorfer et al., 2022; Abermann et al., 2019).'*

538: You contradict yourself here regarding to 440ff.

We've addressed this comment in the General Points.

5.3: Generally, since one of your research questions is to "evaluate implications for the glacier mass balance" I expect you to be a bit more specific and elaborate on this a bit more.

Please see response to specific comments below.

556: Did they use whole outlines for parametrization or just deposits/part of the avalanche as can be detected from Sentinel-1?

*These studies have calibrated their parametrization in a qualitative way, based on the general shape and extents of avalanche deposits visible in relatively coarse optical images. We have specified this (L654-655):*

*'Such calibration has been conducted in a qualitative way based on comparing the deposits from the model and the general shape and extents of deposits in a few optical images.'*

562: This sections content does not really fit the chapter headline.

*On the contrary this section addresses two important points in relation to glacier mass balance:*

- *Glaciers with avalanches have steep headwalls, but glaciers with steep headwalls do not necessarily have avalanches. Therefore one cannot only use topographic arguments to characterize a glacier as 'avalanche-fed'. We have simplified this sentence to improve readability (L640-642):*

  *'At the glacier scale, we could therefore show that the presence of steep slopes within the glacier catchments is a clear necessary condition for avalanches to occur (Fig. S17-S18; Hughes, 2008; Laha et al., 2017), although not a sufficient one'*

- *Avalanches are well correlated with precipitation, indicating that there is little to no snow retention from surrounding headwalls at the scale of ~1 month, which is something important to consider for the rerouting of the snow.*

*We have moved this paragraph to the start of 5.3 for the link with the calibration of the avalanche parametrization to be made clearer.*

5.3/6: I am missing a section with a throughout discussion of the limitations of your method. That would for example include that even though your methodology can detect the approximate area and frequency of avalanches, it does not give you any information on the mass of snow involved.

*We argue that in 5.3 this point is apparent in the first sentence of paragraph 2 (L646-648):*

*'While the Sentinel-1 images do not give any indication on the volume or mass of the redistributed snow, we obtained from these products key information related to the spatial extents of the avalanche deposits and the spatio-temporal variability of the avalanche activity'*

*We have added a similar sentence at the end of the conclusion (L686-687):*

*'While it does not give any information on the mass redistributed by avalanches, our approach enables the mapping of avalanche deposits over long time periods at the scale of a small mountain range'*

573/578/589: "we successfully established a semi-automated framework"- with the manual identification of thresholds, the automatic detection and the extensive manual correction (requiring extensive domain knowledge) it remains unclear to the reader how large this benefit is. What is the gain (e.g., time) compared to full manual mapping and how much of your methodology may be reused and saves whoever uses it time (this is linked to limitations, see comment to 5.3/6). Furthermore, are there ways to translate the avalanche area into mass without additional measurements (e.g., Hynek et al., 2023) that in your case are not available.

*We have mentioned above that applying the automated approach reduces mapping time by at least half, more if only morning scenes are considered. Another major output of this study is a quality-controlled dataset of 16302 avalanche deposits in data-scarce regions, which constitutes an excellent training dataset for future method development. We have added this point to section 5.1 (L558-560):*

*'In the end, this study resulted in a manually checked dataset of 16,302 avalanche deposits, which will be highly beneficial for the training of future mapping approaches.'*

*Regarding the translation of avalanche area into mass, we believe that using avalanche outlines to calibrate simple avalanche redistribution parametrizations is the most straightforward way to get to mass, as direct volume-area scaling without accounting for the actual precipitation amounts is unlikely to work. We agree that the best would be direct measurements of this contribution, and are working on such approach on Argentière Glacier combining high-resolution DEMs and field measurements (similar to Hynek et al., 2023), but such efforts are extremely time and resource consuming and will only ever work for 1 or 2 sites so for larger scale modelling, such combined approaches of remote sensing & modelling are better suited.*

*We have highlighted these elements at the end of the conclusion (L690-692):*

*'While still requiring manual checks, this approach considerably reduces the mapping effort, and the large dataset obtained will help train future mapping approaches, and calibrate mass redistribution parametrizations to be applied in the surface mass balance routines of glacio-hydrological models. '*

Could you please give an outlook on what is still needed for including avalanches into glacier models large scale with your methodology.

*This has been indicated at the end of section 5.3 (L657-659):*

*'Once calibrated, such avalanche redistribution parametrization can be coupled to the mass balance routine of a glacier model, for a more accurate representation of accumulation processes (Bernhardt and Schulz, 2010; Ragettli et al., 2015; Quéno et al., 2023).'*

References:

Abermann, J., Eckerstorfer, M., Malnes, E., and Hansen, B. U.: A large wet snow avalanche cycle in West Greenland quantified using remote sensing and in situ observations, Nat. Hazards, 97, 517–534, https://doi.org/10.1007/s11069-019-03655-8, 2019.

Liu, C., Li, Z., Zhang, P., Huang, L., Li, Z., and Gao, S.: Wet snow detection using dual-polarized Sentinel-1 SAR time series data considering different land categories, Geocarto International, 37, 10 907–10 924, https://doi.org/10.1080/10106049.2022.2043450, 2022.

Eckerstorfer, M., Oterhals, H. D., Müller, K., Malnes, E., Grahn, J., Langeland, S., and Velsand, P.: Performance of manual and automatic detection of dry snow avalanches in Sentinel-1 SAR images, Cold Regions Science and Technology, 198, 103 549, https://doi.org/https://doi.org/10.1016/j.coldregions.2022.103549, 2022.

Hafner, E. D., Techel, F., Leinss, S., and Bühler, Y.: Mapping avalanches with satellites – evaluation of performance and completeness, The Cryosphere, 15, 983–1004, https://doi.org/10.5194/tc-15-983-2021, 2021.

Hafner, E. D., Techel, F., Daudt, R. C., Wegner, J. D., Schindler, K., and Bühler, Y.: Avalanche size estimation and avalanche outline determination by experts: reliability and implications for practice, Natural Hazards and Earth System Sciences, 23, 2895–2914, https://doi.org/10.5194/nhess-23-2895-2023, 2023.

Hynek, B., Binder, D., Citterio, M., Larsen, S. H., Abermann, J., Verhoeven, G., Ludewig, E., and Schöner, W.: Accumulation by avalanches as significant contributor to the mass balance of a High Arctic mountain glacier, The Cryosphere Discuss. [preprint], https://doi.org/10.5194/tc-2023-157, in review, 2023.

Howell, S. E., Small, D., Rohner, C., Mahmud, M. S., Yackel, J. J., and Brady, M.: Estimating melt onset over Arctic sea ice from time series multi-sensor Sentinel-1 and RADARSAT-2 backscatter, Remote Sensing of Environment, 229, 48–59, https://doi.org/https://doi.org/10.1016/j.rse.2019.04.031, 2019.

Nagler, T., Schwaizer, G., Keuris, L., Rott, H., Luojus, K., Moisander, M., Small, D., Metsämäki, S., Malnes, E. and Eckertorfer, M.: SEOM S1-4Sci Snow: Development of Pan-European Multi-Sensor Snow Mapping Methods Exploiting Sentinel-1, Final Report, Deliverable 4.2, https://eo4society.esa.int/wp-content/uploads/2021/06/S14SciSnow.D4.2_v1_2_FR.pdf, 2021.

**Reviewer 2**

In this article, the authors apply methods to manually and semi-automatically map avalanche deposits across the Mt. Blanc, Everest, and Hispar regions in Sentinel-1 Synthetic Aperture Radar (SAR) imagery over a five-year period. By applying their technique, they mapped 16,302 avalanche deposits across multiple glaciers, enabling the quantification of their activity and spatio-temporal variability, thus offering vital insights into mass redistribution processes affecting glacier mass balance. The approach shows enhanced performance for images taken in winter mornings, and it indicates that avalanche deposits are mostly situated at lower elevations within glacier catchments.

I found this article to be interesting and written in polished, articulate English. The topic appears to hold significant relevance for the avalanche/glacier research community and promises to be a valuable reference for future work. The article offers a comprehensive account of the of the significant work accomplished by the authors. However, I recommend some major and minor improvements in the methods, results, and discussion sections, which I will detail and justify in the following text. Consequently, I advise a major revision of this article prior to its publication. Additional specific recommendations and corrections are outlined in the attached PDF.

We would like to thank Reviewer 2 for their thorough review and their very relevant and constructive comments.

**Major Comments**

1. **References and literature review:** The article currently relies - particularly in the introduction but also throughout the whole article - on many outdated references and lacks recent studies, notably in the context of avalanche detection using satellite data. For instance, a recent paper by Thu Trang Lê et al. (2023) demonstrates deep semantic fusion of Sentinel-1 and Sentinel-2 for snow monitoring in mountainous regions, which is highly relevant to this research. The inclusion of more current references, such as this study, is essential to validate and contextualize the findings. Some further examples to incorporate could be Sartori & Darbiri (2023) for the comparison of the methods, Guiot et al. (2023) for avalanche data from the French Alps, Liu et al. (2021) as example of avalanche detection in Asia. I highly recommend to add some more recent references.

   Thanks for these suggestions. We have made sure to add them in the text, specifically at the following locations:

   - Referred to Guiot et al. (2023) and Sartori and Darbiri (2023) in the introduction (L81-82)
   - Referred to Liu et al. (2021) and Lê et al. (2023) in the discussion relative to the use of machine learning approaches for the automated mapping of avalanches in Sentinel-1 images (L537).

2. **Data validation with ground truth records:** The comparison between detected avalanches and actual recorded events in the three studied regions has not been sufficiently addressed. While acknowledging the limited availability of data in some

areas, the integration of ground truth avalanche records, where possible, could substantially improve the credibility and reliability of the findings. Possible references could be Guiot et al. (2023), Acharya et al. (2023) and respective regional avalanche warning services. Please consider adding a comparison or at least a thorough investigation of available ground truth avalanche records in relation to the detected avalanches.

This is a very good point, thanks for bringing it up. Data on avalanches is particularly scarce in remote glacierized regions, which is one of the reasons we decided to go for this automated mapping with Sentinel-1 images. For example, the French historical avalanche maps (CLPA: Carte des limites probables des avalanches, map of probable avalanche limits in English) do not cover the glaciers of the Mt Blanc massif due to a lack of data. We also note that while the work by Acharya et al. (2023) is tremendous and brings a nice perspective on avalanche hazard in HMA, it is biased by populated regions where avalanches were visually witnessed or caused damages. We have checked their dataset and they did not identify any avalanches in the Hispar region and only identified two avalanches in the Everest region, dating from 1997 and 1980. We therefore consider that this comparison is not really relevant to evaluate our outlines.

We did make the comparison with avalanche warning services, as shown in the Supplementary Information, figures S21-S23 for the Mt Blanc massif (shown below), although we forgot to mention it in the main text. There was indeed a good correspondence between this avalanche warning and the detected avalanche activity, at least in the winter months. We have explicitly mentioned this comparison in the discussion section (L613-617):

*'There is also a good correspondence between the avalanche activity and the predicted avalanche danger level in the winter months (Fig. S25-27). The number and size of avalanches decreases and their minimum elevation increases in spring with rising temperatures and their dependence on precipitation and correspondence with the avalanche danger level is less strong (Fig. 9, S25-27), highlighting the transition from dry to wet avalanches (Baggi and Schweizer, 2009).'*

[Figure]

*Figure S25: One year (11/2018-10/2019) of avalanche time series over the Mt Blanc massif in the ascending and descending orbits. (a) Total area and (b) number of avalanches as a function of time across all elevations. (c) Total daily precipitation and mean daily air temperature at 3000 m a.s.l over the Mt Blanc massif according to the SAFRAN reanalysis product (Vernay et al., 2022). The red shaded areas indicate days with a predicted avalanche danger level higher than or equal to 3 (Source: Météo-France).*

[Figure]

*Figure S26: One year (11/2019-10/2020) of avalanche time series over the Mt Blanc massif in the ascending and descending orbits. (a) Total area and (b) number of avalanches as a function of time across all elevations. (c) Total daily precipitation and mean daily air temperature at 3000 m a.s.l over the Mt Blanc massif according to the SAFRAN reanalysis product (Vernay et al., 2022). The red shaded areas indicate days with a predicted avalanche danger level higher than or equal to 3 (Source: Météo-France).*

[Figure]

*Figure S27: One year (11/2020-10/2021) of avalanche time series over the Mt Blanc massif in the ascending and descending orbits. (a) Total area and (b) number of avalanches as a function of time across all elevations. (c) Total daily precipitation and mean daily air*

*temperature at 3000 m a.s.l over the Mt Blanc massif according to the SAFRAN reanalysis product (Vernay et al., 2022). The red shaded areas indicate days with a predicted avalanche danger level higher than or equal to 3 (Source: Météo-France).*

3. **Clarity in methods section:** The Methods section requires further detail and a more coherent structure to improve readability and comprehension. Presently, the steps lack information, making it challenging to follow the methodology applied. For example, it is unclear which images were used for comparison to detect avalanches. Sentinel-1 provides daily images but with different geometry (track number). However, the geometric configuration recurs every 6 or 12 days, depending on the specific region. Clarification is needed on whether only two consecutive images or a series was analysised and if daily images were taking into account. Providing, e.g., the track number would give clarity. Related to this context it should be clarified if avalanches were observed beyond 6 (or 12) days in the Sentinel-1 images.

We apologise for the lack of clarity in the methods section. Relative orbits (what you refer to as track numbers) are indicated in Table 1 of the data section. We always used the same track numbers to keep the same geometric configuration, thus the revisit times of 6 and 12 days obtained for the different regions (Table 1). This has been highlighted in the Data section (L120-121):

*'We used the same orbits for each survey domain to guarantee that the incidence angles remained the same throughout the study periods.'*

In paragraph 3.1 it is specified at two occasions that the images are at 6-day intervals for the Mt Blanc and 12 days for the HMA regions.

We have specified it again in 3.5 (L260-262):

*'After calibration and validation of the mapping approach, we applied it to a five-year time series of Sentinel-1 images over the three survey domains (Table 1), using 6-day intervals for the Mt Blanc region and 12-day intervals for the Everest and Hispar regions.'*

4. **Performance metrics - Dice Coefficient/F1 Score:** The reported F1 score (Dice coefficient) of 0.47 for manual detection appears to be very low in comparison to the automatic detection. In general, automatic detection still lacks the manual detection behind. In addition, the F1 scores of the automatic detection are lower than F1 scores in the literature. Both points should be explained in detail in the discussion.

The F1-score of 0.47 corresponds to 2 things:

- the comparison of the aggregated Sentinel-1 manual outlines for the period 01/11/2019-09/08/2020 and the end-of-season Pléiades manual outlines from 09/08/2020. While we find this comparison interesting, it is not a central part of the analysis, and can be misleading, as indicated by reviewer 1. We have therefore removed panel a of Fig. 5 from the main text and moved it to the SI as figure S11. Similarly, in the text we have shifted this comparison after the description of the scene-by-scene comparison.

- the average score of the ascending orbits. We note that the score of the descending orbits is much higher (average score of 0.62), as detailed in Section 4.1.2. These lower scores for afternoon scenes are discussed in Section 5.1 of the Discussion. While 0.47 is relatively low, 0.62 is quite a high score relative to the F1-scores obtained in other studies. We have made this comparison more explicit in the Discussion (L525-529):

  *'The performance metrics obtained from our automated mapping approach compared to the manual detections in the Sentinel-1 outlines, have a wide range of values (F1-score between 0.29 and 0.78) depending on the season and acquisition time. For most scenes, the F1-score was actually similar to those obtained by manual outlines from independent operators (Table S1, Hafner et al., 2023). These results are similar to that of other studies following similar threshold-based approaches (Leinss et al., 2020; Eckerstorfer et al., 2019; Karas et al., 2022; Wesselink et al., 2017).'*

Following these variable scores, and particularly the low ones for the ascending scenes, we manually updated our dataset to analyse the characteristics and spatio-temporal variability of avalanches. We have insisted on this aspect at the start of the Results section (298-301):

*'Here, we first compare our manually derived outlines with high-resolution Pléiades images and evaluate the performance and transferability of the automated mapping approach (Section 4.1). We then use the manually updated set of outlines to obtain the characteristics of avalanche deposits (Section 4.2) and their spatio-temporal variability (Section 4.3) for all three survey domains.'*

5. **Explanation of results:** The explanation of results in section 4.2 lacks clarity. Further elaboration is required to adequately convey the findings. Please refer to specific comments in the PDF.

   There are no specific comments in 4.2 in the PDF and overall very few comments in the results sections. Following the specific comments in the results, we have specified wherever necessary that only the Sentinel-1 outlines were used for the analysis, the Pléiades only being used as a qualitative check. We have thoroughly checked the results sections and updated the text where more clarity was needed. Specifically we have:

   - Added a sentence at the start of the results to describe the overall organisation of this section (L298-301):

     *'Here, we first compare our manually derived outlines with high-resolution Pléiades images and evaluate the performance and transferability of the automated mapping approach (Section 4.1). We then use the manually updated set of outlines to obtain the characteristics of avalanche deposits (Section 4.2) and their spatio-temporal variability (Section 4.3) for all three survey domains.'*

   - Removed from 4.2 any explanations that belong to the methods

- Removed from 4.3 any interpretations that belong to the discussion

6. **Discussion:** The discussion does not address several critical issues, including the impact of radar shadow, the differences between SAR and optical data (Sentinel-1 and Pleiades images), and the low F1 scores, as mentioned before. Moreover, the comparison with actual avalanche records, although little in number, is missing and should also be added. Additionally, it is important to discuss the effects of radar shadow and layover, especially in regions located in HMA that are significantly impacted by these phenomena. A quantification of the area not taken into account due to radar shadow and layover in relation to the total investigated area should be added.

As mentioned in our answers to the previous general comments:

- We have added a comparison of the avalanche activity and the avalanche danger level in the Mt Blanc massif. No such comparison is possible for the Everest and Hispar regions.
- The F1-scores that we obtained are low for the afternoon scenes, but are in-line (or even higher) than the scores obtained by other studies. We have outlined this comparison in the discussion section (L525-531):

  *'The performance metrics obtained from our automated mapping approach compared to the manual detections in the Sentinel-1 outlines, have a wide range of values (F1-score between 0.29 and 0.78) depending on the season and acquisition time. For most scenes, the F1-score was actually similar to those obtained by manual outlines from independent operators (Table S1, Hafner et al., 2023). These results are similar to that of other studies following similar threshold-based approaches (Leinss et al., 2020; Eckerstorfer et al., 2019; Karas et al., 2022; Wesselink et al., 2017). The performance of such approaches is generally very good in dry snow conditions, with high precision (>0.7) and low false positive rates (<0.4), which correspond to F1-scores Dice values above 0.6-0.7 (Leinss et al., 2020; Eckerstorfer et al., 2019).'*

Regarding the other points raised:

- The comparison of the Pléiades and Sentinel-1 is already discussed in detail in section 5.1 of the Discussion. We therefore kept it as is (L506-524):

  *'Our comparison of the Sentinel-1 with the Pléiades avalanche outlines indicate that avalanches detected with Sentinel-1 are of relatively large size (>4000 $m^2$ deposits) with high surface roughness, which limits the detectability to avalanches with high enough snow temperatures to form granular deposits (Steinkogler et al., 2015), or which are formed from cohesive wind slabs (Fig. 5d) or that entrain rock or ice debris, for instance from serac falls (Fig. 5c). Therefore, cold, low density snow progressively redistributed down steep rock faces or snow gullies (Sommer et al., 2015) is likely to be missed by this method, which likely also explains the upper*

*elevation limits to avalanche detections, especially during the cold season (Fig. 11-13). Similarly, the detection of the avalanche events requires the previous deposits to have regained lower backscatter values for the signal to be visible, meaning that the surface of the deposit needs to have been smoothed by additional precipitation or melt for the next events to be visible at this location. We have observed this smoothing to require several weeks and even months before avalanches can be detected at the location of old deposits, while avalanche events are still occurring in the meantime (Fig. S9d). The avalanche activity that is detected is therefore a lower bound value of the actual avalanche activity, and the aggregation of all Sentinel-1 deposits is still an underestimation of all the glacierized areas affected by gravitational snow redistribution (Fig. 5a). Nevertheless, this semi-automated approach is promising to explore the temporal and spatial variability of avalanches in remote areas, especially in glacierized regions of HMA, where close to no data exists on the occurrence of such events (Ballesteros-Cánovas et al., 2018; Caiserman et al., 2022; Acharya et al., 2023; Singh et al., 2022).'*

- Radar shadows and layover were removed from the surveyed areas (Section 3.1). We have however insisted in the discussion that as a result only 57 to 72% of the surveyed areas were actually covered (as indicated by the numbers in Fig. 1):

  *L504-506: 'We used Sentinel-1 images to detect avalanche events, which enabled us to obtain a massif-wide distributed dataset, at least for the zones unaffected by shadow and layover (57-72% of our survey domains characterized by steep topographies), therefore less spatially biased than ground-based inventories in populated valleys (Eckert et al., 2010; Schweizer et al., 2020).'*

**Minor Comments**

1. **Figures:** Please add latitude/longitude to all figures showing details of Sentinel-1 images. Especially Fig. 1 needs a map context with an overview map showing the location of the respective insets a,b, and c. In addition, the boundaries of the used Sentinel-1 and Pleiades scenes should be added to Fig. 1a,b,and c. Country borders would be also useful addition.

   We have added latitude & longitude to all the different panels of Fig. 1, 4 (for the Pléiades panels) and 8. We have also added a context map to figure 1 (following the recommendations from reviewer 1, this context map replaces panels d and e). These regions are a small subset of Sentinel-1 scenes, and the boundaries of the Sentinel-1 tiles are not visible in this small subset. We have also added the Pléiades boundaries to the figure. We usually refrain from indicating country borders in scientific figures, particularly for such regions which all have disputed borders.

2. **Consistency in abbreviations:** Once introduced, abbreviations should be consistently used throughout the document to ensure clarity and reduce redundancy. The parameters of the threshold calibration have not been introduced at all.

We have consistently used the abbreviations SAR and RGI throughout the text.

The parameters of the threshold calibration were introduced in section 3.2 and Figure 2. We have indicated this more clearly by directly referring to TS and TV for the saturation and value threshold, and by directly referring to the 2nd filtering step in Figure 2 when introducing TO.

3. **Clarification on Dice coefficient/F1 score:** I recommend using the term 'F1 score' instead of Dice coefficient due to its definition in the article. Please refer to the article of Chicco et al. (2020) for a short summary of its history.

Agreed. We have changed this in the main text, SI, and figures.

4. **Detection coverage by different sensors:** It should be noted, e.g., in the results and/or discussion, that Pleiades imagery captures the entire avalanche area, whereas SAR images may only capture part of it. Understanding this difference is critical for evaluating the outcomes of manual detection accurately.

Thanks for pointing this out. While it is possible to map the avalanche path and rupture zone with Pléiades, it is important to note that, as mentioned in section 3.5 of the methods and section 4.1.1 of the results, we only mapped the avalanche deposits in these images. We have explicitly mentioned this term in the discussion as well (L506-507):

*'Our comparison of the Sentinel-1 with the Pléiades avalanche deposit outlines indicate that… '*

5. **Pearsons's correlation coefficient**: should be introduced in the methods section with formula and reference.

This statistical coefficient is widely used throughout all scientific fields (en.wikipedia.org/wiki/Pearson_correlation_coefficient). We have added a reference to the original 1895 article where it was first described (L372):

Pearson, K. (1895). Note on Regression and Inheritance in the Case of Two Parents. *Proceedings of the Royal Society of London Series I*, *58*, 240–242.

**Line-by-line comments**

Title: change to 'Mapping  and characterization of mountain glacier avalanches using Sentinel-1 satellite imagery'

Changed as suggested (L1-2).

L14: 'They' -> The avalanches

Changed to 'Avalanches' (L14).

L21-22: 'at the surface of' -> 'on'

Changed as suggested (L22).

L33: I suggest to change to: Additionally, the mass balance of a glacier is traditionally expected to increase with elevation, as higher altitudes typically have colder temperatures leading to less melting and more snow accumulation (Reference).

Agreed, added as suggested (L32-35):

*'Mountain glaciers usually gain mass via solid precipitation falling in their accumulation area that is then advected downstream with ice flow. The mass balance of a glacier is traditionally expected to increase with elevation, as higher altitudes typically have colder temperatures leading to less melting and more snow accumulation (Benn and Lehmkuhl, 2000)'*

L46: Here I would restructure the sentence - if you are talking about avalanches- because there are observations as you state in the paragraph below. Now it sounds like there are no records of avalanches.

We have removed this sentence as it is repeated in the next paragraph.

L46: 'these events': Do you mean 'avalanches'? Please substitute if so.

We have removed this sentence as it is repeated in the next paragraph.

L59-61: Maybe the Enquête Permanente sur les Avalanches (EPA) or somethoing similar is worth mentioning here. as additional reference for avalanches in general in the Mont Blanc region

Agreed, we have added L63 a reference to the work by Eckert et al. (2013)

L64: Some -> For example, some

Changed as suggested (L66).

L72: I would add here Hafner et al. (2022)

Added as suggested (L75).

L75: Sentinel-1 images are provided daily and not on demand

By near real-time we mean that avalanches can be extracted almost immediately once the images are released. We have kept it here.

L79: add Bianchi et al. (2021)

Added as suggested (L82).

L84: 'high repeat frequency' -> in middle Europe it would be an image every 3 days in the same geometry. SO i would remove this part.

In our opinion this still falls within the definition of 'high repeat frequency'. We kept it as is and mentioned that these are 6-12 days repeat cycles (L91).

Maybe change to: Sentinel1 satellites are independent of light, free of charge and.. .. or sth similar

We added 'free of charge' (L91)

L86-88: Avalanches located in areas affected by radar shadow, layover etc are also difficult to detect. Especially in regions with high mountains and steep topography this can affect the detection results and should be mentioned, here as well as taken into account in the analysis/discussion.

We will mention it as suggested: *'or will not work in areas affected by radar shadow or layover' (L94-95)*

L86-88: here i would rather refer to Eckerstorfer et al. (2022) showing the descrepancy of the detection of wet-snow avalanches in SAR images

Added the reference (L94)

L94: remove 'full'

Modified as suggested (L101)

L104: 'This' - PLease specify: e.g.: The steep topography/Slopes >30% etc.

Modified as suggested (L112)

Fig. 1: PLease add lat and lon values to (a)-(c) or a bigger map to show the location of the 3 areas as insets. It would be nice to see the overall extent of the SAR scenes int the image.

Please refer to response to general comment

L121: 'were applied the avalanche mapping' -> 'the avalanche mapping was applied'

Modified as suggested (L132)

L147-148: Do you mean that you took the average between the VV and VH images? In some studies VV and VH were treated separately for the different information they hold. For different application one of the two used to be more useful and avalanches can be more visible in one of the two. Did you try to detect avalanches in VV and VH separately?

This is a good point and was also pointed out by reviewer 1. We reproduce our answer to reviewer 1 below:

> We averaged the VV and VH polarizations in order to reduce the radar speckle, as has been done in previous studies (e.g. Leinss et al., 2020). However, you are correct in the sense that VV and VH have a difference in backscatter and therefore should in theory be treated separately. We had conducted some initial tests to check if it really made a difference for the automated mapping of avalanches, but results were inconclusive. Some studies have indicated that VH polarization is more suited for avalanche detection (Hafner et al., 2021), but this difference is likely most important for low incidence angles (e.g. Fig. 7 of Tompkin and Leinss, 2021), which

were regions that were in part masked out when applying our shadow/layover and brightness masks (Section 3.1). When calibrating our method separately to the VV and VH RGB images for the Mt Blanc massif for the period 2019-2020, we find generally lower values than those obtained with the averaged polarizations (Table S7). This indicates that our approach is better suited to the combined VV and VH polarizations. We have now mentioned this interesting point in the discussion section and added Table S7 to the SI (L554-557):

*'Future method developments could also benefit from separating the VV and VH polarizations, particularly for regions of the SAR images with low incidence angles (Tompkin and Leinss, 2021). While in our case we obtained better results by averaging the two (Table S7), other machine learning-based approaches would likely benefit from the additional information provided by the two polarizations (Liu et al., 2022).'*

*Table S7: F1-scores obtained for the calibration of our method to VV and VH RGB triplets for the period 2019-2020 over the Mt. Blanc massif.*

| Polarisation | Path | Season | F1-score calibration |
|---|---|---|---|
| VV | Descending | November-April | 0.29 |
| | | May-October | 0.40 |
| | Ascending | November-April | 0.47 |
| | | May-October | 0.31 |
| VH | Descending | November-April | 0.17 |
| | | May-October | 0.18 |
| | Ascending | November-April | 0.14 |
| | | May-October | 0.30 |
| (VV+VH)/2 | Descending | November-April | 0.56 |
| | | May-October | 0.56 |
| | Ascending | November-April | 0.54 |
| | | May-October | 0.49 |

L154-156: Please assess the amount of area that is removed from the total area. and mention it here.

Added as suggested (L172-174):

*'As a result, 35%, 28% and 43% of the considered area was masked out for the Everest, Mt Blanc and Hispar regions, resulting in a total area available for mapping of 492, 140 and 762 km$^2$, respectively (Fig. 1).'*

L161: Can you cite the F1 score or similar measure of he results of Karas here.

Added as suggested (L179-180): *'this approach is well suited to identify avalanche deposits in RGB images, with a true positive rate between 0.36 and 0.58 (Karas et al., 2022)'*

L171-172: Here it is not clear what you did. You filtered the images at 2 different time steps and then differentiated the D image how exactly with the high-pass filtered images?

It is in general easier to understand to talk about activity and reference images instead of D and D-i

We apologize for the confusion, the filter was not a high pass filter but a low pass filter (smoothing). These low-pass filtered images at D and D-i correspond to Sm D and Sm D-i in the figure. We have specified it here (L190-191):

*'Second, we directly differentiated the image at D with low pass filtered images at D and D-i (Sm D and Sm D-i)'*

We however argue that the notations D and D-i, introduced in the previous subsection, are well understandable and we kept them as such.

L174-175: being -> was

Modified as suggested (L193).

Fig. 2: What does Sm indicate? Please mention in the caption.

Added to the caption (L200-201): *'Sm indicates the smoothed images after application of the 45 pixel low-pass filter.'*

L177: 'images' -> VV and VH

Modified as suggested (L197)

L178: PLease explain the meaning of D and D-i

These were introduced in 3.1. Added the meaning in the caption as well (L197-198).

L188: (TS… TD3): Please specify he range of these values and why you chose these values.. PLease indicate what the abbreviations mean, T_S.. saturation threshold etc.

These thresholds were first introduced in 3.2 and we have indicated there the meaning of the abbreviations.

We have indicated a bit lower in the text the range of values and how they were obtained (L234-236):

*'using the following ranges of value obtained from trial-and-error tests: [0.20; 0.65], [0.20; 0.65], [0.01; 0.16], [0.05; 0.11], [0.01; 0.09] and [0.31; 0.43].'*

L190: change to F1-score

Modified throughout the text.

L190: remove 'the'

Disagree. Kept original.

L191: F1 score (also change in equation). Change throughout the text.

Changed throughout the text.

L191: PLease add Sørensen T. A method of establishing groups of equal amplitude in plant sociology based on similarity of species and its application to analyses of the vegetation on Danish commons. K Dan Vidensk Sels. 1948; 5(4):1–34.

Added as suggested (L213)

L195: true positives -> TP-

Modified as suggested (L217)

L196: false positives -> FP false negatives -> FN

Modified as suggested (L218)

L202-203: Could you please clarify in figure (e) which parts are TN, TP, FP, FN f.e. with different colors. Increasing the figure (e) would also help.

Figure 3 updated as suggested

L204: can you state here maybe how many pairs there were in the end?

We have specified this here (L227-228): *'(~28 pairs for the Mt Blanc, ~14 for the Hispar and Everest regions for ascending and for descending scenes).'*

L210: 'Monte Carlo approach': Could you please add some information here, e.g., a reference, more specific details.

We have removed this sentence to prevent any confusion (L234).

L210: Which parameter set are you referring to in FIgure 2? PLease specify.

We have removed this sentence to prevent any confusion (L234).

L213: What do the Sm boxes indicate in Figure 2? this should be mentioned in this paragraph as well.

We have defined this in the caption and section 3.2

L215-216: Why did you not consider T_v? It has higher

We are not sure what you mean here. In any case the same plots for Tv are available in the SI. All thresholds were used in the method, we just thought we'd highlight the fact that Ts is the most sensitive here. However, we agree that this might be confusing and have therefore move Figure 4 to the SI as figure S4.

L215: As a result of the threshold calibration, then saturation threshold T_s was the only ....

We believe the current version is clearer

L224-232: This part should rather be in the Data section or in 3.1

In our opinion, this is rather a methodological point and fits well here, after having focused on the automated mapping with Sentinel-1.

L242: What about the Hispar region?

No such high-resolution optical images were available for this region. We have mentioned it in the text (L258):

'*For the Hispar region also, no such high-resolution (<5m) optical images were available for the study period*'

L250: Why nor 35 degrees? you use it as threshold.

30° is the value used by the two papers we refer to here and is a classic threshold value used for avalanche risk assessment (e.g. https://www.data-avalanche.org/cristal). For consistency we have kept this value.

L254: I'd rather change the section title to: Evaluation results of the manual mapping approach and 4.1 instead to: Sentinel-1 avalanche mapping or similar.

We agree that these titles could be a bit more explicit, we have made the following changes:

- '4.1 Sentinel-1 avalanche mapping' instead of 'avalanche mapping'
- '4.1.1 Comparison of Sentinel-1 and Pléiades manual detections' instead of 'Sentinel-1 avalanche mapping potential.'

L261: season without 's'

Disagree. Kept as is.

L261: remove (dates)

Agreed, removed.

L263: Do you mean 5?

All the figure numbers were updated and carefully checked

L282: +/-: please improve this symbol: (-29 \pm 36)%

We are not using latex. This will be updated in the final typesetting stages before publication.

L286-287: These were the outlines obtained from the Sentinel-1 or the Pleiades images? Please specify!

Good point. We have specified here that these are the Sentinel-1 outlines (L345).

L298: Was this checked with the precipitation time series or how can you link it to to snow wetness changes?

If yes, please mention it, otherwise, there can be other reasons for this false positive!

Furthermore, inferences of the results should be made in the discussion and not in the results section.

This comes from the observation of wide-spread snow backscatter changes, as shown in figure S12a. We have specified this (L356-357):

*'can in some cases be linked to widespread snow backscatter increases likely due to wetness changes, especially during the May-October season (Fig. S12a)'*

We consider this to be an observation result, rather than an interpretation, and will leave it here.

L311: 'number of manually': where the Pleiades detection taken into account? Please indicate this here or earlier in the text.

This section and the following ones only refer to Sentinel-1 images. We have specified it here and at the start of the results (L298-301):

*'Here, we first compare our manually derived outlines with high-resolution Pléiades images and evaluate the performance and transferability of the automated mapping approach (Section 4.1). We then use the manually updated set of outlines to obtain the characteristics of avalanche deposits (Section 4.2) and their spatio-temporal variability (Section 4.3) for all three survey domains.'*

L313: 'Pearson's correlation coefficient': This coefficient should be introduced in the Methods section with forrmula and reference!

Please refer to response to general comments

L325: 'generally above 0.5': here it would be better to judge the results if average values plus/minus stdev are reported.

We have specified 'above 0.5 in 78% of cases' (L385)

L326: also here average values plus/minus stdev

Changed to 'lower than 0.5 in 92% of cases' (L387)

Fig. 9: The color choice is not convenient. Ascending and descending are both in the red spectrum. E.g. red and blue spectra would be better distinguishable.

Changed the descending outlines to blue spectrum as suggested.

**References**

Acharya, A., Steiner, J.F., Walizada, K.M., Ali, S., Zakir, Z.H., Caiserman, A. and Watanabe, T., 2023. Snow and ice avalanches in high mountain Asia–scientific, local and indigenous knowledge. Natural Hazards and Earth System Sciences, 23(7), pp.2569-2592.

Chicco, D. and Jurman, G., 2020. The advantages of the Matthews correlation coefficient (MCC) over F1 score and accuracy in binary classification evaluation. BMC genomics, 21(1), pp.1-13.

Guiot, A., Karbou, F., James, G. and Durand, P., 2023. Insights into Segmentation Methods Applied to Remote Sensing SAR Images for Wet Snow Detection. Geosciences, 13(7), p.193.

Lê, T.T., Atto, A., Trouvé, E. and Karbou, F., 2023, July. Deep Semantic Fusion of Sentinel-1 and Sentinel-2 Snow Products for Snow Monitoring in Mountainous Regions. In IGARSS 2023-2023 IEEE International Geoscience and Remote Sensing Symposium (pp. 6286-6289). IEEE.

Liu, Y., Chen, X., Qiu, Y., Hao, J., Yang, J. and Li, L., 2021. Mapping snow avalanche debris by object-based classification in mountainous regions from Sentinel-1 images and causative indices. Catena, 206, p.105559.

Sartori, M. and Dabiri, Z., 2023. Assessing the Applicability of Sentinel-1 SAR Data for Semi-automatic Detection of Snow-avalanche Debris in the Southern Tyrolean Alps. GI_Forum 2023, 11, pp.59-68

---

## Referee Report (RR1)

**Review 2 of the manuscript 'egusphere-2023-2007':**

**Mapping and characteristics of avalanches on mountain glaciers with Sentinel-1**

Marin Kneib, Amaury Dehecq, Fanny Brun, Fatima Karbou, Laurane Charrier, Silvan Leinss, Patrick Wagnon, Fabien Maussion

Submitted for publication in The Cryosphere

**Comments to the authors**

The authors have clearly invested substantial effort in this study, covering three distinct regions characterized by diverse climatological and topographical features. The analyses conducted are thorough, delving into numerous details. However, for future work, I suggest streamlining the presentation of results to prioritize certain analysis over others. This approach would enable a more focused exploration of fewer topics, facilitating deeper, clearer analysis and discussion. Nevertheless, the manuscript has undergone significant improvements particularly in readability and structure. Figures 1, 3, and 4 have improved in terms of size or geographic context. Furthermore, both, the results and discussion sections have been streamlined, and the overall structure has improved through renaming the (sub)sections.

Upon reviewing the revised script, I've provided some comments and suggestions for further improvement in the manuscript and supplementary material PDFs. These suggestions aim at refining the content further. Consequently, I recommend publication of the manuscript following the implementation of these suggested revisions.

1. **Comparison to actual avalanche data**: I understand the difficulty in finding ground truth avalanche records for comparison. However, a comparison to data from, e.g., the Mt. Blanc Massif would have strengthened the results from the semi-automated detection and should be considered for future work. For example, the ANENA (Association Nationale pour l'Étude de la Neige et des Avalanches) provides yearly publicily available avalanche reports from 2013 onwards, which might be worth considering. I appreciate the effort of the comparison to avalanche risk levels, although they are not the same as a comparison to ground truth data.
2. **Clarity about relative orbit:** While Table 1 provides clarity regarding the images used, why not consider using daily images to enhance avalanche tracking. Of course, only images from same relative orbit can be compared, and coverage of different relative orbits are not always the same**,** but a smaller intervals between images could

have offered a higher temporal resolution, potentially tracking avalanches and their transformations, including those affected by wind.

3. **Low F1 coefficient:** The F1 score ranges as stated in the manuscript: *"above 0.5 in 78% of cases (0.6 in 83% of the cases for the Hispar November-April scenes). The ascending scenes present in general lower F1-scores (lower than 0.5 in 92% of cases), particularly the May-October scenes of Everest for which the F1-score never exceeds 0.32. With an average F1-score of 0.46, the Everest descending November-April parameter set is the most transferable, but still performs poorly (F1-score<0.4)"* and *"F1-score between 0.29 and 0.78"*. I still am convinced that these values are rather low compared to articles using e.g., machine learning. For instance, Bianchi et al. (2021) achieved scores surpassing 0,66, while Hafner et al. (2022) reported 0,625 across diverse topographical regions. While a brief mention of the low values is provided in one paragraph of the discussion, I find that a critical and comprehensive analysis elucidating the underlying reasons for these comparatively lower values is lacking. Especially, the part: *"The performance of such approaches is generally very good in dry snow conditions, with high precision (>0.7) and low false positive rates (<0.4), which correspond to F1-scores above 0.6-0.7 (Leinss et al., 2020; Eckerstorfer et al., 2019). The few studies that targeted extensive periods rather than a specific event also encountered the most difficulties for periods with wet snow conditions, leading to extensive false positive detections which had to be removed manually in situations of dry to wet snow transitions (Eckerstorfer et al., 2019)."* needs to be modified. Please see my comments to this in the annotated PDF. A rectification of the low F1 scores should provide compelling justifications for the decision to refrain from employing machine learning methods in this work.

4. **Reasons behind the lower ascending and descending F1 scores:** In the discussion, I suggest including a brief explanation of potential factors contributing to the lower F1 score observed in the ascending compared to the descending scenes.

5. **Fig. 5**: It is difficult to distinguish between the different lines for the manually and automatically detected avalanches as well as ascending and descending scenes. Is it necessary to separate ascending from descending, because earlier you mention that "*The automated mapping generally underestimates the number and sizes of the avalanche deposits ..*" without distinguishing between the two. Combining the two would make the figure clearer to read.

6. **Fig. 8b-e***:* Yin the caption you mention that the shaded black areas were excluded (masked out) from the analysis. Nevertheless, there seems to be an overlap with some dark blue color that I assume are detected avalanche deposits in the descending views. If you did not take the avalanches in the shaded area into account, these (descending) overlapped areas should be removed. Furthermore, the dark blue does not appear in the color legend. Moreover, the clarity of the figure could be enhanced by combining the ascending and descending views - considering that there is no reference of this categorization in the text specifically related to this figure, and no discernible difference appears to exist. There even seems to be some overlap in parts of the descending and ascending detected avalanches. Did you count them separately? Did you use them to confirm the detected avalanches?

[revised manuscript text omitted]

---

## Author Response (AR2)

**Editor**

Dear authors

I have now received the comments by the reviewers on your revised manuscript.

Please consider those and make according revisions.

I will then make a final decision.

Best regards,

Jürg Schweizer.

Dear Prof. Schweizer,

Many thanks for the consideration of our manuscript. We would also like to thank the reviewers for their comments that helped improve the manuscript in terms of scientific content and readability. There remained a few suggestions, which helped us refine some of the text. In response to the main comments we have conducted the following changes:

- We have added more details to the discussion to clarify the difference between cold and wet snow conditions and the implications for the mapping (F1 scores) and possible resulting biases
- We have added a figure to the SI showing an example of high backscatter values caused by avalanches remaining visible for > 6 months on Hispar Glacier

Our answers to each specific comment are further indicated in blue below and the line numbers indicated correspond to the revised Manuscript with tracked changes.

We think that the manuscript has been strengthened by these revisions, but none of our main results or conclusions have changed.

Thank you for your consideration of our revised manuscript.

Kind regards,

Marin Kneib and co-authors

**Reviewer 1**

We would like to thank Reviewer 1 for their high quality review, and their very relevant and constructive comments.

Dear authors,

Thank you very much for the work you put into improving your manuscript. It reads a lot smoother and is noticeably clearer than the previous version. I have a few points to raise (still). Especially the first one is at the core of your work needing improvement before this may be published:

In your answer to my comments, you differentiate which statements regarding dry/wet snow refer to which processing step/process. There the difference is clearly stated and described while the manuscript is still lacking this clarity. Furthermore, in the same context, you need to elaborate and discuss the effect of snow wetness (detectability of dry or wet snow avalanches) on your results better and clearly put it in the context of previous work stating some of their findings explicitly, comparing and discussing. For example, 561 (1-2 months delay in activity) also points in the direction that it is easier to map the wet snow avalanches.

We apologise if that difference was not made clear enough in the manuscript. We have now added a sentence to the discussion to make this explicitly clear (L494-495):

'Therefore, while dry snow conditions lead to detectability limitations in Sentinel-1 images (Fig. 4), when avalanches are manually detected in Sentinel-1 scenes in dry snow conditions, they are usually also well mapped by the automated approach, as indicated by the high F1-scores (Eckerstorfer et al., 2022). '

We have now also further detailed the effects that snow wetness may have on the observed patterns (potential observational biases), L561-563:

'the detection at these lower elevations could be aided by the wetter snow conditions, leading to lower backscatter background values that are favourable for the avalanche detection (Eckerstorfer et al., 2022; Abermann et al., 2019).'

And L576-578:

'These relatively high values in Spring could partly originate from a bias in the avalanche detection, as low backscatter background values (wet snow) make it easier to detect avalanche deposits (Eckerstorfer et al., 2022; Abermann et al., 2019).'

Regarding the 1-2 months delay in avalanche activity that you refer to, we do not think that it could be caused by an observational bias as in this region (Everest) the monsoon (precipitation peak) occurs during the warm period (temperature peak), so if there is a bias in detection of events caused by dry to wet snow transition, there should be no delay.

You have corrected a lot of the passages with "spoken language", but at the same time you have introduced some new ones with your corrections e.g., "was actually" or "are actually". Please reexamine all newly inserted sentences and avoid spoken language in the whole manuscript.

Apologies for this, we have removed the term 'actually' L289 & L489.

229: The notation of the values in [] is not clear. Please explain what you mean in an easier to understand way and connect well to the sentences around.

We have now specifically mentioned ([min; max]) for the range of values to make this clearer. We have also indicated which parameters these correspond to: ($T_S$, $T_V$, $T_O$, $T_{D1}$, $T_{D2}$ and $T_{D3}$). L230-232.

464: Please include in this discussion a sentence or two that all avalanches occurring on top of eachother within 6d/12d will be counted as one, additionally lowering the total number of detected avalanches.

This was already stated L473-475:

'Similarly, the detection of the avalanche events requires the previous deposits to have regained lower backscatter values for the signal to be visible, meaning that the surface of the deposit needs to have been smoothed by additional precipitation or melt for the next events to be visible at this location.'

We have made this even more explicit L475-477: 'We have observed this smoothing to require several weeks and even months before avalanches can be detected at the location of old deposits, while avalanche events are still occurring in the meantime and are therefore difficult to detect (Fig. S12d).'

**Reviewer 2**

The authors have clearly invested substantial effort in this study, covering three distinct regions characterized by diverse climatological and topographical features. The analyses conducted are thorough, delving into numerous details. However, for future work, I suggest streamlining the presentation of results to prioritize certain analysis over others. This approach would enable a more focused exploration of fewer topics, facilitating deeper, clearer analysis and discussion. Nevertheless, the manuscript has undergone significant improvements particularly in readability and structure. Figures 1, 3, and 4 have improved in terms of size or geographic context. Furthermore, both, the results and discussion sections have been streamlined, and the overall structure has improved through renaming the (sub)sections. Upon reviewing the revised script, I've provided some comments and suggestions for further improvement in the manuscript and supplementary material PDFs. These suggestions aim at refining the content further. Consequently, I recommend publication of the manuscript following the implementation of these suggested revisions.

We would like to thank Reviewer 2 for their thorough review and their very relevant and constructive comments.

**1. Comparison to actual avalanche data:** I understand the difficulty in finding ground truth avalanche records for comparison. However, a comparison to data from, e.g., the Mt. Blanc Massif would have strengthened the results from the semiautomated detection and should be considered for future work. For example, the ANENA (Association Nationale pour l'Étude de la Neige et des Avalanches) provides yearly publicily available avalanche reports from

2013 onwards, which might be worth considering. I appreciate the effort of the comparison to avalanche risk levels, although they are not the same as a comparison to ground truth data.

We agree that more validation data would have been welcome, but as mentioned in the first round of review, this is particularly difficult in HMA. For the Mt Blanc massif, we already noted in our previous response that 'the French historical avalanche maps (CLPA: "Carte des limites probables des avalanches", map of probable avalanche limits in English) do not cover the glaciers of the Mt Blanc massif due to a lack of data.' We can add here that even in non-glacierized regions these maps are generally too coarse to compare to (they also include not just the deposit areas but also the release and propagation areas) - see figure below. We have looked for the ANENA reports you refer to but only found reports of avalanche casualties, which are very sparse and give limited information on the spatial characteristics of the avalanche deposit (in general the location of the event is not even given). For this particular study we therefore considered such a comparison to be of limited value.

[Figure]

Figure R1: CLPA maps of the Mt Blanc region. CLPA couloirs in red, glacier outlines (our area of interest) in black. There is very little overlap between the two and the avalanche couloirs are relatively coarse.

**2. Clarity about relative orbit:** While Table 1 provides clarity regarding the images used, why not consider using daily images to enhance avalanche tracking. Of course, only images from same relative orbit can be compared, and coverage of different relative orbits are not always the same, but a smaller intervals between images could have offered a higher

temporal resolution, potentially tracking avalanches and their transformations, including those affected by wind.

We thank the reviewer for sharing their idea: it would be a nice way to get more details on some of the events. However, the revisit times for most of these survey domains at relatively low latitudes remain limited to 6 and 12 days, even considering other orbits, as there is very little overlap between tracks. This is for example visible in these figures showing the different Sentinel-1 tracks:

[Figure]

Figure R2: Sentinel-1 geographical coverage in 2020 over High Mountain Asia. Background image from Google Earth.

[Figure]

Figure R3: Sentinel-1 geographical coverage in 2020 over Central Europe. Background image from Google Earth.

**3. Low F1 coefficient:** The F1 score ranges as stated in the manuscript: "above 0.5 in 78% of cases (0.6 in 83% of the cases for the Hispar November-April scenes). The ascending scenes present in general lower F1-scores (lower than 0.5 in 92% of cases), particularly the May-October scenes of Everest for which the F1-score never exceeds 0.32. With an average F1-score of 0.46, the Everest descending NovemberApril parameter set is the most transferable, but still performs poorly (F1- score<0.4)" and "F1-score between 0.29 and 0.78". I still am convinced that these values are rather low compared to articles using e.g., machine learning. For instance, Bianchi et al. (2021) achieved scores surpassing 0,66, while Hafner et al. (2022) reported 0,625 across diverse topographical regions. While a brief mention of the low values is provided in one paragraph of the discussion, I find that a critical and comprehensive analysis elucidating the underlying reasons for these comparatively lower values is lacking. Especially, the part: "The performance of such approaches is generally very good in dry snow conditions, with high precision (>0.7) and low false positive rates (<0.4), which correspond to F1-scores above 0.6-0.7 (Leinss et al., 2020; Eckerstorfer et al., 2019). The few studies that targeted extensive periods rather than a specific event also encountered the most difficulties for periods with wet snow conditions, leading to extensive false positive detections which had to be removed manually in situations of dry to wet snow transitions (Eckerstorfer et al., 2019)." needs to be modified. Please see my comments to this in the annotated PDF. A rectification of the low F1 scores should provide compelling justifications for the decision to refrain from employing machine learning methods in this work.

The high scores that you refer to for Hafner et al. (2022) are obtained for automated mapping with SPOT 6/7 optical images, this therefore is not comparable to our mapping with Sentinel-1 images - SPOT images provide high resolution optical images allowing a robust detection of avalanches, but need to be tasked and have therefore much less temporal and spatial coverage than Sentinel-1. Bianchi et al. (2021) did achieve higher scores with machine learning, although it is unclear from their study if these were impacted by the snow wetness changes. In any case, this agrees with our statement acknowledging the low F1-scores in certain conditions and saying that machine learning approaches provide a promising way forward to reduce false positive detections in situations of snow wetness changes (L498-501):

'Such false positive detections can be discarded manually based on size and texture considerations, which indicates that deep learning approaches based on convolutional neural networks, for example, offer a promising way to improve these classifications (Tompkin and Leinss, 2021; Waldeland et al., 2018; Yang et al., 2020; Bianchi et al., 2021; Kapper et al., 2023; Liu et al., 2021; Lê et al., 2023).'

Please see our answer below for details on the low F1 scores in wet to dry snow transition conditions.

**4. Reasons behind the lower ascending and descending F1 scores:** In the discussion, I suggest including a brief explanation of potential factors contributing to the lower F1 score observed in the ascending compared to the descending scenes.

We have added a brief explanation in the discussion (L503-506):

'Indeed, scenes unaffected by snow wetness changes (descending/morning acquisitions during the cold season) are well mapped regardless of the parameter set (Fig. 6). Ascending scenes, acquired in the afternoon, are more likely to be affected by snow wetness changes than descending scenes, acquired in the morning. This explains the lower F-1 scores for these scenes.'

**5. Fig. 5:** It is difficult to distinguish between the different lines for the manually and automatically detected avalanches as well as ascending and descending scenes. Is it necessary to separate ascending from descending, because earlier you mention that "The automated mapping generally underestimates the number and sizes of the avalanche deposits .." without distinguishing between the two. Combining the two would make the figure clearer to read.

We did not want to combine ascending and descending detections as the acquisition dates differ and as the temporal variability shows strong differences between the two, which are worth showing, also for Fig. S14-S16 of the SI. We tried separating figure 5 into four subplots instead of two but this did not improve the visibility of the lines.

**6. Fig. 8b-e:** In the caption you mention that the shaded black areas were excluded (masked out) from the analysis. Nevertheless, there seems to be an overlap with some dark blue color that I assume are detected avalanche deposits in the descending views. If you did not take the avalanches in the shaded area into account, these (descending) overlapped areas should be removed. Furthermore, the dark blue does not appear in the color legend. Moreover, the clarity of the figure could be enhanced by combining the ascending and descending views - considering that there is no reference of this categorization in the text

specifically related to this figure, and no discernible difference appears to exist. There even seems to be some overlap in parts of the descending and ascending detected avalanches. Did you count them separately? Did you use them to confirm the detected avalanches?

Thanks for pointing this out, these small overlaps have now been removed. We have also specified what the mask corresponds to in the caption (L404-405):

'The shaded grey areas correspond to the intersection of the ascending and descending masks.'

The dark blue colour does appear in the legend at the bottom of the figure. We choose to separately treat descending and ascending orbits throughout the manuscript given that they indicated slightly different temporal patterns and that they corresponded to different acquisition dates - we therefore did count them separately. As a result we kept the separation in this figure as well. As described in the results/discussions, we counted them separately and chose not to use them as a confirmation although this is a good idea for future studies and we have now mentioned it in the discussion (L510-511):

'Although scenes acquired in the afternoon may help fill spatial and temporal gaps and could be used as a confirmation for some detections,'

We have also improved the readability of the figure by showing only contour lines spaced every 200 m and reducing the transparency of the masked areas.

**Line-by-line comments**

L14: seldom->seldomly

We respectfully disagree. Seldom is the right adverb, could be replaced with rarely but we've kept seldom here (L14)

L51: 'wind-blown from steep headwalls': I would remove this part as you are speaking of differences of avalanches in glacierized catchments versus off-glacier avalanches.

Done (L51)

L75: 'near real-time': this is still misleading, also in the article by Eckerstorfer et al. (2019) near-real time suggests the avalanche to be detected right after release.

We've removed this term (L75)

L79: increases -> increased

We use here the noun rather than the verb, and have therefore kept it as it is (L82).

L86-87: 'but they have been limited by the lack of large training datasets for this application.': The limits are rather due to the resolution of the Sentinel1 images, the low repeat-frequency in middle Europe, the detection of dry snow, the algorithm, shadow-effects and layover etc. There is at least one large dataset that could be used for training. PLease modify.

We have re-organized this paragraph to make the link clearer between the use of machine learning approaches and the limitations from Sentinel-1 images. We've also removed the limitation caused by the lack of training datasets (L87-94):

'More recently a number of studies have also trained machine learning approaches to improve the mapping of avalanches (Tompkin and Leinss, 2021; Waldeland et al., 2018; Yang et al., 2020; Bianchi et al., 2021; Kapper et al., 2023; Liu et al., 2021). There remain limitations to these approaches, especially as they fail to detect smaller events (<4000 m$^2$) or have a high rate of false detections in the case of transitions from wet to dry snow that also result in increasing the SAR backscatter (Eckerstorfer et al., 2019; 2022; Hafner et al., 2021), or will not work in areas affected by radar shadow or layover.'

L87: remove 'high'

Changed to 'Sentinel-1 satellites have a repeat frequency of 6-12 days for low latitude regions (European Alps and HMA)'

L87:  (6-12 days repeat cycles): please specify about which areas you are talking. When both Sentinel-1 satellites where running, repeat frequency was daily in northern latitudes

Changed to 'Sentinel-1 satellites have a repeat frequency of 6-12 days for low latitude regions (European Alps and HMA)' (L79-80)

L88: remove 'free of charge': repetition and the satellites are not free of charge

Done (L80).

L118: replace 'the same orbits for each survey domain' with 'only one orbit for ascending and descending track respectively being 6 or 12 days apart depending on the survey domain.'

Changed as suggested (L119-121): 'We used one orbit for the ascending and descending tracks, respectively 6 or 12 days apart depending on the survey domain, to guarantee that the incidence angles remained the same throughout the study periods.'

Figure 1: again, in a-c it would be nice to see the complete extent of the Sentinel-1 ascending and descending scenes, since there are only 2 it would have been easy to show them

As stated in our previous response: 'These regions are a small subset of Sentinel-1 scenes, and the boundaries of the Sentinel-1 tiles are not visible in this small subset. ' We show as an example a screenshot from the ASF datasearch showing in blue the different S1 tiles and our survey domain in the yellow polygon:

[Figure]

Figure R4: ASF data explorer for Sentinel-1 images over the Everest region (yellow polygon). Overlapping Sentinel-1 tiles are shown in red (ascending) and blue (descending).

Based on the scale difference, we do not see the added value of having the Sentinel-1 footprints displayed in figure 1.

L251: remove 'All Sentinel-1 images were pre-processed in Google Earth Engine.'. Repetition, already mentioned in 3.1.

Done (L255).

L298: +/-: this still needs to be improved.

This will be modified in the final proofing stage.

L313-314: 'linked to widespread snow backscatter increases likely due to wetness changes, especially during the May-October season': i still would remove this, because it is an assumption that has not been checked and should be in the discussion

We agree in principle, but this is still required to justify the manual filtering of the false positive and false negative detections that comes right after: 'Such false positive or false negative detections were manually removed or added based on considerations of shape, size and location, and this manual filtering was applied to all time series of all survey domains for the results presented in sections 4.2 and 4.3.' (L320-322). We have therefore kept it as it is.

L328-329: 'Pearson's correlation coefficient': please add formula, although it might be a common formula. maybe it fits inline

Considering that this metric is widely used and that any information about it can easily be found online, we consider that the reference to the original study is enough here (L333).

L402: ' interannual variability of deposit activity is not very strong': I find it quite strong comparing e.g. 2017 and 2020.

Agreed, modified as (L413-414): 'There are pronounced seasonal differences (Fig. 9-11, S20-S22, Table S4) enhanced by the interannual variability of deposit activity (Fig. S19).'

L407-408: 'There are avalanches all year round over the Mt Blanc massif, but with a higher activity between January and July (Fig. 9).': It is very difficult to see this trend in Fig. 9. One has to search the rectangles with lighter red and then sort of count how many other rectangles there are for each column to be able to compare the numbers of avalanches over time.. I suggest to either improve the figure or rather refer to Fig. S23-26, in which this information can be seen more clearly.. The years, at least for the Mt. Blanc differ very much for each year (e.g. 2019 has very low numbers of avalanches compared to other years). Where are the same figures for the other 2 regions?

As suggested, we've now referred to Fig. S24-S28 (L420). These figures were meant to show the comparison with the avalanche forecast from Meteo France and as such we did not include these figures for the Hispar and Everest sites to not overburden the SI, given that the patterns for these 2 sites are also much clearer in Fig. 9-10.

L449: no -> little

Changed as suggested (L462)

L458: is -> are?

'Snow' is singular. We've kept 'is'. (L471)

L462-464: 'We have observed this smoothing to require several weeks and even months before avalanches can be detected at the location of old deposits, while avalanche events are still occurring in the meantime (Fig. S12d)': I find it very unlikely that it takes weeks to observe new avalanches on top of the old ones. I rather think, the opposite is the case. Usually, large wind speeds and precipitation can make it hard to detect avalanches within the 6 or 12 days window, removing traces of the avalanches before the next image is provided.

This is what we would have expected as well, but in some cases (serac falls? Large wet snow avalanches?) the high backscatter values from the avalanche deposit signal remain visible for long time periods. We have added the following figure (showing an avalanche deposit visible during 8 months in the Hispar region) in the SI and indexed it in the main text to highlight this:

[Figure]

*Figure S14: (a) Detection of avalanche deposits, with an older deposit visible in light purple, Hispar RGB composite 04/09/2018-16/09/2018. The high backscatter values from these avalanche deposits remained visible until 19/06/2019 (b-d). The different RGB bands range between -25 and -6 dB.*

L476-480: This part needs to be modified. Currently, it sounds like it is easier to detect dry snow avalanches in Sentinel-1 images, and difficult to detect wet avalanches. PLease refer to Eckerstorfer te al. (2022): Performance of manual and automatic detection of dry snow avalanches in Sentinel-1 SAR images

Here the goal was to highlight the fact that when avalanches are manually detected in Sentinel-1 scenes in dry snow conditions, they are usually also well mapped by the automated approach, as indicated by the high F1-scores, and there are less false positive detections than in wet snow conditions. Apologies if this part is not clear, we have added the following sentence to make this clearer (L493-495):

'Therefore, while dry snow conditions lead to detectability limitations in Sentinel-1 images (Fig. 4), when avalanches are manually detected in Sentinel-1 scenes in dry snow conditions, they are usually also well mapped by the automated approach, as indicated by the high F1-scores (Eckerstorfer et al., 2022). '

L496-497: 'Our automated mapping therefore still requires manual edits,': so this would be a semi-automated mapping…

We've changed it to 'semi-automated' (L516)

L522-525: 'This difference is likely due to the detectability threshold, as well as the fact that recurring avalanches are likely to be missed if the surface roughness does not change between two events (Fig. S12c-d)': As mentioned above , I'would reformulate the sentence. High wind speed and new precipitation can mask avalanches within the rather large intervals of up to 12 days of Sentinel-1. Even 6 days can already be a long period if there are large wind speeds. I would also assume that avalanches on glaciers show differences than avalanches on snow.

Please refer to our response above and the figure we've added in the SI showing that high backscatter values from an avalanche deposit can remain visible for > 6 months. What you mention about the masking of deposits by wind/precipitation is however absolutely correct and we've added a sentence to make this clear (L477-478):

'In other cases, high wind speeds or new precipitation are likely to mask the deposits in the time interval of 6 to 12 days.'

L565: 'leading to high backscatter values that may reduce the detectability of avalanches, and especially slab avalanches (Fig. 4)': This needs to me modified: higher backscatter usually increases the detectability of avalanches

Here we referred to the high **background** backscatter values, not the backscatter values from the avalanches. We've modified the sentence to make this explicit (L587-589):

'This seasonality in avalanche activity could partly be explained by the presence of cold and dry snow at high elevations in the winter, leading to high backscatter background values that may reduce the detectability of avalanches, and especially slab avalanches (Fig. 4), in these upper reaches.'

Fig. S19: d-e -> d-f? F-g -> g-i?

Changed, thanks!